# Probing single electrons across 300-mm spin qubit wafers

Samuel Neyens[1,2 ✉], Otto K. Zietz[1,2], Thomas F. Watson[1], Florian Luthi[1], Aditi Nethwewala[1], Hubert C. George[1], Eric Henry[1], Mohammad Islam[1], Andrew J. Wagner[1], Felix Borjans[1], Elliot J. Connors[1], J. Corrigan[1], Matthew J. Curry[1], Daniel Keith[1], Roza Kotlyar[1], Lester F. Lampert[1], Mateusz T. Mądzik[1], Kent Millard[1], Fahd A. Mohiyaddin[1], Stefano Pellerano[1], Ravi Pillarisetty[1], Mick Ramsey[1], Rostyslav Savytskyy[1], Simon Schaal[1], Guoji Zheng[1], Joshua Ziegler[1], Nathaniel C. Bishop[1], Stephanie Bojarski[1], Jeanette Roberts[1] & James S. Clarke[1 ✉]

Building a fault-tolerant quantum computer will require vast numbers of physical qubits. For qubit technologies based on solid-state electronic devices[1–3], integrating millions of qubits in a single processor will require device fabrication to reach a scale comparable to that of the modern complementary metal–oxide–semiconductor (CMOS) industry. Equally important, the scale of cryogenic device testing must keep pace to enable efficient device screening and to improve statistical metrics such as qubit yield and voltage variation. Spin qubits[1,4,5] based on electrons in Si have shown impressive control fidelities[6–9] but have historically been challenged by yield and process variation[10–12]. Here we present a testing process using a cryogenic 300-mm wafer prober[13] to collect high-volume data on the performance of hundreds of industry-manufactured spin qubit devices at 1.6 K. This testing method provides fast feedback to enable optimization of the CMOS-compatible fabrication process, leading to high yield and low process variation. Using this system, we automate measurements of the operating point of spin qubits and investigate the transitions of single electrons across full wafers. We analyse the random variation in single-electron operating voltages and find that the optimized fabrication process leads to low levels of disorder at the 300-mm scale. Together, these results demonstrate the advances that can be achieved through the application of CMOS-industry techniques to the fabrication and measurement of spin qubit devices.

Silicon quantum dot spin qubits[1,4,5] have recently demonstrated single-qubit and two-qubit fidelities well above 99% (refs. 6–9), satisfying thresholds for error correction[14]. Today, integrated spin qubit arrays have reached sizes of six quantum dots[9,15], with larger quantum dot platforms in 1D (refs. 16,17) and 2D (refs. 18,19) configurations also being demonstrated. To realize practical applications with spin qubit technology, physical qubit count will need to be increased substantially[20,21]. This will require fabricating spin qubit devices with a density, volume and uniformity comparable with those of classical computing chips, which today contain billions of transistors. The spin qubit technology has inherent advantages for scaling owing to the qubit size (approximately 100 nm), as well as—in the case of Si-based devices—a native compatibility with CMOS manufacturing infrastructure. It has therefore been posited that manufacturing spin qubit devices with the same infrastructure as classical computing chips can unlock the potential of spin qubits for scaling and provide a path to building fault-tolerant quantum computers with the technology.

The scaling of classical chips according to Moore's law has depended on substantial advancements in device variation ($\sigma V_T$)[22], as well as performance ($I_{on}/I_{off}$, gate delay). For spin qubits today, process variation and yield are notable challenges[10–12]. Although state-of-the-art results are impressive[6–9], associated platforms do not yet include studies of device yield. In practice, most spin qubit results are achieved as a culmination of a device screening process in which many devices are tested until one with satisfactory electrostatic behaviour is obtained. As the spin qubit field progresses towards larger array sizes, such processes will become more challenging as increasing numbers of gates and quantum dot sites must pass these screening criteria. Advancing to the next order of magnitude in spin qubit processor size will demand both higher yield of spin qubit device components (for example, gates, quantum dots), as well as more efficient testing processes to tackle the increasingly complex fabrication process optimization.

It has not yet been clearly shown that CMOS manufacturing infrastructure can bring the same improvements to variation and yield of quantum devices as have been made for classical devices. Spin qubits have been made with hybrid fabrication flows, in which industry-standard techniques are interleaved with research techniques such as e-beam lithography and/or lift-off[23,24]. More fully industry-compatible devices in Si-MOS have also been demonstrated[25,26] but are at present limited by high levels of disorder owing to the qubits being formed directly at the Si/SiO2 interface. Spin qubits hosted in epitaxial group IV heterostructures offer reduced disorder[27–29] but are less straightforward to integrate

[1]Intel Corp., Hillsboro, OR, USA. [2]These authors contributed equally: Samuel Neyens, Otto K. Zietz. ✉e-mail: samuel.neyens@intel.com; james.s.clarke@intel.com

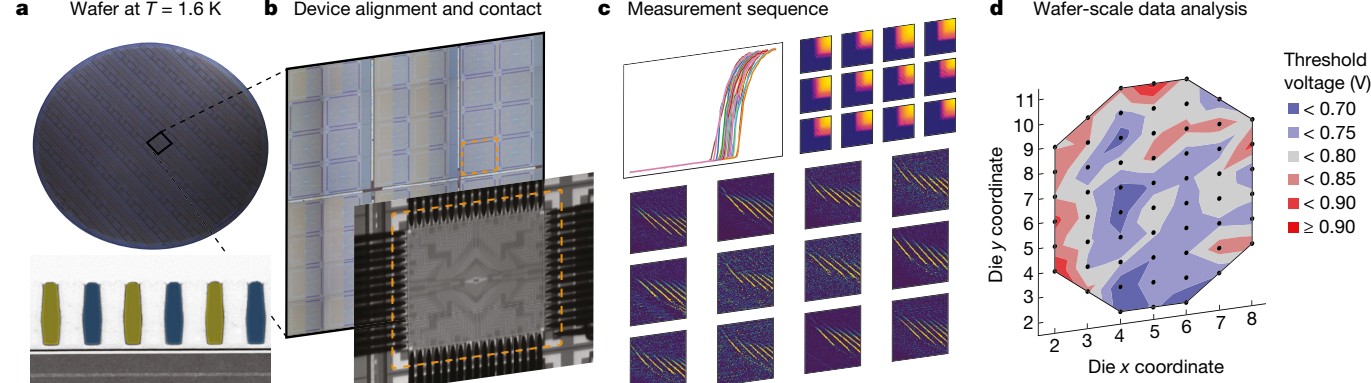

**a** Wafer at $T$ = 1.6 K  **b** Device alignment and contact  **c** Measurement sequence  **d** Wafer-scale data analysis

Threshold voltage (V)
< 0.70
< 0.75
< 0.80
< 0.85
< 0.90
≥ 0.90

Die $x$ coordinate

Die $y$ coordinate

**Fig. 1 | Cryo-prober measurement flow. a**, The cryo-prober cools 300-mm wafers (upper image) to an electron temperature of 1.6 K in around 2 h. Lower image shows a cross-sectional transmission electron micrograph of a Si/SiGe quantum dot qubit device. Gates are false-coloured. Scale bar, 100 nm. **b**, When the wafer is cold, device pads are aligned to the probe pins using wafer stage controls and machine vision feedback. The stage lifts the device pads into contact with the probe pins to connect devices to measurement electronics at room temperature. Device pads are 100 × 100 μm² with 150-μm pitch. **c**, With the device in contact, a wide variety of measurements can be performed to extract device data. **d**, After repeating this process on many devices across the wafer, device data can be used for statistical analysis of wafer-scale trends.

in an industry process owing to the 300-mm SiGe epitaxy, which comes with reduced thermal budget and increased valley-splitting challenges[30] compared with Si-MOS.

As well as fabrication challenges, the bottleneck of cryogenic electrical testing presents a barrier to scaling any solid-state quantum technology, from spin qubits to superconducting[2] and topological[3] qubits. To improve process variation and yield in quantum devices, process changes must be combined with statistical measurements of performance indicators such as voltage variation and component yield. Furthermore, as spin qubit processor size increases, it will be increasingly important to identify the 'leading edge' devices from a given wafer before packaging in a quantum computer stack, requiring thorough testing of a large volume of devices per wafer. Traditional test systems that cool down one device at a time introduce substantial overhead (through dicing, die attaching, bonding and thermal cycling devices), which limits the number of devices per wafer that can be tested. One solution is device multiplexing, using either on-chip[31,32] or off-chip[33] circuitry, but both approaches come with limitations in the wafer area that can be sampled. By contrast, the standard technique in the semiconductor test industry is full-wafer probing. This approach provides maximal flexibility, as all devices on the wafer are simultaneously accessible for electrical measurement. For quantum devices, wafer-scale probing requires further cooling hardware to reach the required temperatures. For spin qubits based on Si/SiGe quantum dots, accessing the single-electron operating regime typically requires temperatures ≲4 K. Only recently has wafer probing at such low temperatures become possible.

In this work, we present two advancements. First, we develop a 300-mm cryogenic probing process to collect high-volume data on spin qubit devices across full wafers. Second, we optimize an industry-compatible process to fabricate spin qubit devices on Si/SiGe heterostructures, combining low process variation with a low-disorder host material. These two advancements are mutually reinforcing: the development of full-wafer cryogenic test capabilities enables the optimization of the complex 300-mm fabrication process and the optimization of the fabrication process improves device reliability to enable much deeper automated measurements across wafers. As we will show, together these culminate in the automated probing of single electrons in spin qubit arrays across 300-mm wafers.

The spin qubit devices studied here are fabricated in Intel's D1 factory, in which the company's CMOS logic processes are developed. The host material is a Si/Si$_{0.7}$Ge$_{0.3}$ heterostructure[34] grown on 300-mm Si wafers. This structure is chosen to exploit the long-lived coherence of electron spins in Si and their applicability for several qubit encodings[5]. Figure 1a shows an optical image of a completed spin qubit wafer. All patterning is done with optical lithography. The quantum dot gate patterning is done in a single pass with extreme ultraviolet lithography, allowing us to explore gate pitches from 50 to 100 nm. The fabrication of all device sub-components is based on fundamental industry techniques of deposition, etch and chemical-mechanical polish[35]. As we will demonstrate, this approach leads to high yield and low process variation across the 300-mm wafer.

The cryogenic wafer prober (cryo-prober) we use[13,36] was manufactured by Bluefors and AEM Afore and was developed in collaboration with Intel. The cryo-prober can load and cool 300-mm wafers to a base temperature of 1.0 K at the chuck and an electron temperature of 1.6 ± 0.2 K (see Extended Data Fig. 2) in around 2 h. Figure 1 shows an overview of the wafer measurement process. After cooldown, thousands of spin qubit arrays and test structures on the wafer are available for measurement. An individual device is aligned to the probe pins using the wafer stage control and a machine vision algorithm. The wafer is brought into contact with the probe pins to electrically connect device pads to voltage sources and current and voltage detectors at room temperature. Measurements are taken with these instruments to extract a variety of metrics, including gate-line resistance, ohmic contact resistance, carrier mobility, gate threshold voltage and transition voltages in the few-electron regime (see Methods and Extended Data Fig. 3 for measurements of gate-line resistance and carrier mobility). These measurements are repeated on many devices across a wafer to generate wafer-scale statistics. The entire process, from alignment to device measurement, is fully automated and programmable, speeding up device data collection by several orders of magnitude compared with the measurement of singular devices in a cryostat.

To achieve high yield, a combination of processes from industrial transistor manufacturing is used. A 3D schematic of the gate stack is shown in Fig. 2a. The quantum dots are defined by a planar architecture. Active gates, used for controlled accumulation, are defined in a single layer. In later devices (discussed below), a second passive layer for screening/depletion is also integrated[37]. The gate electrodes are isolated from the heterostructure by a high-dielectric-constant composite stack, or 'high-κ stack', whereas neighbouring gates are isolated by a 'spacer' stack. Complete process optimization involves many factors; here we highlight two key approaches for improving device variation and performance: reducing fixed charge in the high-κ stack and optimizing the gate layer architecture. Fixed charge in the high-κ stack can arise as a result of the materials and conditions of the

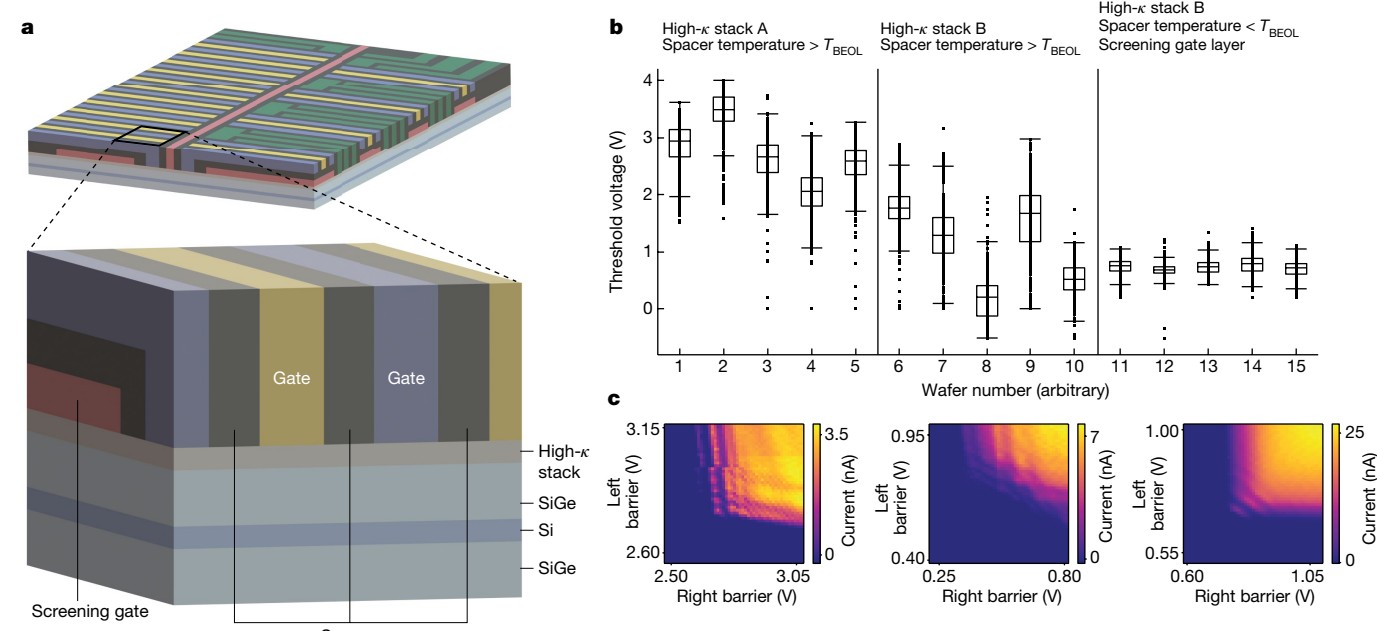

**Fig. 2 | Process optimization aided by cryo-prober feedback. a**, Schematic of the gate structure in an optimized spin qubit array. Gates designed to accumulate charge are coloured yellow, blue and green, whereas gates designed to deplete charge (screening gates) are coloured red. Scale in the vertical direction is approximate. **b,c**, Spin qubit device variation and electrostatics performance are improved through optimization of the gate stack. Three versions of the device fabrication are highlighted. Only the third version includes the lower screening layer shown in **a**. **b**, Gate $V_T$ variation both within and between wafers is improved after process optimization. Box plots show the median and interquartile range of each distribution. Whiskers mark the maximum and minimum values excluding outliers, which are defined as points removed from the median by more than 1.5 times the interquartile range. **c**, Representative quantum dot transport measurements are shown for each of the three versions, with improvements made to disorder and stability.

deposition itself, as well as through exposure to subsequent processing. In particular, we find that fixed charge can be reduced in our devices by limiting the temperature of the spacer process to within the typical thermal budget for back-end-of-line (BEOL) processing, $T_{BEOL} \approx 400$ °C (ref. 38). We attribute the reductions in fixed charge to reduced crystallization of the high-$\kappa$ stack at lower temperatures. Figure 2b shows improvements in flatband voltage variation over 15 wafers, as measured by gate threshold voltage ($V_T$), or the voltage required to turn on and off current with a particular gate (see Methods for measurement details). This plot highlights three distinct versions of the fabrication flow and includes approximately 4,000 data points for each version. Across these versions, we observe a marked reduction in median $V_T$ and a reduction in $V_T$ variation between and within wafers. We attribute these improvements to the reduction in fixed charge, driven by improvements to the high-$\kappa$ stack itself (between stacks A and B) and the reduction in thermal budget of subsequent processing, as well as to the more consistent confinement provided by the extra screening gate layer. The barrier–barrier scans shown in Fig. 2c also highlight improvements in quantum dot confinement, disorder and stability through each stage of device optimization (see Methods and Extended Data Fig. 7 for more).

After process optimization, we characterize the optimized process flow with measurements on 12-quantum-dot (12QD) devices with 60-nm gate pitch. Measurements are again fully automated to maximize the speed and consistency of data collection (see Methods). The 12QD design consists of a linear array of 12 quantum dots with four opposing sensor dots isolated by a centre screening gate. An inline scanning electron microscopy image of this device with a schematic of the measurement configuration is shown in Fig. 3b. Quantum dots on both the qubit side and the sensor side are defined by three gates each: one plunger gate to control the electron number on the dot and one barrier gate on each side to tune the tunnel coupling to the neighbouring dot or charge reservoir. The array of 12 quantum dots can be operated as

physical qubits in a variety of spin encodings, including single spin qubits[39] (in a 12-qubit array) or exchange-only qubits[40] (in a four-qubit array). Depending on the spin qubit encoding, an optional micromagnet layer can be added to the device and the centre screening gate can supply microwave electric fields to control the qubits with electric dipole spin resonance.

As in a CMOS logic process, improving qubit yield is a necessary part of scaling up quantum processors, as larger systems will depend on an increasing number of qubit components to function. To analyse the yield of this fabrication flow, we test 232 12QD devices across a wafer. We calculate component yield for ohmic contacts, gates, quantum dots and full 12QD devices. These yield metrics are summarized in Table 1. Both ohmic contact and gate yield are 100%. The large number of gates tested and working on this wafer (>10,000) highlights the consistency of the gate fabrication process. Quantum dot yield is 99.8%, which further emphasizes the reliability of electrostatic gate control. Last, the full device yield, including the linear array of 12 quantum dots and the four charge sensors, is 96% (see Methods for more details).

Figure 3c shows a summary of gate $V_T$ values collected on 12QD devices across a wafer. The distributions are highly consistent across the 25-gate array. We also observe a systematic shift in median $V_T$ for the two outermost gates in the array. The symmetry of this effect suggests that it is electrostatic in nature, owing to the proximity of the reservoir gates. Although trends such as this might be difficult to confirm through one-off device testing, they are readily observable with full-wafer statistics. The gate $V_T$ distributions also contain information on process variation. To estimate the random variation in $V_T$ within individual devices, we adapt a standard CMOS industry method of analysing matched-pair $V_T$ differences[22] (see Methods for details). The resulting matched-pair $\Delta V_T$ distribution is plotted in Fig. 3d. The standard deviation of this distribution, reduced by a factor of $\sqrt{2}$, is 59 mV, representing the random component of $V_T$ variation within devices.

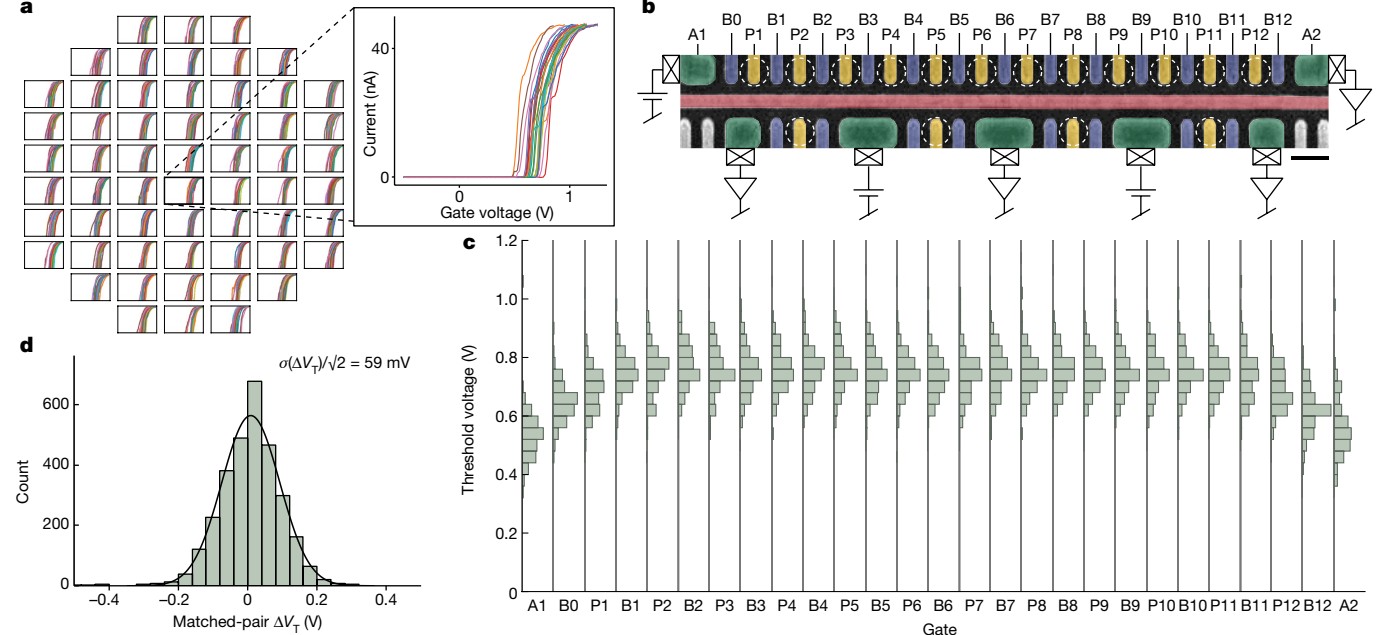

**Fig. 3 | Threshold voltage statistics from 12QD arrays. a**, Tiled array of $I$–$V$ curves taken on 12QD devices across a wafer. $I$–$V$ curves from a single device are shown in the inset, including 27 gates from the linear quantum dot array. **b**, Schematic of the measurement configuration overlaid on an inline scanning electron microscopy image of a representative 12QD device. Quantum dot locations are indicated by dashed circles. Gates are false-coloured by function: yellow for plunger gates, blue for barrier gates, green for reservoir gates and red for centre screening gate. Scale bar, 100 nm. **c**, Histograms of gate $V_T$ values across the 12QD array. Data are taken from 232 12QD devices across a wafer. **d**, Histogram of $\Delta V_T$ calculated between matched gate pairs using the $V_T$ dataset shown in **c**.

The measurements presented so far are all taken in the transport regime, in which devices are operated as 1D transistors or many-electron quantum dots. Spin qubit operations typically require tuning the electron occupancy to one electron per quantum dot. Accessing this single-electron regime can be challenging even for devices that perform well in transport, as reducing the charge number increases sensitivity to atomistic disorder. To characterize the single-electron regime of these devices, we perform automated charge-sensing measurements with each of the 12 quantum dots in the linear array. A typical measurement is shown in Fig. 4a. In this 2D sweep, the horizontal axis is plunger voltage and the vertical axis is the voltage of both barrier gates[41] (see Methods for more details). Charge-sensing scans are taken for all 12 quantum dot sites in the linear array, across 58 die on the wafer, for a total of 696 quantum dot sites. Over the 696 scans taken on a wafer with a 50-nm SiGe barrier, we find a 91% success rate in observing clear transitions (as gauged by eye relative to the noise background). This success rate represents highly consistent device performance and is primarily limited by the measurement algorithm (see Methods).

For further analysis on the 91% of successful scans on this wafer, we apply a numerical algorithm to detect transition curves in the 2D data and extract the coordinates for the first electron (1$e$) transition[36]

### Table 1 | Summary of device component yield across a representative 300-mm wafer

| Component | Yield (%) | Good count | Total count |
| --- | --- | --- | --- |
| Ohmics | 100 | 1,624 | 1,624 |
| Gates | 100 | 10,208 | 10,208 |
| Quantum dots | 99.8 | 3,703 | 3,712 |
| 12QD arrays | 96 | 223 | 232 |

Total count indicates the total number of each component tested. Good count indicates the number of each component found working. Yield is the percentage of the good count out of the total count for each component.

(see Methods). We define the '1$e$ voltage' as the plunger voltage position of the 1$e$ transition at the midpoint of the barrier voltage axis, indicated by the red star in Fig. 4a. We use the distance between the transition voltage and the left edge of the scan window to gain high confidence that these transitions represent the first electron in the quantum dot (see Methods).

A summary of plunger and barrier voltages at the 1$e$ transition is shown in Fig. 4b. These data represent the voltages needed to set the 1$e$ charge state in individual sites of 12QD arrays, sampled across a 300-mm wafer. They can therefore reveal how process variation translates to variation in the spin qubit operating point. Improving variation in spin qubit operating voltage has several benefits. Lower 1$e$ voltage variation makes for easier automation, as operating voltages are more predictable. Lower variation can also enable pathways for alleviating the interconnect bottleneck, as in proposals based on floating memory[42] or on voltage sharing among spin qubit lines[43,44]. For the former, lower variation can reduce the amount of classical circuitry needed to operate an array, whereas for the latter, lower variation will allow larger numbers of qubits to be accurately controlled with shared voltages.

To analyse the variation in 1$e$ transition voltage data, we repeat the same matched-pair voltage difference analysis as above, taking differences between 1$e$ voltages for mirrored pairs of plunger gates. Because this method highlights the variation within individual devices, it is well suited to benchmarking the potential of devices for voltage sharing, in which gate-to-gate variation, as opposed to die-to-die variation, is most relevant. The resulting distributions of voltage differences are shown in Fig. 4c,d for two wafers. The random variation in 1$e$ voltage extracted from wafers with 30-nm and 50-nm SiGe barriers are 61 mV and 63 mV, respectively. Both of these values are close to the random variation in gate $V_T$ (59 mV), suggesting that the random variation of a transistor-like metric (gate $V_T$) is matched by the random variation of a quantum metric (1$e$ voltage). We also observe strong correlation (correlation coefficient $\rho > 0.9$) between $V_{1e}$ and $V_T$ datasets (see Extended Data Fig. 6a). Altogether, this implies that these devices are not subject

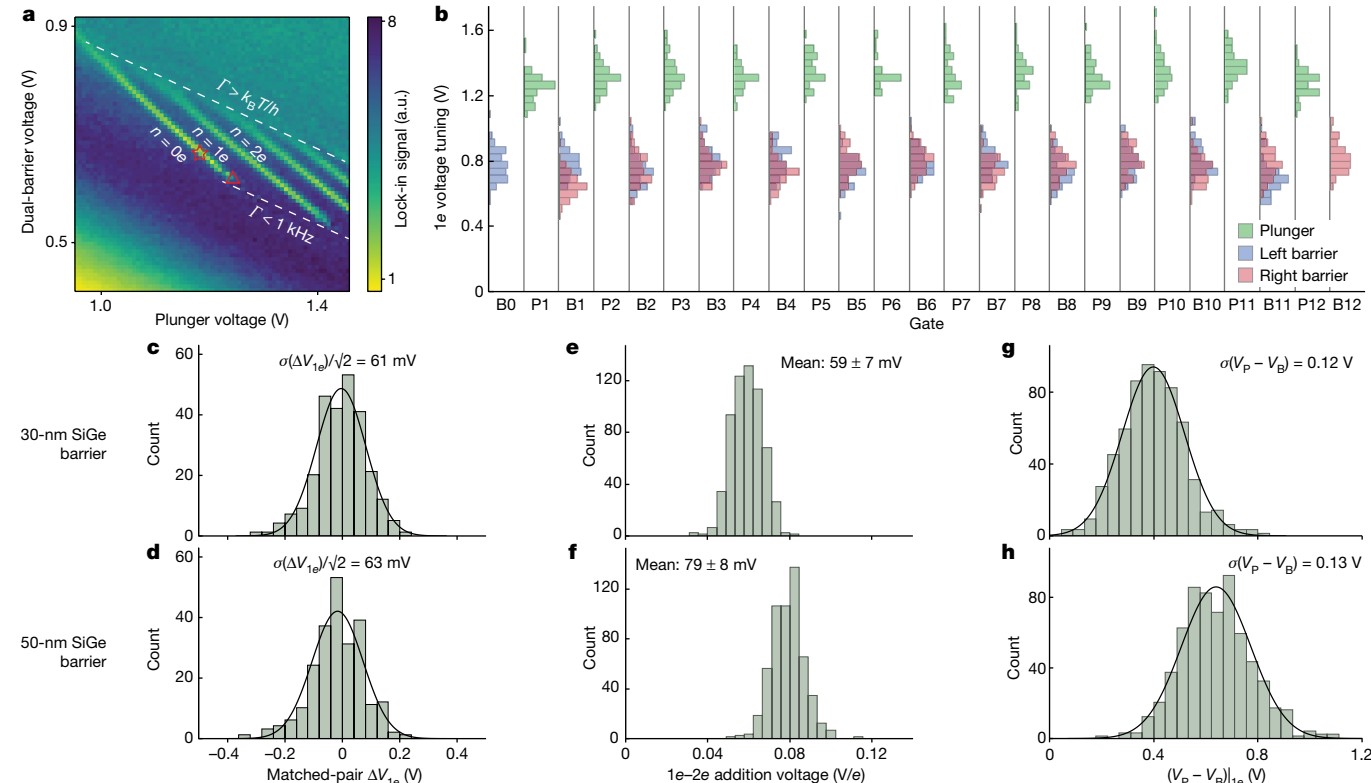

**Fig. 4 | Single-electron voltage statistics from 12QD arrays. a**, Example charge-sensing measurement. Bright lines represent electron number transitions in the quantum dot. The red star marks where the single-electron (1*e*) voltages are extracted for the plunger and barrier gates. The red triangle indicates where the voltage difference between the plunger and barrier gates is extracted. Dashed lines bound regions of extreme tunnel rate ($\Gamma$) (see Methods). **b**, Histograms of 1*e* plunger and barrier voltages across the 12QD array. Data are taken from a wafer with a 50-nm SiGe barrier. **c,d**, Histograms of 1*e* plunger voltage differences calculated between matched gate pairs on a wafer with 30-nm (**c**) and 50-nm (**d**) SiGe barriers. **e,f**, Histograms of 1*e*–2*e* addition voltage taken on a wafer with 30-nm (**e**) and 50-nm (**f**) SiGe barriers. The uncertainties shown are standard deviations. **g,h**, Histograms of voltage difference between plunger and barrier gates at the 1*e* transition taken on a wafer with 30-nm (**g**) and 50-nm (**h**) barriers. a.u., arbitrary units.

to substantially increased disorder at the single-electron regime compared with the many-electron regime. Also, although the 1*e* voltage variation is nearly the same between the two wafers, this variation can be better compared through the ratio of 1*e* voltage variation and 1*e*–2*e* addition voltage (Fig. 4e–f). This ratio effectively converts voltage variation to units of electron number and can be a useful benchmark for voltage-sharing applications[32]. These ratios are $(1.0 \pm 0.1)e$ and $(0.80 \pm 0.08)e$ for the 30-nm and 50-nm barrier wafers, respectively. The observation that the wafer with a deeper quantum well has a reduced ratio of this kind suggests that the 1*e* voltage variation is dominated by sources in the gate stack above the heterostructure. These sources could include charge defects (for example, interface traps or fixed charge in the oxide), gate-line edge roughness, gate work function variation, oxide thickness variation or some combination. These possible sources of variation all have analogies in the transistor field and could be improved by borrowing similar strategies; for example, the impact of oxide charge defects could be reduced by decreasing the oxide thickness between the heterostructure and the gate[29]. Measurements of carrier mobility on wafers with 30-nm and 50-nm SiGe barriers also show that samples with shallower quantum wells are subject to increased remote charge scattering (see Methods and Extended Data Fig. 3), suggesting that gains can be made by further reducing fixed charge in our high-$\kappa$ stack.

The charge-sensing data can also be used to benchmark the compatibility of these devices with voltage-sharing protocols[43,44]. One basic requirement for such schemes could be that all quantum dots in an array be tuned to the same electron number using the same voltage. From the 1*e* and 2*e* voltages obtained here, we estimate that a median of 63% of quantum dots per 12QD device could be set to $n = 1e$ with a common voltage (see Methods for more detail and Extended Data Fig. 5). Although this result is still far from the level of uniformity needed to tune an ensemble of spin qubits to their operating point with shared voltages, the 1*e* voltage variation results in Fig. 4 highlight the device metrics that must be further improved for voltage-sharing protocols to be feasible in large spin qubit processors.

To further assess variation at the single-electron regime, we calculate the standard deviation of the difference between plunger and barrier voltages at the cutoff point of the 1*e* transition line[24]. Using the datasets (see Fig. 4g,h) from the wafer with a 30-nm (50-nm) SiGe barrier, we calculate a standard deviation of 0.12 V (0.13 V), in agreement with the values reported in ref. 24 for six-dot devices with high exchange qubit fidelity[9]. This further confirms that the devices studied here can achieve low levels of disorder at the single-electron regime while being fabricated with a high-yield 300-mm process.

We also find that devices from these wafers perform well when operated as spin qubits (see Extended Data Fig. 1). Across many devices and wafers, we measure, on average, coherence times of $T_2^* = 0.6\ \mu s\ (5\ \mu s)$ and $T_2^{Echo} = 98\ \mu s\ (205\ \mu s)$ for $^{Nat}Si\ (^{28}Si)$ quantum wells, limited by (residual) nuclear spins. In a $^{28}Si$ device, we also demonstrate high single-qubit Clifford fidelities of about 99.9%, on par with leading results across the field. Furthermore, we find that the high electrostatic reliability demonstrated here allows us to efficiently gather data on many qubits towards studies of variability. The high device yield combined with cryo-prober testing enables a straightforward path from device fabrication to the study of spin qubits, eliminating failures owing to yield or electrostatics at the dilution refrigerator stage. Thanks to a

low-disorder host material (Si/SiGe), an all CMOS-industry-compatible fabrication process with low process variation and a high-volume cryogenic testing method, we achieve a large and extensible unit cell of up to 12 qubits. Although future work at mK temperatures will involve expanding operation of this unit cell, high-volume testing with the cryo-prober will continue to enable process optimization to reduce variation and disorder, as well as more advanced performance screening (such as charge noise, interdot coupling and 1*e* transition disorder) to identify the leading-edge test chips for quantum computing applications. Altogether, these results set a new standard for the scale and reliability of spin qubit devices today and pave the way for much larger and more complex spin qubit arrays of the future.

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

## Methods

### Electron temperature measurement

Electron temperature in the cryo-prober is measured from a charge-stability diagram, using a transition line that is tuned to avoid tunnel rate broadening. This stability diagram is shown in Extended Data Fig. 2a. A 1D measurement of the transition line is then taken to extract the width of the transition line. The lock-in data are integrated with respect to swept voltage and subtracted by a linear background. The resulting data are then fit to the model for a temperature-broadened charge-sensor transition[45] to extract an electron temperature of $1.6 \pm 0.2$ K. The processed data and theoretical fit are shown in Extended Data Fig. 2b. The uncertainty is estimated from the uncertainty of the lever arm ($0.08 \pm 0.01$), which is measured from bias triangles. We attribute the relatively large offset between the electron temperature and the base temperature of the stage to two possible limiting factors: a lack of filtering on the DC wiring and thermal resistance between the wafer and the chuck. Improvements to the electron temperature could be made by adding low-pass filters to the wiring to provide better thermalization of the probes and/or by decreasing the thermal resistance between the wafer and the chuck.

### Test structure investigation

The mask set used in this work produces many different device types on each wafer, including fully integrated spin qubit arrays and test structures. These test structures are designed to emulate sub-components of the complete devices and aid in both troubleshooting and targeting specific processes within the fabrication flow. All structures have the same pad design ($100 \times 100$ $\mu m^2$ in size with 150-$\mu m$ pitch) to match the probe pin array (see Extended Data Fig. 3a,b), allowing many different structures to be measured in situ. Switching among device types simply requires changes in software or minor changes at the electronics rack. The performance of all these structures is improved through process optimization, guided by feedback from the cryo-prober. The following sections focus on two such test structures: gate-line resistance structures and Hall bars.

**Gate-line resistance measurements.** The DC gate-line resistance, including both gate and interconnect layer, is an important factor in RF (approximately 0.1–20 GHz) signal delivery during qubit control. Improvements in gate-line resistance across several wafers are shown in Extended Data Fig. 3c. Here gate-line resistance is reduced through optimization of the gate fabrication process with normal-conducting materials and through the introduction of superconducting materials to the stack. Validating the superconducting process in particular is made possible by the 1.6-K base temperature of the cryo-prober. For the fully normal-conducting wafers, improvements come from increasing the cross-sectional area at the smallest bottleneck of the gate line. Of the partially superconducting wafers, the first wafer (with median resistance of around 78 $\Omega$) includes superconducting materials in the gate layer but still has a normal-conducting interconnect layer. The second wafer (with a median resistance of about 6 $\Omega$) includes superconducting materials in both gate and interconnect layers. We note that these measurements are taken using two-point resistance test structures and include a wiring resistance of about 30 $\Omega$, which is subtracted from the plotted data. The small remaining resistance in the metal stack with both layers superconducting could be because of the uncertainty in the wiring and probing resistance or to the via between gate and interconnect layers remaining normal-conducting (see Supplementary Information for more characterization of the superconducting layers).

**Carrier mobility measurements.** Carrier mobility is an important metric for spin qubits. In the case of Si/SiGe devices, electron mobility is a direct measure of the quality of the Si quantum well in which qubits are defined and provides a target for optimizing the heterostructure growth recipe. Although a magnetic field is needed to measure mobility most accurately, we can use cryo-prober measurements to generate a reasonable estimate to compare the quantum well quality of different wafers.

Carrier mobility is estimated from measurements of channel resistance in four-probe Hall bar devices at zero magnetic field. A schematic of the measurement configuration is shown in Extended Data Fig. 3b. Each device has six ohmic contacts, enabling two separate channel resistance (and mobility) measurements per device. The mobility calculation depends on knowing the carrier density, so we approximate a fixed carrier density ($4 \times 10^{11}$ cm$^{-2}$) by measuring the device threshold voltage ($V_T$) and setting the gate voltage to $V_T + \Delta V$, in which $\Delta V = e\Delta n/c_g$, $e$ is the electron charge, $\Delta n$ is the approximated carrier density and $c_g$ is the estimated gate capacitance per area based on transmission electron microscopy imaging of the gate stack. With this method, further uncertainty comes from the unknown threshold density ($n_t$) at which the device first shows a threshold current, so approximating $n = \Delta n$ will lead to a systematic overestimate of mobility by a factor of $(1 + n_t/\Delta n)$. From measurements in a conventional cryostat with magnetic field control, we estimate a typical threshold density to be $n_t \approx 1.5 \times 10^{11}$ cm$^{-2}$, suggesting that actual mobilities are about 30% less than the estimates generated in this way.

Using this estimation method, we observe improvements in mobility distributions across several wafers, as shown in Extended Data Fig. 3d. All wafers shown have the same quantum well thickness of about 5 nm. We attribute the mobility improvements to two changes: increasing the SiGe barrier thickness (from 30 nm to 50 nm), thereby reducing remote scattering from charge centres in the gate stack, and improving the quantum well growth recipe itself ('QW A' to 'QW B') to reduce background oxygen concentration. For the highest-mobility process, we also observe a similar mobility distribution before and after isotopic purification of the quantum well to $^{28}$Si, confirming that epitaxial quality is maintained with the purified growth precursor.

To further understand these observations, we select two samples with the QW B process (one with a 30-nm SiGe barrier and one with a 50-nm SiGe barrier) and perform measurements in a conventional cryostat with magnetic field control (Quantum Design PPMS DynaCool) at a temperature of 1.7 K. These measurements are shown in Extended Data Fig. 3e. Here we confirm the observation from the cryo-prober measurements that samples with the deeper quantum well have higher mobility. We also find that the absolute values of mobility are about 30% less than the estimates from the cryo-prober, confirming the expected systematic offset. The two samples also show a difference in the dependence of mobility on carrier density in the high-density regime (approximately $5 \times 10^{11}$ cm$^{-2}$). These different trends suggest different mobility-limiting mechanisms: remote scattering in the case of the 30-nm SiGe barrier and scattering in or near the quantum well in the case of the 50-nm SiGe barrier[46]. Overall, these measurements confirm that estimated mobility distributions obtained with the cryo-prober are useful for detecting substantial changes in carrier mobility resulting from heterostructure changes.

We note that, in these datasets, all wafers contain a fraction of devices (10–30%) with greatly reduced mobility, as can be seen in Extended Data Fig. 3d. This statistical phenomenon is confirmed with conventional Hall measurements and is not an artefact of the measurement method. By measuring mobility on both halves of devices, we also observe that this mobility degradation can be limited to a single half of the device, suggesting that it arises from a discrete defect mode, such as pile-ups of misfit dislocations[47]. By overlaying the Hall bar outline on a map of such defect pile-ups, we estimate that roughly 25% of Hall bars could be bisected by such defects, roughly matching the observed frequency of mobility degradation. We expect that the bimodal distribution of mobility is also related to the size of the Hall bars (6 $\mu m$ in width), which could allow a single defect pile-up to have an outsized effect on the mobility

extracted from a single device half. The comparatively high yield of 12QD arrays on these wafers could be explained by the much smaller size of those arrays (at least two orders of magnitude), making them much less likely to overlap these defects. By the same reasoning, we expect that larger Hall bars fabricated on the same wafers would overlap more of these defects, averaging out the impact of individual defects and possibly resulting in a more unimodal distribution of mobility.

## Automated device measurements

After a device is contacted with the probes, each current channel in the device (including the qubit channel and the four charge-sensor channels) is turned on with all gates over that channel at the same voltage. Once the $V_T$ of each channel is recorded, the gates of each channel are set to a fixed voltage relative to the channel $V_T$. The qubit channel is then isolated from the sensor channels by reducing the centre screening gate voltage until the cross-conductance between channels drops to zero (within the noise floor). The voltage of individual gates is then fine-tuned to set a roughly uniform carrier density across the channel. This is done through an iterative process in which the transconductance of each gate is sampled and the voltage on that gate is increased (decreased) if the transconductance is above (below) a threshold value. These transconductance thresholds are calculated relative to the absolute value of device current ($I_0$) and are set at $0.5I_0$ A V$^{-1}$ and $2.0I_0$ A V$^{-1}$ for the low and high thresholds, respectively. This effectively sets the voltages of all gates so that they are at roughly the same point on their pinch-off curves relative to their $V_T$. The $V_T$ data for all gates are extracted from pinch-off curves taken with a source–drain bias of 1 mV. $V_T$ is identified using the constant-current method[48] with a constant current of 1 nA. This current is chosen to be well above the offset of the current preamplifiers (<100 pA). The sweep range for the pinch-off curves is set from well below zero (−0.5 V) to the accumulated voltage of the gate after fine-tuning, ensuring that the scan range includes the pinch-off point despite variation in $V_T$ from gate to gate.

The voltages needed to tune up a quantum dot at each site are identified by setting each plunger gate to a fixed voltage relative to its $V_T$ and varying the barrier gate voltages about their individual $V_T$ values in a 2D sweep (a barrier–barrier scan). A phenomenological 2D function is fitted to these data to extract the corner point, which—combined with the plunger voltage—is used to define the 'tune-up' parameters for the quantum dot site. Defining the barrier sweep range based on the $V_T$ values of the gates ensures that the scan window is positioned to include this tune-up point despite variation in its location from gate to gate.

The charge-sensing measurements shown in Fig. 4 are taken with one quantum dot tuned up at a time on the qubit side. The closest charge sensor to that quantum dot is also tuned up, and neighbouring charge sensor dots are pinched off with their respective plunger gates. Changes in electron number are detected using a lock-in technique. A modulation voltage of 3 mV (root mean square) at a frequency of approximately 1 kHz is applied to the screening gate on the qubit side and the current through the charge sensor is read out with a lock-in amplifier at a sample time of 10 ms. To generate the charge-sensing measurement, the plunger voltage is swept at a fixed range relative to its $V_T$, and the two barrier gate voltages are stepped simultaneously. The barrier gates are stepped over the same voltage interval but with different voltage values. The step values of each barrier gate are defined relative to the individual 'tune-up' voltage of that gate extracted from the barrier–barrier scan. In the example shown in Fig. 4a, the barrier voltage range shown on the vertical axis is the voltage of the left barrier gate. The sweep range is chosen to take each quantum dot from zero-electron to several-electron occupation along the plunger axis and from low tunnel rate ($\Gamma \ll 1$ kHz) to high tunnel rate ($\Gamma \gg 1$ GHz) along the barrier axis. Transition lines disappear at the bottom of the scan window, at which the tunnel rate falls below two times the lock-in frequency (roughly 1 kHz), and at the top of the scan window, at which the lines become broadened by tunnel coupling energy. For wafer-level

maps of the charge-sensing measurements used to collect the 1$e$ voltage data summarized in Fig. 4, see Extended Data Figs. 8 and 9.

Automated charge-sensing measurements can also be taken on double quantum dots. The three barrier gates that define each double quantum dot are first set to a fixed voltage relative to their individual $V_T$ values. The plunger gate voltages for each dot are then swept to generate a 2D charge-stability diagram. Although these scans are not analysed quantitatively in this work, a demonstration of this type of measurement can be seen in Extended Data Fig. 10.

We note that the overall device measurement rate is predominantly set by the speed of measurement hardware. Notable gains can therefore be made by implementing faster hardware (for example, arbitrary waveform generators) and higher-bandwidth amplification (for example, cryogenic amplifiers[49]) without any further changes to the tune-up procedure.

## Threshold voltage measurements

The $V_T$ data shown in Fig. 2b are collected using the procedure described in Methods on automated device measurements. The data summarized in Fig. 2b contain a combination of $V_T$ data from plunger and barrier gates on both the qubit and the charge sensor sides of devices. For the earlier versions of devices (first ten wafers shown), data are taken from a combination of three-quantum-dot and 12QD arrays. For the optimized version (last five wafers shown), all data are taken from 12QD arrays.

## Barrier–barrier scans

To qualitatively characterize quantum dot confinement in our devices, a measurement referred to as a barrier–barrier scan is used. This involves a 2D sweep of the barrier gate voltages that define each quantum dot while measuring the transport current through the quantum dot. Current oscillations in these scans indicate the formation of a quantum dot between the two barrier gates, as transport between source and drain becomes dominated by Coulomb blockade[50]. Figure 2c shows examples of these measurements from each of the three fabrication versions featured in Fig. 2b. The first two versions show substantial disorder and/or instability in these measurements. By comparison, the optimized process, incorporating reductions in fixed charge and the extra screening gate layer, leads to clean confinement with the barrier gates and stable current throughout the length of the scan. Extended Data Fig. 7 shows more examples of these scans taken across wafers with each of the three versions of fabrication.

## Yield analysis

The measurements of yield summarized in Table 1 are taken from a total of 232 12QD devices, spanning 58 die across the wafer and including four nominally identical devices per die. We exclude the outermost ring of die at the edge of the wafer as these are not targeted in all steps of fabrication. The component yield metrics are calculated using the following definitions. Ohmic contact yield is defined as the fraction of contacts through which current in the Si quantum well can be linearly controlled. Gate yield is defined as the fraction of gates that can be used to turn on and pinch off their respective current channel. Quantum dot yield is defined as the fraction of quantum dot sites at which a viable quantum dot tune-up point can be identified from barrier–barrier scans. Failure to identify this tune-up point is determined by the fitting procedure failing to converge and therefore not outputting any barrier voltage values. This occurs when the data fail to conform to the phenomenological model of a 'corner point', at which current is pinched off simultaneously by both gates. For the data used here to calculate yield, we also examine all instances of failed fits by eye to confirm that they are not the result of an error in the fitting procedure. Last, full device yield is defined as the fraction of devices for which all sub-components (all ohmic contacts, gates and quantum dots) yield.

Out of 3,712 quantum dot sites tested and summarized in Table 1, the nine that fail to tune up are also observed to have anomalously

low pinch-off voltage (<0.2 V) on at least one of the three gates defining that quantum dot. These nine sites are also confined to the charge sensor side, at which gate geometry is most complex. This indicates that this small number of non-yielding quantum dots is because of the processing of the 0.3% most marginal gates as opposed to, for example, quantum well defects. We attribute these edge cases on the charge sensor side to a known failure mode in the gate lithography process. We note that the paths to improving the robustness of this process to fix these extreme outlier cases are well understood.

## Matched-pair voltage difference analysis

When working with distributions of gate parameters such as threshold voltage ($V_T$) or 1-electron voltage ($V_{1e}$), there are a variety of possible methods for analysing variation. The simplest is the standard deviation of each gate voltage distribution. For the 25 plunger and barrier gate distributions shown in Fig. 3c, this standard deviation ranges from 63 to 89 mV. This standard deviation incorporates all causes of cross-wafer variation, including both random effects and systematic cross-wafer phenomena arising from processes such as deposition and etch. As a measure of individual device performance, we focus our attention on the random component of this variation, which leads to variation (of $V_T$ or $V_{1e}$) within the length scale of individual devices. To estimate this random component of variation, we adapt a standard CMOS industry method of analysing matched-pair voltage differences[22]. The standard approach for transistor devices is to take the difference between $V_T$ values ($\Delta V_T$) of neighbouring devices to compare gates that are as close together as possible. For quantum dot devices, which have a more complex, multigate structure, there are several ways that the matched-pair method can be adapted. Simply taking the difference between nearest-neighbour gate pairs within the array minimizes the distance between matched pairs but comes with the drawback of introducing systematic effects of gate geometry. Because different gates along the array are subject to different cross-capacitances from their surrounding environment, systematic differences in $V_T$ can be present within the array that can appear in the resulting matched-pair $\Delta V_T$ distributions. Such systematic effects are seen clearly in Fig. 3c, in which gates nearest to the edge of the array tend to have lower $V_T$ owing to their different capacitive environment. To factor out these effects of geometry, we choose to perform the matched-pair variation analysis using mirror-symmetric pairs rather than nearest-neighbour pairs. This ensures that both gates in every pair are subject to nominally the same capacitive environment, owing to the mirror symmetry of the array. Using this approach, we combine the raw $\Delta V_T$ data into one distribution and extract the standard deviation. This resulting metric, reduced by a factor of √2, represents random variation within the length scale of an individual device, excluding aforementioned systematic sources of die-to-die variation as well as the systematic voltage offsets owing to cross-capacitance changes at the edges of the array. The matched-pair distributions that result from this analysis are shown in Figs. 3d and 4c,d for $V_T$ and $V_{1e}$, respectively.

As a check of our approach using mirror-symmetric matched pairs, we take the $V_{1e}$ dataset from the wafer with a 50-nm SiGe barrier (shown in Fig. 4b) and generate matched-pair $\Delta V_{1e}$ distributions using both mirror-symmetric pairs and nearest-neighbour pairs. For each method, the median values of all gate pair distributions are plotted in Extended Data 6b,c. In general, median values near zero indicate that the method is capturing random variation, whereas median values farther from zero indicate that systematic sources of variation are also playing a role. The distributions generated from nearest-neighbour pairs include median values that are clearly larger than those generated from mirror-symmetric pairs: the largest absolute value median generated from nearest-neighbour (mirror-symmetric) pairs is 89 (48) mV. In the case of nearest-neighbour pairs, this is driven by systematic effects of gate position, as evidenced by the antisymmetric trend of

median value as a function of pair position visible in Extended Data Fig. 6c. When gate pair distributions are combined for both methods, we also find that the nearest-neighbour pair method gives rise to a larger matched-pair standard deviation compared with the method of mirror-symmetric pairs (68 mV compared with 63 mV), as shown in Extended Data Fig. 6d,e. This confirms that the matched-pair variation in the case of nearest-neighbour pairs is being inflated by systematic geometric effects and that the result from using mirror-symmetric pairs is closer to the intrinsic random variation we intend to capture. Altogether, these findings suggest that the use of mirror-symmetric pairs is superior to the use of nearest-neighbour pairs when extending the matched-pair variation analysis method to the case of multigate quantum dot arrays.

We note that this approach of using mirror-symmetric pairs may need to be revised as quantum dot arrays become much larger, as increased separation between gate pairs could lead to the systematic components of variation being incorporated into the analysis. In this case, plots of median $\Delta V_{1e}$ as a function of gate pair such as those shown in Extended Data Fig. 6b will serve as a useful check for whether or not systematic effects are starting to contribute. The 12QD arrays studied here are still of a size at which the mirror-symmetric method is valid, as confirmed by the median values found in Extended Data Fig. 6b, as well as the finding that all gate pairs within the array can be well approximated as having the same correlation coefficient (see Supplementary Information). Future larger arrays could be handled by limiting the mirror-symmetric pair method to apply only within repeating unit cells of the array, in which each unit cell is of a similar size to the 12QD arrays studied here (approximately 1 μm).

## Charge-sensing success rate

The charge-sensing success rate (91%) reported in the main text depends on several factors: the relevant sensor quantum dot must yield, the sensing signal must be high enough relative to background noise to resolve transitions and the charge sensor must remain stable throughout the length of the scan. We attribute the success rate to be mainly limited by factors related to the measurement algorithm: the automated tuning of the charge sensor and instances of charge sensor instability occurring during the scan. Even in cases in which both quantum dots (sensor dot and sensed dot) yield, the automated charge sensor tune-up procedure can lead to insufficient signal relative to background noise. Signal can also be degraded by drift of the charge sensor tuning over the timescale of the measurement (several minutes). We expect that the success rate can therefore be improved with a more sophisticated measurement algorithm, such as by adding further sweeps of charge sensor gate voltages to optimize sensitivity or by incorporating active feedback into the measurement loop to analyse data quality[51] and retake measurements after charge sensor shifts occur. We expect that the success rate can also be improved by reducing electron temperature, which will increase charge sensor sensitivity and will possibly improve charge offset stability[52] through deactivation of two-level fluctuators[53].

## Charge-sensing transition curve analysis

Transition line coordinates are extracted from charge-sensing measurements using the following procedure. The raw lock-in amplifier data are first filtered with a first-order Gaussian filter to remove slowly varying features. A maximum filter is then used to identify features of high signal in the pre-filtered data. An algorithm is then used to convert the set of 'maximum points' into a set of 'curve segments'. Curve segments are found by searching for groupings of maximum points that satisfy the following criteria: each point in the curve segment must be the closest maximum point to its nearest neighbour; the slope between each pair of neighbouring points must be within a target window; and the set of points must span a minimum specified 'length' in the vertical direction. Overlapping curve segments are

then merged into transition curves. Transition curves are then further filtered to remove outlier curves and ordered by their coordinate means. The first and second transition curves generated from this algorithm are identified with the 1-electron and 2-electron transitions, respectively. An example of the entire sequence is shown in Extended Data Fig. 4. The '1$e$ (2$e$) voltage' is defined as the plunger voltage at which the 1$e$ (2$e$) transition line crosses the midpoint of the barrier voltage axis. This point corresponds to both barrier gates being tuned to their respective 'tune-up' points extracted from the barrier–barrier scans. The 1$e$–2$e$ addition voltage is calculated as the difference between these voltages. We note that, in some cases (15%), the 1$e$ (2$e$) transition in the scan window does not cross the midpoint of the barrier voltage axis, in which case no 1$e$ (2$e$) transition voltage is extracted from that scan.

## Impact of tuning the barrier voltages on 1$e$ voltage variation

The analysis of variation in matched-pair 1$e$ voltage differences ($\Delta V_{1e}$) presented in the main text reports the variation in voltage of a single gate voltage (the plunger gate) per quantum dot, analogous to how the analysis is performed for transistors. Given that the gate layout of quantum dot arrays is more complex than a typical transistor, it is important to consider the effect of cross-capacitance from other gates on the extracted $\Delta V_{1e}$ variation. In particular, the barrier gates that surround each quantum dot can have high cross-capacitance relative to the plunger gate (as can be seen in Fig. 4a).

We perform further analysis and measurements to quantify the impact of tuning the barrier voltages (as opposed to using fixed barrier voltages) on extracted metrics of 1$e$ voltage variation (see Supplementary Information for complete analysis). These results include two main conclusions. First, we find that tuning the barrier voltages can reduce the absolute standard deviation of 1$e$ voltage distributions ($\sigma(V_{1e})$) in the presence of device-level correlations between barrier and plunger voltage offsets. Second, we find that tuning the barrier voltages does not reduce the standard deviation of matched-pair $\Delta V_{1e}$ distribution ($\sigma(\Delta V_{1e})$), the main variation metric in this work, owing to this metric factoring out the effects of device-level correlations. In fact, this metric of variation tends to increase when tuning the barrier voltages, owing to the coupling of uncorrelated voltage offsets on barrier gates to plunger voltage values through cross-capacitance. This increase is greater for devices with larger amounts of cross-capacitance. In cases of notable cross-capacitance, such as devices studied here with a 50-nm SiGe barrier (about 55% between nearest neighbours), this increase can be about 20%. We note that, between the wafers studied in Fig. 4c,d, cross-capacitance is greater for the wafer with a 50-nm SiGe barrier than it is for the wafer with a 30-nm SiGe barrier, meaning that this effect of increasing $\sigma(\Delta V_{1e})$ through barrier tuning is also stronger for the former wafer. This effect therefore does not change the conclusion presented in the main text that the impact of voltage variation is reduced in the wafer with the deeper quantum well.

In general, although fixing barrier voltages could make for more precise comparison between $\Delta V_{1e}$ distributions from wafers with different amounts of cross-capacitance, there are also benefits to tuning the barriers before measurement. Using fine-tuned barrier voltages results in a higher success rate in identifying the 1$e$ transition in the charge-sensing scan window. In our tests, approximately 20% fewer matched pairs are obtained for analysis in a 'barriers fixed' dataset compared with a 'barriers fine-tuned' dataset (see Supplementary Information). Tuning the barrier voltages is therefore a benefit for collecting large and representative datasets through automated measurements. Also, if barrier voltage variation is high, there is some risk of sample bias when using fixed barriers, as quantum dots with the highest barrier voltage offsets may result in 1$e$ transitions being missed in the automated measurements and therefore not counted. For these reasons, we have maintained using tuned barrier voltages as our standard method for collecting $V_{1e}$ statistics.

## 1$e$ transition validation

To validate that the 1$e$ voltages we report are actually the first electron in the quantum dot, we extract the margin between the 1$e$ transition voltage and the left edge of the scan window and compare it with the distribution of addition voltages between the 1$e$ and 2$e$ transitions. To have high confidence that the first transition represents the first electron, we require this 'scan margin' be greater than two times the typical addition voltage. For the 50-nm SiGe barrier wafer characterized in Fig. 4b, 98% of 1$e$ voltage data points have a scan margin value above this threshold, giving us high confidence that the 1$e$ transition data summarized in Fig. 4b is actually single-electron data. See Extended Data Fig. 4f,g for histograms of the 1$e$–2$e$ addition voltage and 1$e$ scan margin data from this wafer.

## Voltage-sharing analysis

To estimate the proportion of quantum dots in each 12QD device that could be set to single-electron occupation with shared voltages, we analyse the 1$e$ and 2$e$ voltage data from the 50-nm SiGe barrier wafer and search for a common voltage that best divides the 1$e$ and 2$e$ voltage distributions for each 12QD device. In this scheme, any 1$e$ voltage value above the common voltage ($V_{common}$) corresponds to $n = 0e$ and any 2$e$ voltage value below $V_{common}$ corresponds to $n \geq 2e$. The remaining instances correspond to quantum dots tuned to $n = 1e$. For each device, the optimal $V_{common}$ is found by minimizing the number of instances in which $n = 0e$ or $n \geq 2e$. Extended Data Fig. 5 shows a histogram of 1$e$ and 2$e$ voltage data points shifted relative to their assigned device-specific $V_{common}$ value. A scatter plot also shows the proportion of quantum dots in each category of electron number for all 12QD devices. The median success rate for tuning dots to $n = 1e$ is 63%.

We note that the data used in this analysis come from measurements of quantum dots tuned one at a time and that this method does not take into account the individualized set points of other gates in the array during measurements. We do not expect that tuning the barrier voltages results in an overestimate of the percentage of quantum dots tunable to 1$e$, because we observe that the variation of matched-pair 1$e$ voltage differences increases rather than decreases when the barrier voltages are tuned, owing to the factoring out of device-level correlation effects (see Supplementary Information). Similarly, this method of estimating the success rate of voltage sharing is also a measure of the variation within a device, in this case done by comparing individual 1$e$ and 2$e$ voltages to a common device-level voltage. Therefore, this method can be expected to factor out the impact of device-level correlations and, for the same reason as the matched-pair case, tuning the barrier gates will—if anything—slightly increase the 1$e$ variation observed for the plunger gates. Overall, we find that it is beneficial to perform the analysis after fine-tuning the barriers because that process can increase the proportion of 1$e$ data successfully obtained from a set of devices and therefore give a more representative sample of 1$e$ voltages for analysis.

Furthermore, we note that this success rate, or the fraction of quantum dots in an array that can be tuned to $n = 1e$ using a common voltage, can depend on both the size of the array and the method for choosing $V_{common}$. The dependence on array size can be considered to have two limits. In the limit of an array with a number of quantum dots $N = 1$, a success rate of 100% is guaranteed. In the 'large array limit', in which $V_{1e}$ and $V_{2e}$ data from each device can be well approximated by a normal distribution, the fraction of quantum dots in an array that can be tuned to $n = 1e$ using a common voltage can be estimated by assuming that each 'failure' results from each instance of a $V_{1e}$ ($V_{2e}$) value being above (below) the mean by more than half the addition voltage. The success rate can then be described by:

$$1 - 2\Phi\left(\frac{-V_{add}}{\sqrt{2}\,\sigma(\Delta V_{1e})}\right), \tag{1}$$

in which $\Phi$ is the cumulative distribution function of the standard normal distribution, $V_{add}$ is the addition voltage and $\sigma(\Delta V_{1e})$ is the standard deviation of matched-pair $V_{1e}$ differences. In the range of 'intermediate' array size, the success rate will decrease from 100% to this limiting value, but the rate of its decrease will depend on the particular method of choosing the value of $V_{common}$. To better understand this intermediate range, we simulate the success rate as a function of array size for two different methods of choosing $V_{common}$. The first method is that described above and shown in Extended Data Fig. 5a,b, in which $V_{common}$ is optimized to give the maximum number of $n = 1e$ successes. The second method naively sets $V_{common}$ to the mean of the combined $V_{1e}$ and $V_{2e}$ data for each device. We first simulate devices that reflect the experimental results from the wafer with a 50-nm SiGe barrier; we generate $V_{1e}$ and $V_{2e}$ data from a random normal distribution with a standard deviation equal to the measured $\sigma(\Delta V_{1e})/\sqrt{2}$ and use the average measured $V_{add}$ from that wafer. Extended Data Fig. 5c shows the results of simulated success rate as a function of array size, taking an average over 10,000 simulated devices at each array size. We find that the success rate of both methods decreases as a function of array size, saturating at the expected fraction based on a normal distribution. We also find that using the method in which $V_{common}$ is optimized can boost the success rate over a much larger range in array sizes compared with the simpler method based on the mean, only saturating at the large array limit around $N \approx 1,000$. We interpret this difference as an effect of sampling noise, in which—for intermediate array sizes ($N < 1,000$)—the distribution of $V_{1e}$ data departs from the ideal normal distribution, so optimizing $V_{common}$ for the sampled distribution of each device can outperform the method of simply setting $V_{common}$ from the mean. We also note good agreement between these simulated results and the results of both methods being applied to the measured data (marked as stars in Extended Data Fig. 5c).

Although these findings show that our reported success rate (63%) will tend to decrease as a function of array size, they also reveal how intermediate gains can be made by choosing an optimal $V_{common}$ value for each array. As arrays become much larger ($N > 1,000$), one way to preserve this benefit would be to assign different $V_{common}$ values to different unit cells of the array, in which each unit cell could contain $N < 1,000$ quantum dots. This approach would also mitigate the challenge of voltage variation across an array increasing as the array size increases. We also note that substantial gains can be made even in the large array limit through improvements in $V_{1e}$ variation. For example, decreasing $\sigma(\Delta V_{1e})$ by a factor of four while keeping $V_{add}$ fixed would lead to an expected success rate of around 99%, even in the large array limit (see Extended Data Fig. 5d).

## Qubit measurement setup

The qubit measurements are performed in Bluefors XLD dry dilution refrigerators with a base temperature of 10 mK. Each sample is mounted and wire bonded onto a custom printed circuit board (PCB) and placed on a cold finger that sits in the middle of the bore of a superconducting magnet. DC voltages from battery-powered voltage digital-to-analogue converters (QuTech SPI Rack) are applied to each gate electrode of the device. The signals are routed to the sample PCB using twisted-pair cables and pass through RC filters that are also thermalized on the cold finger. AC and MW signals are delivered to the sample PCB through coax cables with attenuators from room temperature to mK totalling between 21 and 28 dB. AC signals are applied to the plunger and barrier gates of the devices by adding them to the DC signals using RC bias tees ($R = 1$ MΩ, $C = 100$ nF) on the sample PCB. The microwave signal is added to the DC signal for the centre screening gate using an LC bias tee ($L = 1.7$ nH, $C = 1$ pF), also on the sample PCB. AC signals are generated using arbitrary waveform generators (Zurich Instruments HDAWG8 and custom DDS-based arbitrary waveform generators). MW signals are generated using I/Q modulation of either a Keysight E8267D or R&S SGS100A vector microwave source.

The charge sensor is measured using an AC coupled dual-stage SiGe heterojunction bipolar transistor (HBT) amplifier[54] on the sample PCB board. The design of the dual-stage amplifier is similar to other high-electron-mobility transistor-based amplifiers[55]. A stimulus voltage is applied to one of the ohmics of the charge sensor by means of a bias tee, generating an AC current through the charge sensor that gets amplified by the dual-stage amplifier. The small distance between the device and the base of the HBT in the first stage of the amplifier leads to a low parasitic capacitance, enabling bandwidths >1 MHz. The amplified current signal is demodulated at room temperature using the Zurich Instruments MFLI lock-in amplifier. In this setup, we achieve electron temperatures between 100 and 200 mK, dependent on the stimulus amplitude and bias applied to the emitter of the HBTs.

## Qubit readout and initialization

In the qubit measurements shown in Extended Data Fig. 1, two methods are used for readout and initialization. The first method is Elzerman readout[56], which involves spin-selective tunnelling of the qubit electron to a nearby reservoir. To perform this readout, the Fermi level of the reservoir is aligned between the spin-up and spin-down state, split by the Zeeman energy. If the electron is spin-up, the electron can tunnel out, followed by a spin-down electron tunnelling back in. This movement of charge can be detected in real time with the nearby charge sensor. If the electron is spin-down, it cannot tunnel out and therefore there is no change in the charge sensor signal. Because in either case the quantum dot ends with a spin-down electron, this readout can also be used to initialize the qubit.

The second method is Pauli spin blockade (PSB) parity readout of a pair of electron spins[15,57], which involves spin-selective tunnelling within a double quantum dot and does not need nearby reservoirs. This method uses the valley-orbit splitting, $E_{vo}$, between the singlet ground state and the triplet excited state that is found for certain electron numbers (that is, $2e$ or $4e$). Often, we observe that $E_{vo}$ in the $2e$ state is low with respect to the sample electron temperature, which degrades the readout fidelity. We expect this splitting to be limited by a combination of interface disorder, such as alloy disorder[30], and electron–electron interactions of the $2e$ state[58]. Consequently, we typically opt to define one qubit of the pair to contain three electrons, allowing us to use the much larger $E_{vo}$ typically found with the $4e$ state. The $3e$ state typically shows similar coherence times to the $1e$ state. We note that alternating the electron number between $1e$ and $3e$ across arrays in this manner could add overhead to scaling solutions based on voltage sharing.

To give an example of how parity readout is performed, consider a double dot in the (1,3) charge configuration in which the (0,4) state is used for readout. The plunger gates of the devices are pulsed to the PSB readout point in the (0,4), in which only the S(0,4) state is accessible and tunnelling to the T(0,4) state is not energetically possible. At this point, owing to the large Zeeman energy difference between the two dots[57], the $|\downarrow\uparrow\rangle$ and $|\uparrow\downarrow\rangle$ states quickly relax to the singlet, allowing tunnelling into the S(0,4) charge state. By contrast, $|\downarrow\downarrow\rangle$ and $|\uparrow\uparrow\rangle$ map onto the $T_+(1,3)$ and $T_-(1,3)$ states and tunnelling to the T(0,4) state is not allowed. Hence, the final charge state of the double dot determines the parity of the two electron spins and can be measured using the nearby charge sensor using integration times typically between 20 and 100 μs. For the single-qubit measurements in Extended Data Fig. 1, the state of the other qubit is fixed, allowing the full state of the measured qubit to be extracted.

To initialize the system, the S(0,4) state is prepared using postselection[15]. In particular, at the start of each sequence, PSB readout is used to determine whether the state is T(1,3) or S(0,4). If the state is T(1,3), then the measurement run is discarded. After preparing S(0,4) by means of postselection, the state is mapped to $|\downarrow\uparrow\rangle$ by applying an adiabatic ramp to the (1,3) regime in which $J \ll \Delta B_z$. Here we can perform single-qubit operations, followed by a second PSB readout to determine the final state.

## Micromagnet design and EDSR

Coherent manipulation of single-electron spins is performed using electric dipole spin resonance (EDSR) mediated by magnetic field gradients from cobalt micromagnets[59]. EDSR enables high-fidelity and local electrical control of spin qubits[60], and micromagnets can also be used to engineer the qubit frequencies along an array enabling addressability and high-fidelity two-qubit gates[6–8]. The micromagnets (Extended Data Fig. 1a) are patterned on top of the quantum dot samples using electron-beam lithography and standard lift-off techniques. The micromagnets are based on the design in ref. 15 and are magnetized in the direction indicated by the white arrow by ramping the external magnetic field to 3 T. The micromagnets are used to generate a magnetic field gradient, $dB_z/dy$, at each of the quantum dot sites, with simulations giving values ranging between 0.4 and 0.5 mT nm$^{-1}$. Microwaves are applied to the centre screening gate (highlighted in red), which displaces the electrons in the quantum dot in the $y$ direction, resulting in the electron effectively seeing an oscillating magnetic field in the $z$ direction that is perpendicular to the external magnetic field ($B_0$) in the $y$ direction. The micromagnets also generate a magnetic field gradient $dB_y/dx$ along the array ranging from 0.007 to 0.03 mT nm$^{-1}$ at full magnetization, with the gradient decreasing from Q1 to Q12. This field gradient is in the direction of the external field (that is, aligned to the quantization axis) and leads to different qubit frequencies along the array. Also, this field gradient can lead to decoherence, as charge noise can cause fluctuations in qubit position and hence the qubit frequency. The field gradient in the $dB_y/dy$ direction can also cause decoherence but is minimized close to zero by centring the qubit array between the two micromagnets. The coherent rotation of a single electron using EDSR as a function of MW burst time is shown in Extended Data Fig. 1b. In this measurement, we estimate that we apply a microwave power at the sample of about −35 dBm, taking into account the microwave source power of −2 dBm, attenuation of −21 dBm from attenuators in the cryostat and frequency-dependent cable losses of −14 dBm at the resonance frequency of 7.5 GHz.

## Randomized benchmarking

Randomized benchmarking is used to characterize the single-qubit gate fidelity in a $^{28}$Si sample. The experiment is performed by first applying a randomized sequence of a varying number ($m$) of Clifford gates to the qubit, followed by a final Clifford gate that is the inverse of the randomized sequence, then measuring the resulting spin-up probability[61,62]. For each data point, we perform 100 repetitions for 80 different randomized sets of gates for each sequence length. Also, we interleave Ramsey frequency calibrations for the qubit between every two randomizations (approximately every 1 min). Examples are shown in Extended Data Fig. 1c for two qubits, labelled Q1 and Q2, from a $^{28}$Si device labelled dev12. We perform the measurement with the qubit initialized in either the $|0\rangle$ or the $|1\rangle$ state and extract the difference in the measured spin-up probability for those two starting states, $P'_{|1\rangle} − P_{|1\rangle}$, as a function of sequence length $m$ (ref. 62). From an exponential fit of the data, $P'_{|1\rangle} − P_{|1\rangle} = ap^m$, we estimate average Clifford-gate fidelities $F_C = 1 − (1 − p)/2$ of 99.90 ± 0.01% and 99.88 ± 0.02% for Q1 and Q2, respectively.

## Coherence measurements

The dephasing time ($T_2^*$) of a qubit is measured using a Ramsey sequence, shown in Extended Data Fig. 1d for the qubit labelled Q3 from dev12. In this sequence, the wait time between two $X_{\pi/2}$ pulses is varied. An artificial oscillation is introduced to the data to improve the reliability of the fit by making the phase of the last $\pi$ pulse dependent on the evolution time. We fit the spin-up probability as a function of the free evolution time $\tau$ to extract $T_2^* = 15.6$ μs. In this fit, the decay exponent is kept as a free parameter.

As well as measuring the dephasing time, for most of the qubits we also measure the Hahn echo decay time $T_2^{Echo}$, in which a $X_\pi$ pulse is used to refocus low-frequency (quasi-static) noise, extending the qubit coherence time. Similar to the Ramsey sequence, we also introduce an artificial oscillation for fitting purposes. An example of this measurement for Q3 from dev12 is shown in Extended Data Fig. 1f. We fit the spin-up probability as a function of the free evolution time $\tau$ to extract $T_2^{Echo} = 225$ μs. In the fit, the decay exponent is again a free parameter.

Extended Data Fig. 1g shows coherence time measurements from 39 qubits formed in 14 devices (dev1–dev14) from five different wafers (w1–w5). Data are collected from a mix of two device types, either a linear array of three qubits (3Q) or a linear array of 12 qubits (12Q). $T_2^*$ is measured for each qubit using the Ramsey sequence (as shown in Extended Data Fig. 1d) and $T_2^{Echo}$ is measured using the Hahn echo sequence (as shown in Extended Data Fig. 1f). In dev3, the coherence times are measured for each qubit after tuning up the entire 12Q array. In dev11 and dev13, for some qubits, we plot several points measured for $T_2^{Echo}$ that varied greatly owing to device tuning. We observe that moving from $^{Nat}$Si (w1–w3) to $^{28}$Si (w4–w5) leads to about an order of magnitude improvement in $T_2^*$.

$T_2^*$ is determined by the integrated noise spectrum during the Ramsey experiment and therefore is dependent on the total measurement time[63]. In Extended Data Fig. 1g, the total measurement time for each of the $T_2^*$ data points varies between 1 and 10 min. Extended Data Fig. 1e shows the dependence of $T_2^*$ on the total measurement time for a subset of qubits measured in Extended Data Fig. 1g. The cumulative plots in Extended Data Fig. 1e are generated by performing many repetitions of the Ramsey experiment. From this dataset, we calculate the average $T_2^*$ for different measurement times. This is done by applying a moving average to the dataset with a window size that equals a particular measurement time. We then fit each averaged time trace to extract $T_2^*$ as a function of the window position and calculate the average $T_2^*$ from this. Here the $T_2^*$ decreases as a function of measurement time and saturates. Between 1 and 10 min, the $T_2^*$ can vary by a factor of about 2 and explains some of the variation in Extended Data Fig. 1g. The approximate $T_2^*$ saturation point, labelled $T_2^*(\infty)$, for each of the curves in Extended Data Fig. 1e are also plotted in Extended Data Fig. 1g and allows a better comparison between the different samples and with theoretical estimates of $T_2^*$.

In $^{Nat}$Si and $^{28}$Si samples, the average ratio between $T_2^{Echo}$ and $T_2^*(\infty)$ is about 150 and about 50, respectively. These numbers indicate that the exponent of the noise model, given by a power law $1/f^\alpha$, is $\alpha > 1$, consistent with nuclear spins dominating $T_2^*$. Also, the coherence times are not dependent on dot number/position in the devices (for example, Q1 versus Q12), despite the decoherence gradient decreasing by a factor of about four from Q1 to Q12. This suggests that, for most of the qubits, $T_2^*(\infty)$ and $T_2^{Echo}$ are predominantly limited by nuclear spins rather than charge noise. However, we note that—for some qubits—we sometimes find lower than expected values for $T_2^{Echo}$ that can be improved with device tuning. Although we have not fully investigated the cause of this, two potential reasons could be either that the dot position is offset with respect to the centre of the micromagnets (that is, in the $y$ direction), increasing substantially the decoherence gradient, or that—in some tuning configurations—charge traps are activated, leading to higher amounts of charge noise.

## Coherence modelling

To obtain an estimate of the dephasing time $T_2^*$ from nuclear spins, we consider the qubit electron to be confined in a crystalline lattice consisting of a 5-nm-thick strained Si quantum well (Si-QW) and a Si$_{0.7}$Ge$_{0.3}$ barrier on both sides of the well. The electron confinement is assumed to be given by (1) the harmonic oscillator potential with an orbital splitting $\Delta_{orb}$ for the in-plane direction ($x,y$) and (2) the potential barrier between the Si-QW and Si$_{0.7}$Ge$_{0.3}$ for the out-of-plane ($z$) direction. In this confinement potential, we estimate the electron wavefunction

$\psi(\mathbf{r}_i)$ at each nuclear-spin site $i$, in which the nuclear spins are distributed in the lattice with a probability given by their concentration. $\psi(\mathbf{r}_i)$ acts as a handle to the hyperfine interaction $A_{ik}$ between the electron and nuclear spins and the resultant $T_2^*(\infty)$, given by the equations[5]:

$$A_{ik} = \frac{2\mu_0}{3}\gamma_e \gamma_{nk}\eta_k |\psi(\mathbf{r}_i)|^2, \tag{2}$$

$$\left(\frac{1}{T_2^*(\infty)}\right)^2 = \frac{1}{2}\sum_{k={}^{29}\text{Si},{}^{73}\text{Ge}} \frac{I_k(I_k+1)}{3}\sum_i A_{ik}^2, \tag{3}$$

in which index $k$ denotes the spin-carrying nuclei of ${}^{29}$Si and ${}^{73}$Ge, with their total nuclear spins being $I_k = 1/2$ and $I_k = 9/2$, respectively, $\eta_k$ are their bunching factors and $\gamma_e$ and $\gamma_{nk}$ are the gyromagnetic ratio of the electron and nuclear spins, respectively.

For our calculations, we assume $\Delta_{orb}$ to be uncertain in the range of 1 meV and 2 meV, calculate $T_2^*(\infty)$ for 50 different distributions of nuclear spins for a given concentration and then estimate the bounds of the resultant $T_2^*(\infty)$ shown in Extended Data Fig. 1g (ref. 64). Hence this calculation accounts for both the uncertainty of the orbital splittings and the variation in location of nuclear spins in the lattice. We note from our simulations that ${}^{29}$Si and ${}^{73}$Ge nuclei in the Si quantum well and the Si$_{0.7}$Ge$_{0.3}$ barrier limit the $T_2^*(\infty)$ to be in the range 0.73–0.98 μs and 4.9–8.3 μs for both natural Si and isotopically enriched Si (800 ppm), respectively. The strength of the contribution from nuclear spins in the Si$_{0.7}$Ge$_{0.3}$ barrier can depend sensitively on the width of the Si/Si$_{0.7}$Ge$_{0.3}$ interface. Simulations based on a sigmoidal interface[33] and using a measured interface width of $4\tau = 1$ nm predict that residual ${}^{29}$Si nuclei in the quantum well are the main limiter to our coherence. The range of theoretical estimates of $T_2^*(\infty)$ for ${}^{Nat}$Si and ${}^{28}$Si with 800-ppm residual ${}^{29}$Si are shown in Extended Data Fig. 1g as shaded regions outlined by dashed and dashed-dot lines, respectively. The simulated ranges show reasonable agreement with the data, indicating that $T_2^*$ times are indeed limited by nuclear spins rather than charge noise.

## Data availability

The data that support the findings in this study are available in the Zenodo repository at https://doi.org/10.5281/zenodo.10601293 (ref. 65).

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

**Author contributions** S.N., O.K.Z. and T.F.W. designed the automated cryo-prober measurements. S.N., O.K.Z. and A.N. performed the cryo-prober measurements. F.L. contributed to the measurement software. H.C.G., E.H., A.J.W. and M.I. fabricated the devices. S.N. and O.K.Z. analysed the cryo-prober data. R.P., R.K. and S.P. contributed to the cryo-prober data analysis. O.K.Z., R.P. and K.M. enabled the cryo-prober installation. T.F.W., F.B., E.J.C., J.C., M.T.M., R.S. and G.Z. performed the qubit measurements. M.J.C., D.K., L.F.L., F.L., M.R., S.S. and J.Z. contributed to the preparation of the qubit experiments. F.A.M. performed the simulations of qubit coherence and micromagnets. N.C.B., S.B., J.R. and J.S.C. supervised the project. S.N. and O.K.Z. wrote the manuscript, with input from all authors.

**Competing interests** The authors declare no competing interests.

**Additional information**
**Correspondence and requests for materials** should be addressed to Samuel Neyens or James S. Clarke.

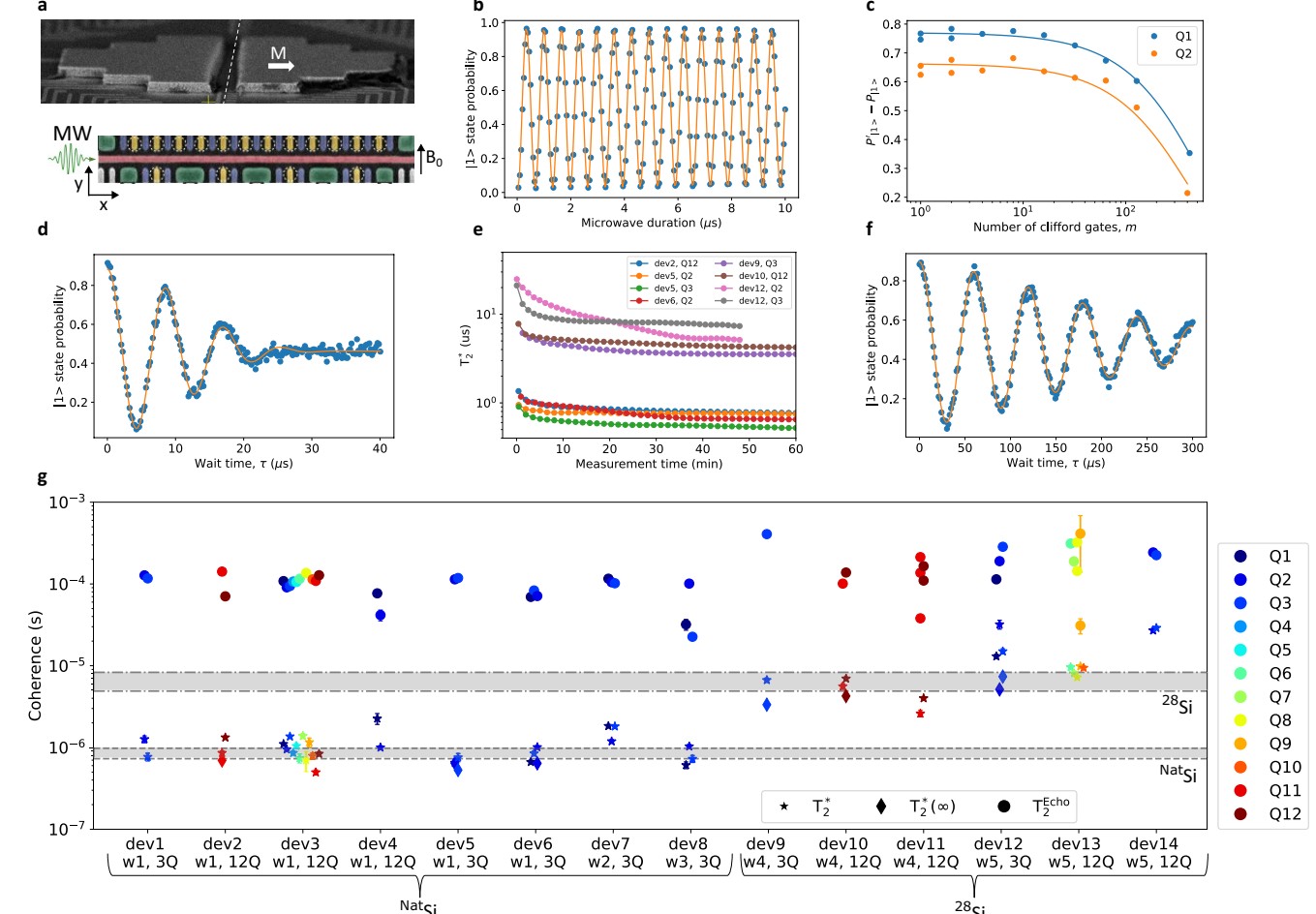

**Extended Data Fig. 1 | From single electrons to spin qubits. a**, Scanning electron microscopy image of the cobalt micromagnets fabricated on an Intel 12Q device to enable electric dipole spin resonance for single-qubit control. The white arrow indicates the direction of magnetization M. The dashed line shows where the linear array of quantum dots is formed with respect to the micromagnets. **b**, Rabi oscillations between the spin-up |1⟩ and the spin-down state |0⟩ driven by EDSR. **c**, Randomized benchmarking of single-qubit Clifford gates for two qubits, Q1 and Q2, from a $^{28}$Si device (dev12). The difference in the measured spin-up probability is plotted for two different starting states, |0⟩ or |1⟩, as a function of sequence length $m$. From exponential fits (solid lines) of the data, we estimate average Clifford-gate fidelities of 99.90 ± 0.01% and 99.88 ± 0.02% for Q1 and Q2, respectively. **d**, A Ramsey sequence performed on

Q3 from dev12. By fitting the decay (solid line), we extract $T_2^* = 15.6$ μs. **e**, Cumulative $T_2^*$ as a function of measurement time for a subset of devices described in **g**. The dephasing time saturates at long measurements to the limit $T_2^*(\infty)$. **f**, A Hahn echo sequence performed on Q3 from dev12. By fitting the decay (solid line), we extract $T_2^{\text{Echo}} = 225$ μs. **g**, $T_2^*$ (stars), $T_2^*(\infty)$ (diamonds) and $T_2^{\text{Echo}}$ (circles) data points measured from 39 qubits formed in 14 devices (dev1–dev14) from five different wafers (w1–w5). Two device types are featured: a linear array of three qubits (3Q) or 12 qubits (12Q). The colour of each point corresponds to the position of the qubit in the array, which is labelled Q1–Q3 for the 3Q samples and Q1–Q12 for the 12Q samples. Error bars represent uncertainty of the fit.

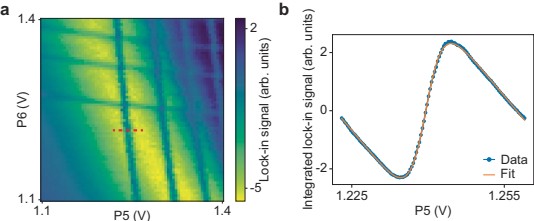

**Extended Data Fig. 2 | Electron temperature measurement in the cryo-prober. a**, Charge-stability diagram showing the configuration in which electron temperature is extracted. **b**, 1D measurement across the transition indicated by the dashed red line in **a** with theoretical fit overlaid.

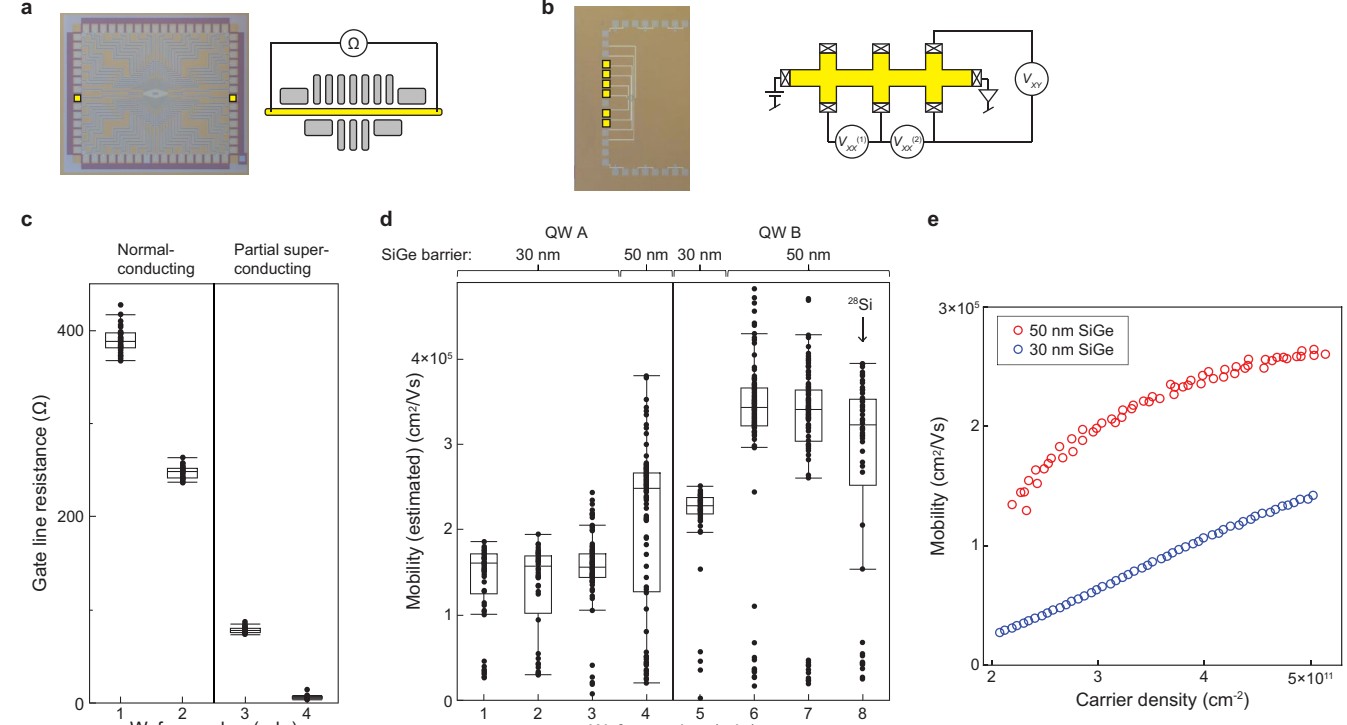

**Extended Data Fig. 3 | Process optimization aided by cryo-prober feedback. a,b**, Optical images of two test structures with the same pad layout: gate-line resistance test structure (**a**) and Hall bar (**b**). In each case, the active probe pads are highlighted and a schematic of the measurement is shown. **c,d**, Improvements in device metrics from process optimization. Box plots show the median and interquartile range of each distribution. Whiskers mark the maximum and minimum values excluding outliers, which are defined as points removed from the median by more than 1.5 times the interquartile range. **c**, Gate-line resistance is reduced through optimization of the gate process and introduction of superconducting materials. **d**, Estimated carrier mobility is increased through improvements in epitaxy and increase in quantum well depth. **e**, Hall measurements taken in a conventional cryostat show mobility as a function of carrier density for two samples with the QW B process and different SiGe barrier thickness.

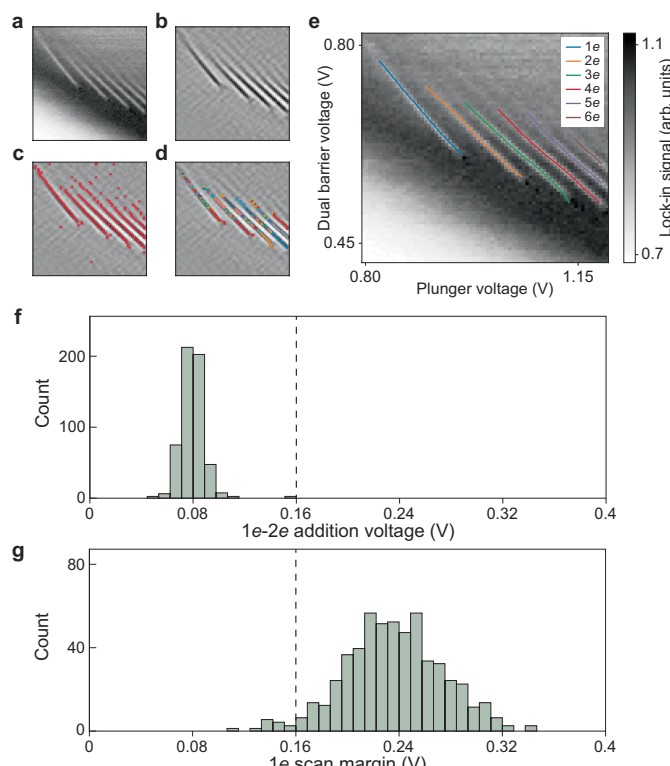

**f**

**g**

**Extended Data Fig. 4 | Charge-sensing scan analysis. a–e,** Sequence to extract transition curve data. **a,** Raw lock-in data are taken in by the analysis algorithm. **b,** A first-order Gaussian filter is applied to remove background charge sensor features from the data. **c,** A maximum filter is applied to locate points of high signal. **d,** Local maxima are filtered and binned into 'curve segments'. **e,** Curve segments are merged into a set of continuous transition curves. The coordinates of these transition curves are collected and used to analyse 1*e* and 2*e* transition voltage statistics. **f,g,** Validation of 1*e* transition data. **f,** Histogram of 1*e*–2*e* addition voltage statistics from a wafer with a 50-nm SiGe barrier. The vertical dashed line indicates two times the median addition voltage (0.158 V). **g,** Histogram of the 1*e* scan margin, or distance between the purported 1*e* transition and the left edge of the scan window, extracted from charge-sensing scans on the same wafer as **f**. 98% of scans have a margin more than twice the median addition voltage.

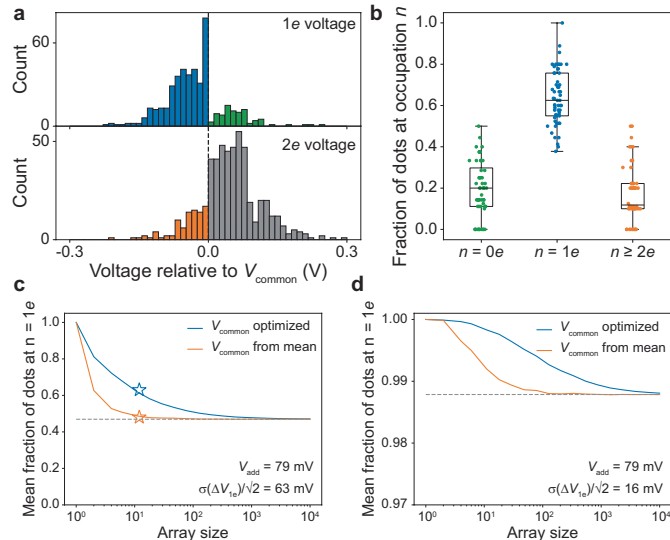

**Extended Data Fig. 5 | Voltage-sharing analysis. a**, Histogram of 1*e* and 2*e* electron voltages taken across a full wafer. Each voltage is plotted relative to the common voltage ($V_{common}$) assigned to the 12QD device from which the data point is taken. 1*e* voltages above $V_{common}$ (green) represent dots tuned to $n = 0e$. 2*e* voltages below $V_{common}$ (orange) represent dots tuned to $n \geq 2e$. All other data points (blue and grey) represent dots tuned to $n = 1e$. **b**, Scatter plot indicating the fraction of quantum dots tuned to various electron configurations at $V_{common}$ for 12QD devices across a wafer. The fraction of dots in $n = 1e$ represents the success rate, giving a median success rate of 63%. Box plots show the median and interquartile range of each distribution. Whiskers mark the maximum and minimum values excluding outliers, which are defined as points removed from the median by more than 1.5 times the interquartile range. **c,d**, Simulated success rate for tuning all quantum dots in an array to $n = 1e$ with a common voltage, plotted as a function of array size in number of quantum dots. $V_{common}$ is chosen with two methods: one method in which $V_{common}$ is optimized to maximize the $n = 1e$ fraction and one method in which $V_{common}$ is set based on the mean of $V_{1e}$ and $V_{2e}$ data. The dashed horizontal line indicates the expected success rate for normally distributed $V_{1e}$ and $V_{2e}$ data in the limit of large array size. Stars indicate the success rate extracted from the measured 12QD data using each method. Simulations are performed with experimentally observed $V_{1e}$ variation (**c**) and with four times lower $V_{1e}$ variation (**d**).

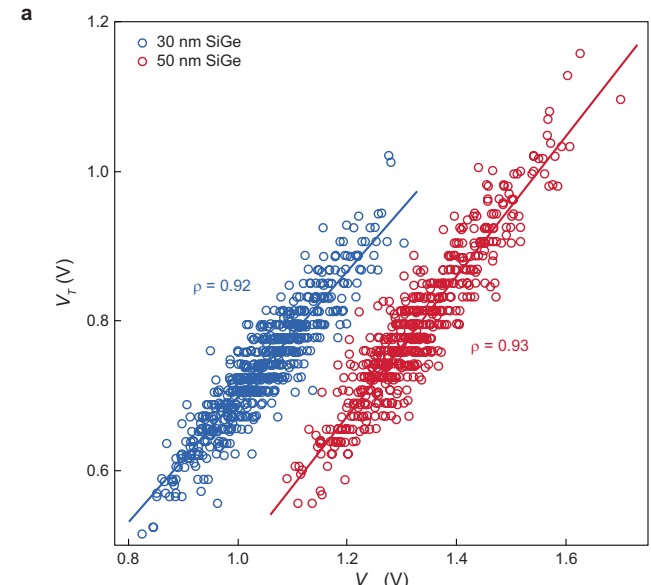

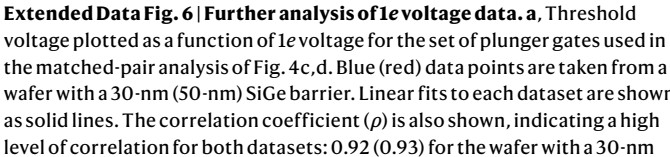

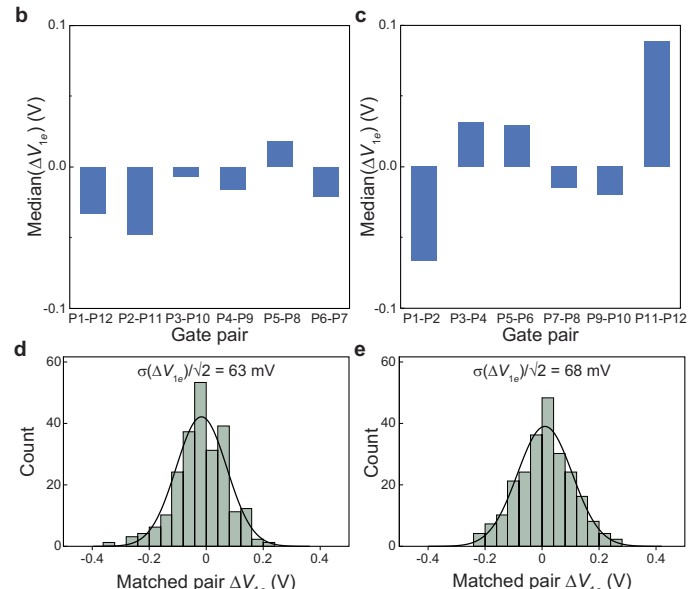

**Extended Data Fig. 6 | Further analysis of 1e voltage data. a**, Threshold voltage plotted as a function of 1e voltage for the set of plunger gates used in the matched-pair analysis of Fig. 4c,d. Blue (red) data points are taken from a wafer with a 30-nm (50-nm) SiGe barrier. Linear fits to each dataset are shown as solid lines. The correlation coefficient ($\rho$) is also shown, indicating a high level of correlation for both datasets: 0.92 (0.93) for the wafer with a 30-nm (50-nm) SiGe barrier. **b,c**, Median of matched-pair 1e voltage difference distributions from the wafer with a 50-nm SiGe barrier, plotted as a function of gate pair for sets of mirror-symmetric pairs (**b**) and nearest-neighbour pairs (**c**). **d,e**, Histograms of matched-pair 1e voltage differences from combined distributions of gate pairs for sets of mirror-symmetric pairs (**d**) and nearest-neighbour pairs (**e**).

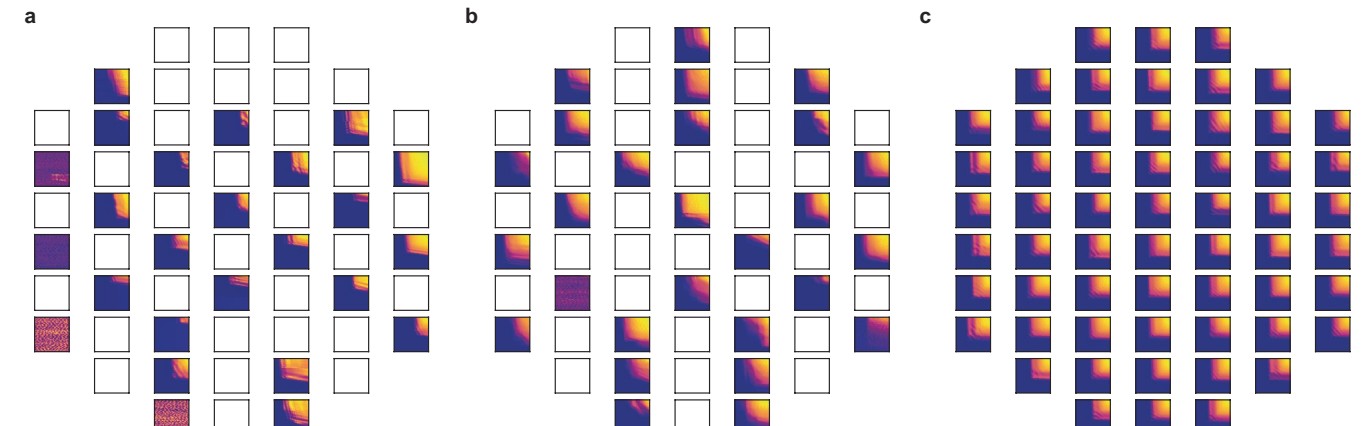

**Extended Data Fig. 7 | Barrier–barrier scans representing three versions of wafer fabrication.** Wafer-scale maps of barrier–barrier scans are shown to represent the three versions of wafer fabrication highlighted in Fig. 2b,c: high-$\kappa$ stack A with high-temperature spacer (**a**), high-$\kappa$ stack B with high-temperature spacer (**b**) and high-$\kappa$ stack B with low-temperature spacer and an integrated screening gate layer (**c**). Each set of scans shows a measurement from one quantum dot per device and represents the complete set from which the individual examples in Fig. 2c are taken. Scans are arranged by device location on the wafer. For the first two sets of measurements, only half of die are measured by sampling in a chequerboard pattern across the wafer. Further missing scans are because of non-yielding quantum dots on the earlier versions of fabrication.

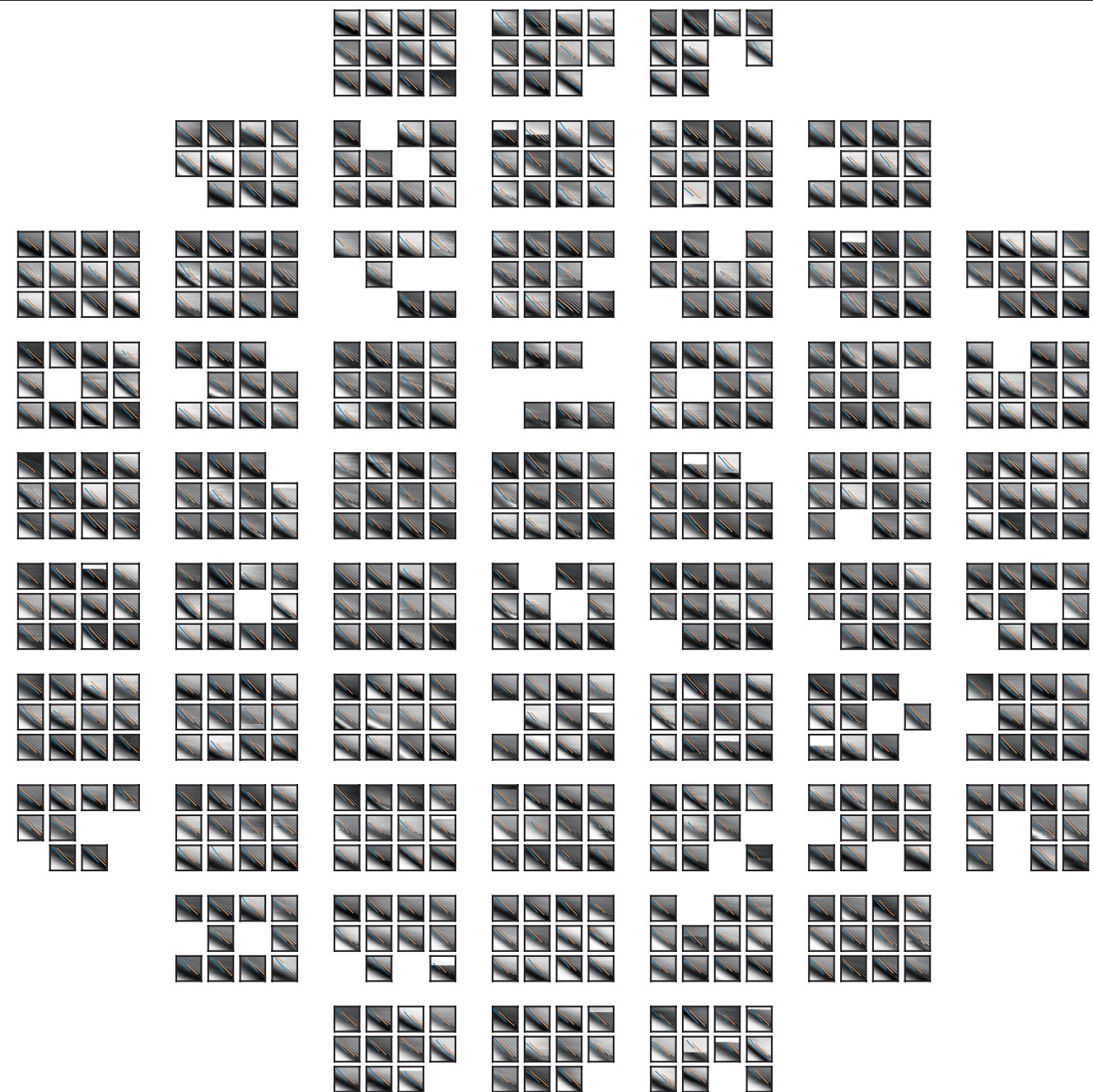

**Extended Data Fig. 8 | Charge-sensing data from wafer with a 30-nm SiGe barrier.** Charge-sensing scans are grouped by 12QD device and arranged by wafer location. Scans with unresolved transitions and/or fitting errors are removed. 1$e$ and 2$e$ transition curves identified by the analysis algorithm are plotted in blue and orange, respectively.

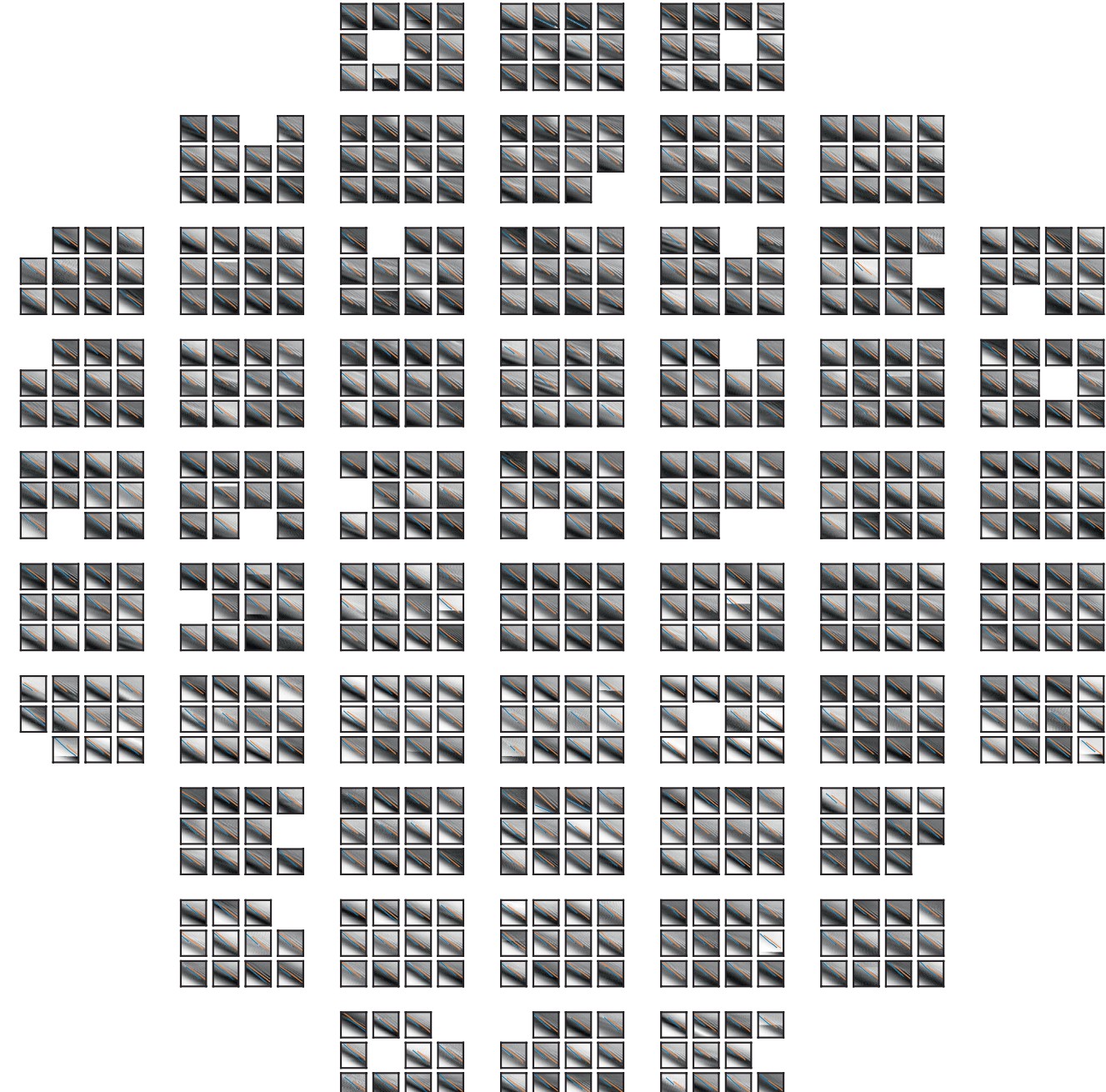

**Extended Data Fig. 9 | Charge-sensing data from wafer with a 50-nm SiGe barrier.** Charge-sensing scans are grouped by 12QD device and arranged by wafer location. Scans with unresolved transitions and/or fitting errors are removed. 1*e* and 2*e* transition curves identified by the analysis algorithm are plotted in blue and orange, respectively.

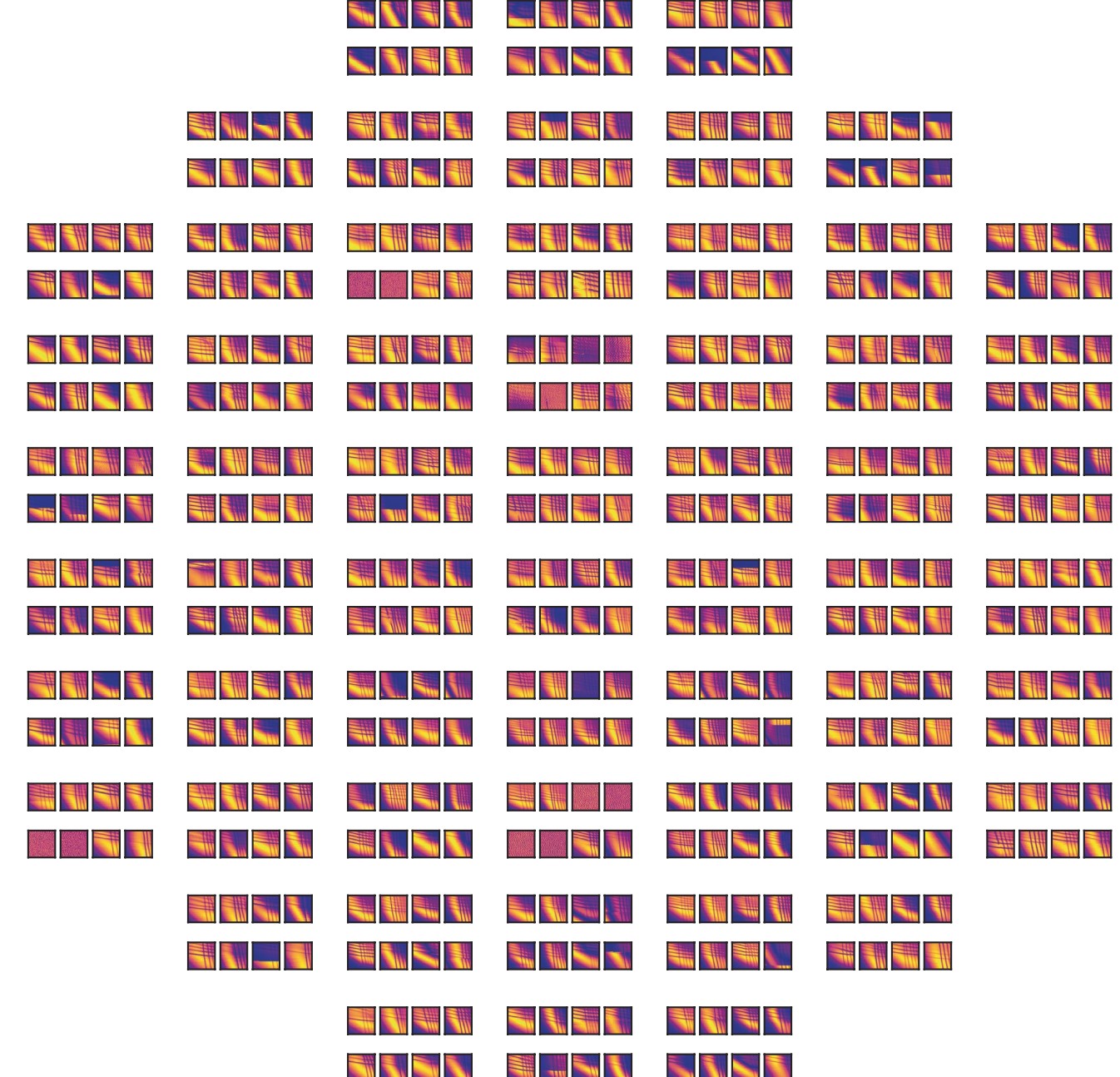

**Extended Data Fig. 10 | Charge sensing of double quantum dots across a wafer.** Charge-sensing scans are taken on eight double quantum dots per 12QD device (two pairs of quantum dots for each charge sensor) and arranged by wafer location.