## [Peer Review File · Nature]

Manuscript Title: Probing single electrons across 300 mm spin qubit wafers

Reviewer Comments & Author Rebuttals

Reviewer Reports on the Initial Version:

Referees' comments:

Referee #1 (Remarks to the Author):

The manuscript by Neyens et al. reports the Intel team's recent efforts to develop Si/SiGe spin qubit devices fabricated on 300 mm wafers. The paper is generally well-written and reports state-of-the-art results. Unlike the competitors in the field (CEA-Leti, IMEC, etc.), the authors utilize Si/SiGe heterostructure instead of Si-MOS for the reduced disorder. In addition, the fabrication is fully performed by EUV lithography, while the competitors partially rely on less reliable e-beam lithography. The research also involves a lot of technical advances that make the Si/SiGe material compatible with industrial 300 mm fabrication. The quality of the device reported seems to be comparable to the best academic devices used in some recent spin qubit demonstrations. A possible criticism from the community is the fact that the entire manuscript is written in a way somewhat vague about the materials and techniques used to improve the device quality, albeit it is to some extent understandable because the authors belong to a for-profit company.

For considering this manuscript for publication in Nature, a good reference would be their previous publication on Si-MOS devices (Ref. 23). In the paper, high-volume characterization at 1.6 K has already been reported (e.g., threshold voltage analysis), and some basic spin qubit characterizations (Rabi, Ramsey, spin echo, etc.) have also been shown. The main advances in this work are the use of low-disorder material, deeper statistical analysis, and autotuning. In my opinion, these are remarkable achievements toward scaling up semiconductor quantum computers. However, I also think that, while I have no doubt that the reported Si/SiGe devices work as spin qubits, perhaps it is reasonable to include basic examples of qubit operations in the manuscript given that the previous paper already demonstrates qubit operations and considering the high publication standard of Nature.

The followings are minor comments I would like the authors to address:

L82: For a comparable measurement of singulated dies in a standard cryostat, I don't expect such a large difference between the fridge temperature and electron temperature at 1K. Is there any particular reason why the cryo-prober system causes this discrepancy?

Fig. 2d: Is the superconducting material compatible with the external magnetic field needed for magnetizing micromagnets?

Fig. 2e: What kind of epitaxy improvements result in mobility improvement?

Fig. 2f: Can you elaborate more on what improvements have been made to improve the gate stack? I understand the full details of the materials and fabrication cannot be revealed due to the company's policy, but some more information about the problems resolved by improving the gate stack (interface quality, fixed charge in oxide, etc.) would be beneficial for the readers.

L235: Are the similar variations of $1e$ voltage (59mV and 60mV for 30nm QW and 50nm QW) just a coincidence?

L328-334: What kind of lithographical errors can cause the low pinch-off voltage? Naively, I expect

defective gates to result in high pinch-off voltage for these devices.

Extended Data Fig. 5: In some scans, double-dot-like features are visible (probably it means QD formation under the barrier gate). Are those considered successful QDs? For example, does this behavior cause problems in CZ gate implementation?

Extended Data Fig. 5: Some lines are curved due to the double-dot feature. How do you extract the charging energies in these cases?

Referee #2 (Remarks to the Author):

Silicon spin qubits from academic labs often suffer from low fabrication yield and large process variation. Since useful applications of quantum computing require the integration of a very large number of qubits, a key research to evaluate the scaling potential of silicon spin qubits is to explore how the device yield and variability improve leveraging state of the art industrial process facilities. To this end, gathering statistical data on the qubit quality is at the heart of process improvement, but the requirement of cryogenic operation usually hinders such feedback to the integration teams.

The paper presents the advancements of cryogenic wafer prober statistical measurements and the advances on the Si/SiGe spin qubit platform developed at Intel. Both aspects follow previous communications of the authors (Pillarisetty et al., IEDM (2019), Pillarisetty et al., IEDM (2021), or Henry et al., APS March meeting 2023 for instance). The latest progress reported in the manuscript on this Si/SiGe platform are certainly among the most advanced in terms of process quality and variability, and automatized statistical data acquisition.

Researchers from the spin qubit community or working on other quantum information processing platforms will be interested in the presented results. However, a lot of technical information is missing, probably considered as a trade secret, and the process optimizations that enabled such progress are not discussed. I think it affects the quality of the paper and the readers may finally find it of limited interest. Although the manuscript reports impressive advances in terms of technological achievement, it does not discuss the understanding behind such improvement, and do not present substantial advances in physics or characterization methodology. (see e.g. Ha et al., Nano Lett. 22, 1443, 2022). Therefore, I doubt that the paper significance meets Nature's publication standards.

You can find below a list of more technical comments and suggestions to improve the manuscript.

Line 74-76: "The quantum dot gate patterning is done in a single pass with extreme ultraviolet (EUV) lithography, allowing us to explore gate pitches from 50-100 nm."

- What is the gate pitch of the measured devices? How do the different gate pitches compare?

Line 107-110 & Fig. 2a,d: "Here gate line resistance is reduced through optimization of the gate fabrication process with normal-conducting materials and through the introduction of superconducting materials to the stack. Validating the superconducting process in particular is made possible by the 1.6 K base temperature of the cryo-prober."

- Details of the different gate stacks is missing. Without revealing the exact deposition processes, discussing the different materials and the conclusions about their pros and cons would be of interest for the reader.

- From Fig. 2a, I understand that this is a two-point measurement. Therefore, I guess that the

gate resistance plotted in Fig. 2d corresponds to the gate resistance itself + the finite access resistances (explaining the non-zero values of the superconducting stacks). What is the typical access resistance value? Is it comparable for all the studied gate stacks? How is explained the large resistance on wafer 3 despite the superconducting stack? Is the access resistance higher or the gate material itself is not fully superconducting? Is the CWP temperature comparable to the material TC or much lower?

- Did you explore the superconducting material properties? Critical magnetic field is not accessible in CPW but the critical currents and temperatures can be measured.
- The changes in the gate stack may have some impact on the device mobility, it would be interesting to show and discuss these results for the different stacks as well.

Line 116-118 & Fig. 2b, e: "Estimated carrier mobility across multiple wafers is shown in Fig. 2e. These measurements show a significant increase in the median mobility with a change in the epitaxial growth process designed to reduce defect density."

- Again, details of the different epitaxy processes is missing. Without revealing any trade secret on the exact growth conditions, do we compare similar heterostructure thicknesses?
- There are two populations on Fig. 2e, one showing a significantly reduced mobility. A comment in the Methods section attribute it to "a discrete defect mode of the larger-area Hall bar devices". It is not clear to me why discrete defects would have such a significant impact on large area devices. On the contrary, I would expect their impact to be averaged out. Can the authors give more details on their interpretation?

Line 121-126: "For quantum dot spin qubit arrays, process optimization involves many factors, including gross yield, quantum dot confinement, device stability, and voltage variation. To optimize these factors, we iterate through a wide variety of changes to the fabrication flow, including but not limited to fixed charge in the gate stack, thermal budget, etch impacts, and the integration of a screening gate layer. Through all these changes, a simple but useful metric for wafer quality is the cross-wafer spread in threshold voltage (V_T), the voltage required to turn on and off current with a particular gate. Fig. 2f shows V_T distributions for 15 wafers, highlighting three versions of the device stack: two intermediate stacks and the optimized stack."

- Does wafer quality only refers to the technology variability? Does it include more criteria?
- What is the definition of a device stack? Does it refer to the gate stack or does it include the heterostructure stack or more general process variations? Details of the device stack and a physics discussion about the limitations of the intermediate stacks would be interesting.
- The V_T spread reduction is commented in the text, but the mean V_T value evolves as well. Was it an objective to reduce the operation voltage of the devices (and in that case could you comment on the motivation)?
- There is no description of the device used for the V_T measurements. Fig. 2c,f suggest that the extraction is based on an array of 3 quantum dots. Since it is a multi-gate device, the V_T extraction procedure is non-standard and should be described, or refer to the Methods section on automated device measurements if it also applies here.

Line 135-138: "Fig. 2g shows examples of these measurements from each of the three stacks featured in Fig. 2f. The intermediate stacks show significant disorder and/or instability in these measurements. By comparison, the optimized stack shows clean confinement with the barrier gates and stable current throughout the length of the scan."

- Single stability diagrams are shown to illustrate the statistical device behavior from the different stacks. It raises the question of how representative they are. I would suggest adding wafer maps of the barrier-barrier diagrams (similar to the Extended Data Fig. 5 – 7) of wafers from each stack so that the reader can reliably see the effect of the process improvement.
- In addition to instabilities and disorder, the cross-capacitances are very different from one stack to the other. It seems almost negative on the optimized stack diagram. I doubt this is solely the

effect of an improved disorder. Can you comment on this point?

Line 323-324: "Gate yield is defined as the fraction of gates that can be used to turn on and pinch off their respective current channel."

- What are the current thresholds used to define on and off channels? Subthreshold slopes or pinch-off voltages are not criteria for the gate yield and possible outliers will be discarded only if they compromise the ability to identify a QD tune-up point, correct?

Line 324-326: "Quantum dot yield is defined as the fraction of quantum dot sites where a viable quantum dot tune-up point can be identified from barrier-barrier scans."

- What is the criteria for viable tune-up point? Is it based on the goodness of fit or gauged by eye? Please provide more details.

Line 173-176: "To estimate the random variation in V_T across gates and devices, we follow a standard CMOS industry method of analyzing matched pair V_T differences [19], calculated between mirror-symmetric pairs of gates. We subtract the mean from each gate-pair distribution to center them at zero and merge them into one distribution."

- From my understanding, each gate-pair distribution should be centered at zero due to the mirror-symmetry. Are the distribution means significantly away from zero? Unless there is a physical explanation for non-centered distributions, it seems to me that subtracting the means to center the distributions is inappropriate, since it artificially erases some statistic fluctuations, resulting in an over-optimistic matching distribution.

Line 192-200: "Changes in electron number are detected by modulating the voltage on an exterior screening gate and using lock-in detection of the charge sensor current at that frequency. A typical measurement is shown in Fig. 4a. In this 2D sweep, the horizontal axis is plunger voltage, and the vertical axis is the voltage of both barrier gates [36]. The sweep range is chosen to take each quantum dot from zero-electron to several-electron occupation along the plunger axis and from low tunnel rate ($\ll 1$ kHz) to high tunnel rate ($\gg 1$ GHz) along the barrier axis. Transition lines disappear at the bottom of the scan window where tunnel rate falls below the lock-in frequency (~ 1 kHz) and at the top of the scan window where the lines become broadened by tunnel coupling energy."

- Does the screening gate refers to the central screening gate separating the QDs from the sensors, or a gate from the first layer?

- On Extended Data Fig. 5, it seems that latching effects frequently occur (I suggest using vector rendering to avoid pixelation). It seems less frequent on Extended Data Fig. 6. Is it an effect of the larger SiGe barrier?

Line 231-237: "To analyze the variation in 1e transition voltage data, we repeat the same matched pair voltage difference analysis as above, taking differences between 1e voltages for mirrored pairs of plunger gates. The resulting distributions of voltage differences are shown in Fig. 4c-d for two wafers. The random variation in 1e voltage extracted from wafers with a 30 nm and 50 nm SiGe barrier are 59 mV and 60 mV, respectively. Both of these values closely agree with the random variation in gate V_T , meaning the random variation of a transistor-like metric (gate V_T) is matched by the random variation of a quantum metric (1e voltage). This implies that these devices are not subject to significantly increased disorder at the single electron regime compared with the many electron regime."

- On Fig. 4a, the cross capacitance between the dual barriers and the plunger are relatively the same (the charge degeneracy lines are almost diagonal). It means that both sets of gates have a comparable impact on the QD chemical potential and the position of the 1e voltage.

V_T is extracted from a pinch-off curve with a voltage compensation applied to the other gates so that their transconductance remains within a given range (please specify the thresholds). The $1e$ voltage is extracted after a finer barrier voltage compensation (corner point fitting on a barrier-barrier scan). Therefore, I do not think that we can make a direct comparison between both distributions. The $1e$ voltage distribution could be intrinsically larger but effectively reduced through the finer compensation on the barrier gates.

It would be very interesting to compare distributions with and without refined adjustment of the barriers to quantify its influence (either set to their value after recursive transconductance adjustment or even with both barriers set to a common gate voltage). How is the QD yield affected if the barrier-barrier fitting step is skipped? Is the experiment feasible?

Line 239-242: "Also, while the $1e$ voltage variation is nearly the same between the two wafers, the variation in chemical potential is better reflected by the ratios between $1e$ voltage variation and $1e$ - $2e$ addition voltage (Fig. 4e-f). These ratios are 1.0 ± 0.1 and 0.76 ± 0.08 for the 30 nm and 50 nm barrier wafer, respectively. The observation that the wafer with a deeper quantum well has a reduced ratio of this kind suggests that the $1e$ voltage variation is dominated by sources in the gate stack above the heterostructure."

- I expect a lower gate lever arm for a 50 nm SiGe barrier (compared to 30 nm). Assuming similar charging energies between the two splits, a difference in lever arm translates in a shift of the mean value of the addition energy (as can be seen on Fig. 4e-f) but also in a proportional increase in the standard deviation (that I do not see). I would also expect a similar lever arm dependence on the matching analysis (Fig. 4c-d), which is not observed. Is it another way to infer that the impact of disorder above the heterostructure surface is reduced in the 50 nm SiGe case?

I understand the ratio between the matched pair standard deviation and the addition voltage as a renormalization by the lever arm to obtain an estimate of the chemical potential distribution. Again the underlying hypothesis is that the average addition energy is indeed the same in both configuration. However with gates further away from the Si well (and same gate pitch), I expect a shallower potential profile that may result in different charging energies.

To support the assumption that charging energies are similar in both cases, were you able to deduce it e.g. from Coulomb diamonds measurements? Could you use a more precise energy-voltage conversion in the few electron regime, for instance using PAT, or is it incompatible with the CWP setup?

Line 414-418: "We note that the data used in this analysis comes from measurements of quantum dots tuned one at a time and that this method does not take into account the individualized setpoints of other gates in the array during measurements. Nevertheless, we believe it gives a reasonable estimate of the success rate of using shared voltages across a device to set a common charge state."

- Following a previous comment, I believe that the individual compensation on the barrier gates tends to narrow the distribution, resulting in an over-estimation of the percentage of QDs tunable to the $1e$ regime with a common voltage. Since this metrics is one of the novelty of the paper, I think it would be interesting to discuss the best extraction protocol and the associated uncertainties more extensively.

Referee #3 (Remarks to the Author):

This paper describes the operation of a 300mm cryogenic probe station for quantum dot devices and shows the level of uniformity that can be achieved with industrial fabrication. With the increased attentions on industrial integration of spin qubit devices, this result is very timely. The

quantum dot performance has been presented in a statistical relevant manner, and this is important to understand where the current device stands for upscaling considerations. The wafer-scale charge sensing down to the last electron is indeed impressive.

While the collection of cryogenic probe station results is an important step forward for industrial fabrication of better spin qubits, I am not convinced that the degree of advance, its interests to general readers, and the scientific insight for the research community meet the merits of publication in Nature. To be more specific:

1. Where to place this work on the roadmap towards the ambitious goal of a large scale spin qubit array? The works in Ref. 6-9, 12, 13, 14, 15, 16 showed proof-of-principle building blocks for a large-scale array, and the work in Ref. 17-18 explain qubit requirements for practical applications. It is hard for me to find a fit of this work as a novel achievement. Large-scale characterization is an important element in the industrial fabrication learning cycle, but by itself, readers cannot picture the advancement of spin qubit with charge only measurements, nor how to move forward with 1D design. The process variation can be a showcase of industrial fabrication, but qubits with industrial fab has been demonstrated. Studies on variations have also been reported before on both lab and fab devices (Ref. 21, Contamin et al IEDM2022 pp. 22-1, Elsayed et al arXiv:2212.06464, Bavdaz et al arXiv:2202.04482). The author also noticed that their reported uniformity is not enough for share control schemes (line 257, page 13). Moreover, not all spin qubit architecture requires voltage-sharing (e.g. Ref. 39).

2. How relevant is this study to spin qubits? The high yield of wafer-scale single electron regime does not necessarily promise high spin qubit yield. Moreover, the devices are quantum dot arrays, not full spin qubit devices. As authors pointed out in line 149 page 8, there is no micromagnet for spin control. Is micromagnet compatible with your industrial fabrication, and will placing the micromagnet degrade the quantum dot array? The devices can be, in principle, operated as exchange-only qubits. However, exchange-only qubits could suffer from the Meissner screening effects of the superconducting material in the gate stack (Ref. 9), which is presented in this work to reduce the line resistance (line 109, page 5).

3. Can you give more insight on optimization in the fabrication and device layout? As authors write in the paper, one of the two advancement of this work is "we optimize an industry-compatible process to fabricate spin qubit devices on Si/SiGe heterostructures, combining low process variation with a low disorder host material" (line 62, page 4). However, the discussion on the optimization is really limited. I believe more discussion on the fabrication optimization and how the critical steps are reflected on a wafer-scale measurement will be very beneficial to the spin qubit community and increase the impact of the paper. Additionally, the gate layout of the quantum dots is not show (Fig. 3b is the upper fanout of the gate, rather than the quantum dot defining region). If the authors want this study to "set a new standard for what can be achieved with spin qubit devices today and pave the way for significantly larger and more complex spin qubit arrays of the future", more detail about gate layout and geometry should be discussed, so that readers can have a starting point to contribute to the design of a large spin qubit array in an industrial compatible manner.

Some further comments about the paper:

1) Line 6, page 2. "Spin qubits [1, 4, 5] have shown impressive control fidelities [6-9] but have historically been challenged by yield and process variation." The yield and process variation are important problems this paper tries to solve. Good to back it up with references or more clarifications.

2) Line 13, page 2. "...this fabrication process..." Not clear to me what "this" is referring to. I assume you mean the final optimized process.

3) Line 15, page 2. "...to the fabrication and measurement of spin qubits." It would be helpful to clarify that the study focuses on quantum dots rather than full spin qubits.

4) Line 40, page 3. One of the biggest challenges of SiGe is the valley splitting. Good to also include that in the discussion for completeness.

- 5) Line 54, page 3. Is "full wafer probing" really necessary for spin qubits? Should all qubits in the 300mm wafer be uniform for practical operation considerations? The size of a large-scale spin qubit array would be in the mm to cm scale, which is way smaller than the 300mm wafer. I would think qubits need to be uniform within the array area, rather than the 300mm wafer scale. Other way of saying is that the short-range uniformity could be more relevant.
- 6) Line 72, page 4. As discussed above, it would be helpful to include a picture on the gate layout.
- 7) Line 100, page 5. Technically speaking, the structure shown in Fig. 2b is not Hall bar. There is also no Hall effects in the measurements. The carrier number is determined from an estimated gate capacitance. Out of curiosity: can you perform CV measurements to extract the capacitance as well as the threshold voltage? If you do, will the signal be limited by the area of the gate? An additional remark: from this discuss you can see the importance of showing device layouts and geometries to readers for easier access of your work.
- 8) Line 134, page 7. I would suggest rewriting for clarification. An example could be "The diagonal lines at the onset of the current is a signature of a single quantum dot between the barrier gates, where Coulomb blockade dominates the transport between source and drain".
- 9) Line 175, page 9. The match-pair method in standard CMOS uses nearest neighbour pairs. The work here uses mirror-symmetric pairs. This new analysis method need justification for the relevance of qubit operations. My intuitional feeling is that the bare standard deviation of each gate could be more relevant for the shared-control scheme. It can also give insight on if the variation is gate position dependent.
- 10) Line 192, page 10. What is the modulation energy? If it is too high, higher valley or orbital state could come into the pulsing window and affect the tunnelling rate extraction. I would also suggest more clarification or a reference on the charge sensing. It is not clear why electron number can be detected now.
- 11) Line 199, page 10. When the transition lines disappear, the tunnel rate is the lockin frequency or 2x the lockin frequency? In a lockin cycle, there should be a tunnel in and a tunnel out even.
- 12) Line 201, page 10. For charge sensing, one 12QD devices are measured per die. For gate V_T , four 12QD are measured per die (line 155). The reduced device for charge sensing is only measurement time concerns, or there are screening already based on gate V_T ?
- 13) Line 204, page 10. What kind of "noise" are you referring to? Lockin based charge sensing should be very immune to noise, unless its device instability, which I would not call it noise.
- 14) Line 208, page 10. It is not clear to me that lower charge noise would help charge sensing, unless the charge noise is really bad. It is more likely that at lower electron temperatures, the Coulomb peak will be narrower due to reduced thermal broadening, which gives larger slopes and increased the sensitivity.
- 15) Line 216, page 11. I think it would be clearer to the reader to say that the 1e transition voltage is measured at a fixed barrier gate voltage (of 0.7V?)
- 16) Line 235-239, page 12. It is not clear to me that the similarity in variation of gate threshold and the 1e transition could justify they are the same disorder limited. Can you make the scattered plot of the gate threshold vs the 1e transition of the same gate for different dots, and see if there is a correlation? Additionally, It would be good to add a reference to Contamin et al IEDM2022 pp. 22-1, where similar studies have been performed.
- 17) Line 240, page 12. It would be good to clarify why the ratio matters. Essentially, you are comparing the variation to the charging energy for share control.
- 18) Line 243-245, page 12. The discussion and reasoning are nice. I think more should be presented in the paper. Out of curiosity: do you have separate magnetic field based hall mobility analysis to compare the heterostructure of different interface depth?
- 19) Line 318, page 15. Are you referring to threading dislocations? With the hallbar area and the reduced mobility percentage, you can estimate a defect density (assuming 1 per low mobility hallbar). Will that agree with the TDD of the SiGe wafer?
- 20) Line 349, page 16. You define V_T with a threshold current method. In many CMOS studies, V_T is defined as the maximum transconductance voltage. Can you comment on why you use this V_T definition?

Referee #4 (Remarks to the Author):

Paper Ref : 2023-07-11832

Review of "Probing single electrons across 300 mm spin qubit wafers"

Authors : Samuel Neyens*, Otto Zietz*, Thomas Watson, Florian Luthi, Aditi Nethewala, Hubert George, Eric Henry, Andrew Wagner, Mohammad Islam, Ravi Pillarisetty, Roza Kotlyar, Kent Millard, Stefano Pellerano, Nathan Bishop, Stephanie Bojarski, Jeanette Roberts, and James S. Clarke.

Feedbacks:

The authors present an in-depth study on a 300 mm wafer characterization methodology for linear spin devices and multi-qubits (12 QD) at cryogenic temperature (1.6K). The proposed solution is based on an advanced 50nm-100nm gate pitch CMOS process with a way to automate the measurements and a way to optimize the process step to improve QD quality/fidelity. This opens the door to a large-scale quantum dot solution with industrial techniques for massive integration/qualification. As a results, the text and context of the study are well structured with clear argumentation and data.

They present the methods and the results as well as the cryo-probe equipment used for this development based on their own technology with Si/SiGe heterostructure and specific gate stack with/without ²⁸Si isotopic substrate at cryogenic temperature of 1.6K. The Ohmic; The barrier/plunger line structures are characterized with their key parameters and describe the distribution before and after process correction with a standard deviation metric (at least). Then, the 12 QDs with charge sensor are also analyzed with the full distribution measurements along the device topology based on the barrier/plunger biases. Additionally, a Hall bar structure is carried out to evaluate the mobility of the carrier which is an important parameter for QD. The methodology leads to systematic characterization with a wide range of data and KPIs for the process and QD and their quality results provide a 0e/1e/2e stability diagram with detection lock in.

It is an interesting and significant study that reaches today with their industrial approach (according to KPIs measurements): 100% for ohmics and gates, 99.8% for 1 QD and 96% for 12QD. These results are very promising, even if other challenges remain to be overcome.

To enrich this proposal, please find below some suggestions, comments and open questions:

- for the qubit, specify whether it is physical or logical and explain that it is an electronic qubit at the beginning of the text
- for statistical metrics on Quantum Dot indicate which KPI will be used throughout the study
- for process variation, indicate which physical parameters are monitored
- scaling is also driven by the Ion/Ioff KPI or indicate the FOM (lines 30-31)
- on line 45 indicate the parameters monitored with the statistical measurements
- in line 69 a cross section device will be welcome (see fig. 2)
- It is indicated that it takes 2 hours for thermal cooling, what about the other steps?
- On line 89: what does "variety of metrics" mean, please list parameters
- On line 96, the PAD dimension must be indicated with the BEOL type/specification
- line 300: For the hall bar, indicate the type of carrier being investigated. What are the results when applying a magnetic field (1T? or with the EDR map)? What about in the 1e regime?
- On line 107, indicate the value of the RF signal applied to the device.

- Indicate the gate stack structures (link to fig.2) with the threshold voltage 3V, 1.8V & 0.8V. What is the gate stack optimization strategy?
- What is the specification of the left/right barrier "immunity" and how to choose the right value (in figure 2g the range is 3.5 nA (StackA) for V [3.15 V – 3, 05 V] to 25 nA V [1 V – 1.05 V] (optimized stack)
- Line 110, the temperature is indicated at 1.6K so what about IP self-heating?
- Line 139, what is the way to optimize the process as the text claims?
- Line 149 which type of micro magnet is targeted: Type and level in metal stack, magnitude of field B and orientation, Gradient with Map according to confinement.
- Please justify the choice of EDSR compared to ESR?
- On line 166, what is the root cause of the structure's VT boundary shifting (only due to gate accumulation)? How to solve it?
- Line 180, the standard deviation is improved to 58mV, what is the criterion/value to be considered "correct" ?
- Line 192, what is the voltage modulation range?
- On line 206, what does "as gauged by eye" mean (behind it will be a parameter value!)
- Explain the "Gamma" factor in the text (see fig. 4)
- Line 201, what is the noise immunity for charge sensing to follow the 1e/2e transition? And what about for a large linear array?
- Line 245-250, fixed charge process parameters are discussed with reduced gate oxide thickness lever, what about sidewall (if relevant in this technology)?
- The comparison on two wafers in the 1e transition measurement, is it the same batch?
- In line 271, the study is about QD, so what are arguments to extend to the Spin qubit?

- Note: The Methods section is well introduced.
- Note line 349, what is the noise immunity strategy for all measurements?
- Note: It should be interesting to explain why the study focuses on the electron spin qubit rather than the hole spin qubit.
- Note: It should be interesting to comment on some perspectives/vision on the next steps in the full characterization of spin qubits: Rabbi oscillation ... Exchange interaction function and entanglement ... Spin manipulation cross talk... Inter connection and power budget for large scale quantum computing solution(s).

This robust and relevant study could be published with an improvement in the results following the previous questions/remarks. While waiting for the pleasure of seeing the update again if it is accepted for the future.

The results of this study will be useful to the community and industry in this field

Author Rebuttals to Initial Comments:

Referees' comments:

Referee #1 (Remarks to the Author):

The manuscript by Neyens et al. reports the Intel team's recent efforts to develop Si/SiGe spin qubit devices fabricated on 300 mm wafers. The paper is generally well-written and reports state-of-the-art results. Unlike the competitors in the field (CEA-Leti, IMEC, etc.), the authors utilize Si/SiGe heterostructure instead of Si-MOS for the reduced disorder. In addition, the fabrication is fully performed by EUV lithography, while the competitors partially rely on less reliable e-beam lithography. The research also involves a lot of technical advances that make the Si/SiGe material compatible with industrial 300 mm fabrication. The quality of the device reported seems to be comparable to the best academic devices used in some recent spin qubit demonstrations.

We thank the referee for their careful reading of our manuscript and for these supportive comments. Below we give our full response to the referee's comments and questions and discuss our modifications to the manuscript.

A possible criticism from the community is the fact that the entire manuscript is written in a way somewhat vague about the materials and techniques used to improve the device quality, albeit it is to some extent understandable because the authors belong to a for-profit company.

We thank the referee for this comment. We accept the referee's criticism (shared by other referees) that the manuscript could use further description of the methods used to improve device quality in order to be clearer to the reader. We are limited to some extent based on restrictions from our institution, but we have made changes in the revised manuscript that we think will help address this criticism. We have now added to the manuscript a deeper discussion about the structure of devices as well as how they have been improved. Specifically, we now highlight two physical changes to devices to which we attribute improved performance. The first is a reduction in fixed charge in our dielectric stack through a combination of the choice of high-k material and a reduction in thermal budget of processes that occur after high-k deposition. The second is the addition of a screening gate layer to explicitly provide lateral confinement for our quantum dots. We have also added a 3D device schematic to Fig. 2 to more clearly show the structure of the gate and dielectric layers that define the quantum dots. We hope that, with this figure, readers are better able to visualize the full structure of the active device area and can better understand how these results are achieved. We also expect the more explicit discussion of fixed charge serves to deepen later discussions of voltage variation, as remaining charge disorder is likely a significant contributor to such variation.

For considering this manuscript for publication in Nature, a good reference would be their previous publication on Si-MOS devices (Ref. 23). In the paper, high-volume characterization at 1.6 K has already been reported (e.g., threshold voltage analysis), and some basic spin qubit characterizations (Rabi, Ramsey, spin echo, etc.) have also been shown. The main advances in this work are the use of low-disorder material, deeper statistical analysis, and autotuning. In my opinion, these are remarkable achievements toward scaling up semiconductor quantum computers. However, I also think that, while I have no doubt that the reported Si/SiGe devices work as spin qubits, perhaps it is reasonable to include basic examples of qubit operations in the manuscript given that the previous paper already

demonstrates qubit operations and considering the high publication standard of Nature.

We thank the referee for this suggestion. We agree that, while the focus of this paper is on electrostatic characterization, the lack of qubit data in the manuscript could leave the question open for the reader whether the devices discussed really represent an improvement over previously published devices. To address this, we have added a new figure to the Extended Data section showing measurements of Loss-DiVincenzo spin qubits on multiple wafers, including a case where the entire 12-qubit linear array in a single device is fully tuned up. In total, we now show data from 39 different qubits compared to three shown in [Zwerver et al. 2022]. Furthermore, we also demonstrate single qubit gate errors more than an order of magnitude lower than in [Zwerver et al. 2022] and now in-line with state-of-the-art gate fidelities for Si/SiGe Loss-DiVincenzo spin qubits. This results in a gate fidelity that is by far the highest reported for a device made with CMOS industry manufacturing. We believe this will help to clarify to the reader that this platform really is compatible with high performing spin qubits and, in combination with the improvements in yield and electrostatic reliability already demonstrated, represents a genuine leap forward in the scalability of semiconductor spin qubits. We believe this figure, as well as new text added to the conclusion of the manuscript, will help clarify the important link from advancements in yield and variation to continued scaling of the spin qubit technology.

The followings are minor comments I would like the authors to address:

L82: For a comparable measurement of singulated dies in a standard cryostat, I don't expect such a large difference between the fridge temperature and electron temperature at 1K. Is there any particular reason why the cryo-prober system causes this discrepancy?

We attribute this temperature difference to one or both of the following effects: insufficient thermalization of the DC wiring and/or thermal resistance between the wafer and the chuck on which it is mounted. We have added additional comments to this effect in the "Electron temperature measurement" section of the Methods.

Fig. 2d: Is the superconducting material compatible with the external magnetic field needed for magnetizing micromagnets?

To address this as well as related questions about the superconducting layer properties, we have added an expanded section on this topic to the Supplementary Information, including critical field and critical current measurements. The short answer is that both superconducting layers (interconnect and gate) do have a critical field (>0.7 T) that could support spin qubit operation. While higher magnetic fields (~ 3 T) are typically used to magnetize our micromagnets, we expect no need for materials to remain superconducting during this process, since such a magnetization sweep occurs in advance of qubit operation. The more significant limitation we find with the superconducting gate layer is the critical current. By contrast, the superconducting interconnect layer has much higher critical current and field. We also include in this new section a discussion of the advantages and limitations of these two superconducting layer processes as they stand today, and the potential benefits and drawbacks to using them. We also want to note that the data and discussion of gate line resistance test structures have now been moved to the Extended Data and Methods section "Gate line resistance measurements," respectively, to make space to address other comments from all referees in the main text.

Fig. 2e: What kind of epitaxy improvements result in mobility improvement?

We first note that data and discussion of Hall bar test structures has now been moved to the Extended Data and Methods section “Carrier mobility measurements,” respectively, in order to make space in the main text. In this Methods section, we have expanded our discussion of mobility measurements, including our explanation of how mobility improvements are achieved. The epitaxy improvements that we mention in the text are a change in growth conditions that result in a reduction in background oxygen concentration in the Si quantum well. Since submitting the first version of the manuscript, we have also become aware that this mobility improvement can also be attributed to a second effect, which was an increase in SiGe barrier thickness (from 30 nm to 50 nm) that occurred around the same time as the change in growth conditions. We have now added data from two more wafers to the original plot in order to illustrate the effect of each of these factors. The resulting best mobility that we report has been improved through a reduction in remote impurity scattering (from moving to deeper quantum wells) as well as a reduction in scattering centers in or near the quantum well (from reducing background oxygen concentration).

Fig. 2f: Can you elaborate more on what improvements have been made to improve the gate stack? I understand the full details of the materials and fabrication cannot be revealed due to the company’s policy, but some more information about the problems resolved by improving the gate stack (interface quality, fixed charge in oxide, etc.) would be beneficial for the readers.

As mentioned above, we have now added more discussion to the main text about the process changes to which we attribute improvements in the device performance. We have also added a new graphic to Fig. 2 to illustrate what we mean by “gate stack,” which we believe will help the reader better understand these changes as well as the resulting device structure after process optimization. The referee’s suggestion is correct that a major part of this optimization has been the reduction of fixed charge in the dielectric stack. Concurrent with this, improvements in confinement have been achieved by integrating a designated screening gate layer, which we find has significantly improved the reliability of the devices. To better elucidate this development for the reader, we have moved the first reference of a screening gate layer to this section on device improvements.

L235: Are the similar variations of 1e voltage (59mV and 60mV for 30nm QW and 50nm QW) just a coincidence?

We believe that the similarity in 1e voltage variation between these two wafers with different barrier thickness could indeed be a coincidence. We do not necessarily expect the voltage variation to be independent of barrier thickness. In fact, the metric we think is more revealing, which is the ratio between 1e voltage variation and 1e-2e addition voltage, is different between the two wafers, which we attribute to the 50 nm deep quantum well being less impacted by sources of variation at/above the surface of the heterostructure. We have also observed that, as we experiment with process changes on wafers, this voltage variation can go up and down. (e.g., a later wafer with 50 nm SiGe barrier is now featured in a Supplementary Information section on barrier voltage tuning, and this wafer shows lower voltage variation than either wafer shown in the main text.) We also note that the similarity in 1e voltage variation between the two wafers is not a result we are seeking to highlight in the manuscript. Rather, the result we believe to be most impactful is the similarity in variation between 1e voltage and threshold voltage, showing that voltage variation at the spin qubit operating regime (1e) is close to the voltage variation in the transistor regime (many electrons). This is emphasized further in a new Extended Data figure showing strong correlation between 1e voltage and threshold voltage values for both wafers. As discussed in the text, we believe that this points towards a path to reducing variation at the

spin qubit operating regime using similar techniques that have been successfully used to reduce variation in transistors.

L328-334: What kind of lithographical errors can cause the low pinch-off voltage? Naively, I expect defective gates to result in high pinch-off voltage for these devices.

We believe this is an error in achieving the right gate dimension that results in those gates having weaker “action,” or capacitance to the channel. While the gates can still pinch off current, their pinch-off voltage (or V_T) is an outlier in the distribution. If these gates were solely responsible for turning on the channel, we agree with the referee’s expectation that this would result in higher V_T . However, we expect that the cross-capacitance from other gates effectively makes the neighboring gates responsible for turn-on in that region. The low pinch-off voltage on the gate in question is therefore a result of its comparative weakness in overcoming the action of surrounding gates to turn off the channel.

Extended Data Fig. 5: In some scans, double-dot-like features are visible (probably it means QD formation under the barrier gate). Are those considered successful QDs? For example, does this behavior cause problems in CZ gate implementation?

Here we draw a distinction between yield and performance. “Yielding” quantum dots (as defined in the “Yield analysis” Methods section) are those where a tune-up point can be identified in barrier-barrier scans taken in the transport regime. In practice, we find this is an excellent indicator that quantum dots will be able to be tuned to the single electron regime, where more in-depth measurements (such as extraction of transition voltages) can be taken. In this regime, there are indeed quantum dots that have different amounts of disorder, as can be seen in the Extended Data figures. We consider this to be a question of the “performance” of those quantum dots. Cases where transition lines have more curvature are indicative of worse performance and could indeed cause difficulty for multi-dot operations at mK temperatures, such as exchange operations between two single spin qubits or within an exchange-only qubit. We note that not all cases of such curvature would necessarily impact exchange operation, such as the cases where curvature emerges at extremely high tunnel coupling, beyond where such operations would typically be done. For this reason, systematically grading the performance of such scans is still an active area of development, but we point out that the wealth of charge sensing data obtained using these automated measurements will be a significant aid in developing these more advanced spin qubit screening criteria based on single electron performance. Finally, we note that we have observed significant reductions in the probability of these disordered transitions by moving from 30 nm to 50 nm SiGe barrier, further suggesting the important role played by disorder in the dielectric stack.

Extended Data Fig. 5: Some lines are curved due to the double-dot feature. How do you extract the charging energies in these cases?

In all cases, we extract the addition voltages (our proxy for charging energies) using the procedure described in the Methods section titled “Charge sensing transition curve analysis.” Specifically, the $1e$ and $2e$ transition voltages are defined as the plunger voltages at which the respective transition curves cross the midpoint of the barrier voltage axis. We do not adapt the procedure to account for cases of transition line disorder (i.e., curvature). We believe the effect of this procedure is that such disorder is expressed in the spread of the distribution of $1e$ - $2e$ addition voltages (shown in Fig. 4e-f of the main text).

Referee #2 (Remarks to the Author):

Silicon spin qubits from academic labs often suffer from low fabrication yield and large process variation. Since useful applications of quantum computing require the integration of a very large number of qubits, a key research to evaluate the scaling potential of silicon spin qubits is to explore how the device yield and variability improve leveraging state of the art industrial process facilities. To this end, gathering statistical data on the qubit quality is at the heart of process improvement, but the requirement of cryogenic operation usually hinders such feedback to the integration teams.

The paper presents the advancements of cryogenic wafer prober statistical measurements and the advances on the Si/SiGe spin qubit platform developed at Intel. Both aspects follow previous communications of the authors (Pillarisetty et al., IEDM (2019), Pillarisetty et al., IEDM (2021), or Henry et al., APS March meeting 2023 for instance). The latest progress reported in the manuscript on this Si/SiGe platform are certainly among the most advanced in terms of process quality and variability, and automatized statistical data acquisition.

We thank the referee for their careful reading of our manuscript and for these supportive comments. Below we give our full response to the referee's comments and questions and discuss our modifications to the manuscript.

Researchers from the spin qubit community or working on other quantum information processing platforms will be interested in the presented results. However, a lot of technical information is missing, probably considered as a trade secret, and the process optimizations that enabled such progress are not discussed. I think it affects the quality of the paper and the readers may finally find it of limited interest. Although the manuscript reports impressive advances in terms of technological achievement, it does not discuss the understanding behind such improvement, and do not present substantial advances in physics or characterization methodology. (see e.g. Ha et al., Nano Lett. 22, 1443, 2022). Therefore, I doubt that the paper significance meets Nature's publication standards.

We thank the referee for these constructive comments. We accept the referee's criticism (shared by other referees) that the manuscript could use further description of the methods used to improve device quality in order to be clearer to the reader. We are limited to some extent based on restrictions from our institution, but we have made changes in the revised manuscript that we think will help address this criticism. We have now added to the manuscript a deeper discussion about the structure of devices as well as how they have been improved. Specifically, we now highlight two physical changes to devices to which we attribute improved performance. The first is a reduction in fixed charge in our dielectric stack through a combination of the choice of high-k material and a reduction in thermal budget of processes that occur after high-k deposition. The second is the addition of a screening gate layer to explicitly provide lateral confinement for our quantum dots. We have also added a 3D device schematic to Fig. 2 to more clearly show the structure of the gate and dielectric layers that define the quantum dots. We hope that, with this figure, readers are better able to visualize the full structure of the active device area and can better understand how these results are achieved. We also expect the more explicit discussion of fixed charge serves to deepen later discussions of voltage variation, as remaining charge disorder is likely a significant contributor to such variation.

We argue furthermore that, while this work does follow other publications sharing measurement and analysis of $1e$ voltage statistics [Ha et al. 2022; Contamin et al. 2022], this work takes such analysis significantly further. This is made possible through the increased size of the charge sensing datasets presented here, including hundreds of spin qubit arrays (now emphasized in the abstract) and

thousands of quantum dots. This increase in measurement scale is achieved through the unique combination of the cryo-prober measurement system, the automated measurement and analysis procedure, and the high yield and low disorder of the presented Si/SiGe devices. In addition to the matched pair 1e voltage variation analysis presented in the first version of the manuscript, we are pleased to share in this revision: the analysis of correlation between 1e voltages of all plunger gate pairs in the 12 quantum dot devices (Supplementary Fig. 3); analysis of correlation between 1e voltages and threshold voltages (Extended Data Fig. 7); and an analytical treatment of the effects of cross capacitance and barrier voltage tuning on metrics of 1e voltage variation (Supplementary Information section II). This last addition is also backed up by a new dataset comparing two methodologies of extracting the 1e voltage. We believe these comprise significant advancements in the understanding of the factors involved in the statistical analysis of 1e voltages, something we believe will be helpful for the field as it moves towards the practice of measuring 1e voltage statistics at the scale presented in this work.

We would also like to highlight several more factors that we believe contribute to the paper's significance. Firstly, this work demonstrates the first combination of a low disorder heterostructure stack (Si/SiGe) with CMOS industry manufacturing, a key step towards extending the impressive results achieved with Si/SiGe devices towards larger array sizes. Secondly, the devices studied here are leading in size and complexity for spin qubit devices. They contain arrays of up to 12 spin qubit sites with four companion charge sensor quantum dots. We demonstrate reliable charge sensing and tuning to the single electron regime at all sites of the array. Thirdly, with the addition of qubit data to this revision (Extended Data Fig. 1 and Methods), we show measurements of Loss-DiVincenzo spin qubits on multiple wafers, including a case where the entire 12-qubit linear array in a single device is fully tuned up. In total, we now show data from 39 different qubits compared to three shown in an earlier work [Zwerver et al. 2022]. Furthermore, we also demonstrate single qubit gate errors more than an order of magnitude lower than in [Zwerver et al. 2022] and now in-line with state-of-the-art gate fidelities for Si/SiGe Loss-DiVincenzo spin qubits. This results in a gate fidelity that is by far the highest reported for a device made with CMOS industry manufacturing. Altogether, we argue these factors represent significant novelty and serve to move the field forward on the path towards large-scale spin qubit processors.

You can find below a list of more technical comments and suggestions to improve the manuscript.

Line 74-76: "The quantum dot gate patterning is done in a single pass with extreme ultraviolet (EUV) lithography, allowing us to explore gate pitches from 50-100 nm."

- What is the gate pitch of the measured devices? How do the different gate pitches compare?

The measurements presented in the manuscript are all from devices with 60 nm gate pitch. We have added a line to the manuscript to clarify this. The comment about other gate pitches is mainly intended to show the range of pitches achieved with this mask set using EUV patterning. We have tested 50 nm and 70 nm pitch devices and found them to also work well, although we have not tested them extensively enough for a statistical comparison at the same level of depth as our data on 60 nm gate pitch. We consider a comparison like this to be a possible subject of future work.

Line 107-110 & Fig. 2a,d: "Here gate line resistance is reduced through optimization of the gate fabrication process with normal-conducting materials and through the introduction of superconducting materials to the stack. Validating the superconducting process in particular is made possible by the 1.6 K base temperature of the cryo-prober."

- Details of the different gate stacks is missing. Without revealing the exact deposition processes,

discussing the different materials and the conclusions about their pros and cons would be of interest for the reader.

We first want to note that the data and discussion of gate line resistance test structures have now been moved to the Extended Data and Methods section “Gate line resistance measurements,” respectively, in order to make space to address other comments from all referees in the main text. In this Methods section, we have expanded our discussion of how resistance improvements are achieved. We have also added a new section in the Supplementary Information that includes characterization of the superconducting critical parameters and a discussion of the tradeoffs of using one or both superconducting layer processes in a device stack for qubit applications.

- From Fig. 2a, I understand that this is a two-point measurement. Therefore, I guess that the gate resistance plotted in Fig. 2d corresponds to the gate resistance itself + the finite access resistances (explaining the non-zero values of the superconducting stacks). What is the typical access resistance value? Is it comparable for all the studied gate stacks? How is explained the large resistance on wafer 3 despite the superconducting stack? Is the access resistance higher or the gate material itself is not fully superconducting? Is the CWP temperature comparable to the material TC or much lower?

The cryo-prober measurements are indeed two-point resistance measurements. For the data shown in the figure, the access resistance of 30 ohms is subtracted. The non-zero resistance on wafer 3 is due to one of the metal layers (the interconnect layer) still using a normal-conducting material. For wafer 4, both gate and interconnect layers are made using superconducting materials. While the resistance is much lower than wafer 3, since it is still non-zero (median ~6 ohms), we cannot confirm full superconductivity. Possible sources of remaining resistance could include the imprecision of the access resistance subtraction and/or a normal-conducting via between the gate and interconnect layers. For this reason, we refer to both wafer 3 and 4 as “partial superconducting” rather than fully superconducting. We measure the critical temperature (T_c) to be ~2.4 K and ~6.5 K for the gate and interconnect layer, respectively. While the gate layer T_c is comparable to the cryo-prober temperature, we note it is much higher than the mK temperatures at which qubit experiments would be performed. We have added clarifications of all of the above to Methods section titled “Gate line resistance measurements” and Supplementary Information section I.

- Did you explore the superconducting material properties? Critical magnetic field is not accessible in CPW but the critical currents and temperatures can be measured.

We now report these details in section I of the Supplementary Information. We note that they are all measured in a conventional cryostat, as the cryo-prober is not currently configured with precise enough temperature control to collect temperature dependent data.

- The changes in the gate stack may have some impact on the device mobility, it would be interesting to show and discuss these results for the different stacks as well.

While we do not have measurements comparing the mobility between devices made with different gate materials, we do find that, for the heterostructures with a deeper quantum well (50 nm SiGe barrier), peak mobility is not limited by remote scattering centers, and so likely is not influenced as much by gate stack materials as the structures with shallower quantum well (30 nm SiGe barrier). That being said, our observation that mobility is reduced by remote scattering centers in the case of the 30 nm SiGe barrier points to the benefits of further improvements to our dielectric stack, namely through reductions in

fixed charge (as we now say in the main text). Therefore, measurements of mobility as a function of gate dielectric process and gate material (for heterostructures with 30 nm SiGe barrier) will be an important test when making further improvements to the gate stack in our future work.

Line 116-118 & Fig. 2b, e: "Estimated carrier mobility across multiple wafers is shown in Fig. 2e. These measurements show a significant increase in the median mobility with a change in the epitaxial growth process designed to reduce defect density."

- Again, details of the different epitaxy processes is missing. Without revealing any trade secret on the exact growth conditions, do we compare similar heterostructure thicknesses?

We note that data and discussion of Hall bar test structures has now been moved to the Extended Data and Methods section "Carrier mobility measurements," respectively, in order to make space in the main text. In this Methods section, we have expanded our discussion of mobility measurements, including our explanation of how mobility improvements are achieved. The epitaxy improvements that we mention in the text are a change in growth conditions that result in a reduction in background oxygen concentration in the Si quantum well. In the course of answering the referee's question about heterostructure thickness, we have also found that the improvements in mobility can be attributed to two mechanisms: the aforementioned reduction in background oxygen concentration, as well as a concurrent transition to heterostructures with a deeper quantum well, namely, from 30 nm to 50 nm SiGe barrier. For all wafers shown, we have now specified the SiGe barrier thickness as well as the Si quantum well thickness (~5 nm for all wafers). In order to help separate out the two sources of mobility improvement, we have also added data from two more wafers to the existing figure as well as Hall mobility measurements taken as a function of density in a conventional cryostat. These confirm that our observed best mobility is achieved through a combined reduction in remote scattering, by moving the quantum well further from the gate stack, and a reduction in scattering in or near the quantum well, by reducing the background impurity concentration.

- There are two populations on Fig. 2e, one showing a significantly reduced mobility. A comment in the Methods section attribute it to "a discrete defect mode of the larger-area Hall bar devices". It is not clear to me why discrete defects would have such a significant impact on large area devices. On the contrary, I would expect their impact to be averaged out. Can the authors give more details on their interpretation?

We have added a longer discussion of our interpretation of this effect to the Methods section "Carrier mobility measurements." We agree with the referee's expectation for the case that the defect size is small compared to the width of the Hall bar device. However, for the case where defect pile-ups occupy an area that is comparable to the width of a hall bar, then we would expect an instance of overlap between a defect pile-up and a Hall bar to have a significant impact on extracted mobility, as the effective width of high mobility paths through the Hall bar would become highly restricted in the vicinity of the defect pile-up. As we now discuss in the Methods section, we expect if we were to move to larger Hall bar dimensions, we would see more averaging out of the impact of individual defects, likely resulting in a more unimodal distribution of mobility.

Line 121-126: "For quantum dot spin qubit arrays, process optimization involves many factors, including gross yield, quantum dot confinement, device stability, and voltage variation. To optimize these factors, we iterate through a wide variety of changes to the fabrication flow, including but not limited to fixed charge in the gate stack, thermal budget, etch impacts, and the integration of a screening gate layer. Through all these changes, a simple but useful metric for wafer quality is the cross-wafer spread in

threshold voltage (VT), the voltage required to turn on and off current with a particular gate. Fig. 2f shows VT distributions for 15 wafers, highlighting three versions of the device stack: two intermediate stacks and the optimized stack."

- Does wafer quality only refers to the technology variability? Does it include more criteria?

To help answer this question and related questions about optimization of the devices, we have now added to the manuscript a deeper discussion about the structure of devices as well as how they have been improved. Specifically, we now seek to highlight two physical changes to devices to which we attribute improved performance. The first is a reduction in fixed charge in our dielectric stack through a combination of the choice of high-k material and a reduction in thermal budget of processes that occur after high-k deposition. The second is the addition of a screening gate layer to explicitly provide lateral confinement for our quantum dots. The reduction in fixed charge is represented not only by a reduction in variability of threshold voltage but also a reduction in the absolute values of threshold voltage. The improvements in confinement are represented by improvements in the stability and reduction in the disorder observed in the quantum dot transport scans ("barrier-barrier scans") also highlighted.

- What is the definition of a device stack? Does it refer to the gate stack or does it include the heterostructure stack or more general process variations? Details of the device stack and a physics discussion about the limitations of the intermediate stacks would be interesting.

As mentioned above, we have now added a longer discussion of the device improvement process, as well as more explanation of the structure of devices including a 3D schematic of the gate and dielectric layers that define devices. We hope that together these changes help clarify for readers how these devices work and the factors that contribute to process optimization.

- The VT spread reduction is commented in the text, but the mean VT value evolves as well. Was it an objective to reduce the operation voltage of the devices (and in that case could you comment on the motivation)?

As we now discuss in the manuscript, the reduction in fixed charge in dielectric layers contributes to a reduction in both the variation and in the absolute value of operation voltages. For this reason, reducing both the spread and the mean VT values are objectives during process optimization. We note that reducing fixed charge in this way also improves device stability, as we mention in our discussion comparing the barrier-barrier scans between different wafers. The fabrication optimization process therefore aimed to improve all of these indicators: mean VT, VT spread, and device stability.

- There is no description of the device used for the VT measurements. Fig. 2c,f suggest that the extraction is based on an array of 3 quantum dots. Since it is a multi-gate device, the VT extraction procedure is non-standard and should be described, or refer to the Methods section on automated device measurements if it also applies here.

We have now added an additional Methods section titled "Threshold voltage measurements" to include more explanation of how this VT data is extracted and from what kinds of devices. The data is indeed collected following the procedure described in Methods section "Automated device measurements." In the case of older wafers, the data comes from a combination of three- and 12-quantum-dot devices, while for the wafers representing the optimized process, the data all comes from 12-quantum-dot devices. We also now reference the Methods at the point where this data is introduced in the main text.

Line 135-138: "Fig. 2g shows examples of these measurements from each of the three stacks featured in Fig. 2f. The intermediate stacks show significant disorder and/or instability in these measurements. By comparison, the optimized stack shows clean confinement with the barrier gates and stable current throughout the length of the scan."

- Single stability diagrams are shown to illustrate the statistical device behavior from the different stacks. It raises the question of how representative they are. I would suggest adding wafer maps of the barrier-barrier diagrams (similar to the Extended Data Fig. 5 – 7) of wafers from each stack so that the reader can reliably see the effect of the process improvement.

A figure showing wafer maps of barrier-barrier diagrams from the three versions of device fabrication is now included in the Extended Data section.

- In addition to instabilities and disorder, the cross-capacitances are very different from one stack to the other. It seems almost negative on the optimized stack diagram. I doubt this is solely the effect of an improved disorder. Can you comment on this point?

We observe that the addition of the screening gate layer in the optimized version of devices increases lateral confinement and improves the localization of the quantum dots. This has the effect of reducing cross-capacitance between gates, since quantum dots tend to be more centered under their defining plunger gates and do not extend as much in the direction perpendicular to the channel. Due to the higher charge disorder and lack of screening gates in the earlier devices, there are cases in those measurements where multiple dots form in parallel, which further complicates the barrier-barrier scans and makes them more difficult to interpret using a traditional model for the measurement. By contrast, the barrier-barrier scan data for the optimized stack does become much cleaner and the effect of barrier gates becomes more orthogonal, which we believe explains the stark visual difference that the referee is referring to here.

Line 323-324: "Gate yield is defined as the fraction of gates that can be used to turn on and pinch off their respective current channel."

- What are the current thresholds used to define on and off channels? Subthreshold slopes or pinch-off voltages are not criteria for the gate yield and possible outliers will be discarded only if they compromise the ability to identify a QD tune-up point, correct?

The current threshold used to identify VT is 1 nA. This is described in the "Automated device measurements" section of the Methods and now also referred to in the "Threshold voltage measurements" section that we have added in response to related questions above. The referee's comment is correct—gate yield is simply defined by a gate's ability to turn off current below this threshold of 1 nA. Pinch off voltages themselves are not used to define yield but are used in other performance analyses, such as variation. The gates that perform worse (i.e., have outlier threshold voltage) impact subsequent measurements of quantum dot yield through the failure of those barrier-barrier measurements to yield a viable quantum dot tune-up point.

Line 324-326: "Quantum dot yield is defined as the fraction of quantum dot sites where a viable quantum dot tune-up point can be identified from barrier-barrier scans."

- What is the criteria for viable tune-up point? Is it based on the goodness of fit or gauged by eye? Please provide more details.

We have now added more detail in the “Yield analysis” section of the Methods to describe this. The criteria here is that our least squares fitting procedure, based on a phenomenological 2D function for the “corner point” where current is pinched off by both barrier gates, must converge and return fit values for the barrier voltage coordinates of the corner point. We check by eye to confirm that in the instances where the fit does not converge, there actually is no corner point in the range of the scan, as opposed to an error occurring in the fit procedure. We find that when this fit converges, then the quantum dot site works well for subsequent charge sensing measurements.

Line 173-176: "To estimate the random variation in V_T across gates and devices, we follow a standard CMOS industry method of analyzing matched pair V_T differences [19], calculated between mirror-symmetric pairs of gates. We subtract the mean from each gate-pair distribution to center them at zero and merge them into one distribution."

- From my understanding, each gate-pair distribution should be centered at zero due to the mirror-symmetry. Are the distribution means significantly away from zero? Unless there is a physical explanation for non-centered distributions, it seems to me that subtracting the means to center the distributions is inappropriate, since it artificially erases some statistic fluctuations, resulting in an over-optimistic matching distribution.

We agree that when mirror-symmetric pairs are chosen for this analysis, the mean of each gate-pair distribution should be zero, in which case subtracting the residual mean from each gate-pair distribution is at best superfluous and could slightly reduce the resulting variation. This was not our intention, and we have now re-done all our calculations of matched pair standard deviations without this mean subtraction. The data in Fig. 3d and Fig. 4c-d have been updated accordingly along with the text reporting the extracted variation numbers. We do find that when we remove the mean subtraction step, the resulting variation is higher by a few percent. All our conclusions remain unchanged. We also avoid performing any mean subtraction in the new measurements presented in Supplementary Information section II characterizing the effect of tuning the barrier voltages.

Line 192-200: "Changes in electron number are detected by modulating the voltage on an exterior screening gate and using lock-in detection of the charge sensor current at that frequency. A typical measurement is shown in Fig. 4a. In this 2D sweep, the horizontal axis is plunger voltage, and the vertical axis is the voltage of both barrier gates [36]. The sweep range is chosen to take each quantum dot from zero-electron to several-electron occupation along the plunger axis and from low tunnel rate ($\ll 1$ kHz) to high tunnel rate ($\gg 1$ GHz) along the barrier axis. Transition lines disappear at the bottom of the scan window where tunnel rate falls below the lock-in frequency (~ 1 kHz) and at the top of the scan window where the lines become broadened by tunnel coupling energy."

- Does the screening gate refers to the central screening gate separating the QDs from the sensors, or a gate from the first layer?

This refers to a screening gate in the first layer, on the qubit side, which is chosen to avoid directly modulating the charge sensors during the lock-in measurement. We have now added a line to this section explaining this more clearly and referring also to the 3D device schematic in Fig. 2a.

- On Extended Data Fig. 5, it seems that latching effects frequently occur (I suggest using vector rendering to avoid pixelation). It seems less frequent on Extended Data Fig. 6. Is it an effect of the larger SiGe barrier?

We thank the referee for this question and suggestion. We acknowledge the charge sensing color plots in the Extended Data have been down-sampled to avoid the file sizes becoming too large. We will work with the editor to determine the best way to display these figures in way that is compatible with manuscript size limitations.

Our interpretation of the latching effects in these scans is that the physical mechanism is likely always present in the scan window (i.e., transitions persist at low tunnel rate resulting in poorly defined transition lines), but whether or not they are visible in the scan depends on details of the charge sensor tuning. In these measurements, the appearance of latched transitions represents the transition occurring at too low of a rate to be on resonance with the lock-in tone, but a high enough rate to still shift the charge sensor background on the timescale of the measurement. Whether or not that shift in charge sensor background is visible is a function of how strongly varying the charge sensor background is in the region of the transition. When the charge sensor background is strongly varying, the latched transitions are clear, whereas when the background is smoother, the latched transitions do not show up. While we have not systematically compared the frequency of these effects between the two SiGe barrier thicknesses, we have observed that, at $T=1.6$ K, the charge sensor peaks tend to be more clearly defined in devices with 30 nm SiGe barrier. (We attribute this to higher cross-capacitance in devices with 50 nm SiGe barrier leading to source/drain tunnel rate and electron number being more difficult to tune independently.) This suggests that the charge sensing scans may tend to have a more strongly varying background for devices with 30 nm SiGe barrier, resulting in a higher frequency of observable latching effects.

Line 231-237: "To analyze the variation in 1e transition voltage data, we repeat the same matched pair voltage difference analysis as above, taking differences between 1e voltages for mirrored pairs of plunger gates. The resulting distributions of voltage differences are shown in Fig. 4c-d for two wafers. The random variation in 1e voltage extracted from wafers with a 30 nm and 50 nm SiGe barrier are 59 mV and 60 mV, respectively. Both of these values closely agree with the random variation in gate V_T , meaning the random variation of a transistor-like metric (gate V_T) is matched by the random variation of a quantum metric (1e voltage). This implies that these devices are not subject to significantly increased disorder at the single electron regime compared with the many electron regime."

- On Fig. 4a, the cross capacitance between the dual barriers and the plunger are relatively the same (the charge degeneracy lines are almost diagonal). It means that both sets of gates have a comparable impact on the QD chemical potential and the position of the 1e voltage.

We agree that for devices with 50 nm SiGe barrier, cross-capacitance between neighboring gates is fairly high (~55%), resulting in both barrier gates together having a combined cross capacitance that is >~100% with respect to the plunger gate. As discussed below, we agree that such high cross-capacitance motivates a careful examination of the impact of tuning the barrier voltages on the resulting 1e plunger voltage variation.

V_T is extracted from a pinch-off curve with a voltage compensation applied to the other gates so that their transconductance remains within a given range (please specify the thresholds).

These thresholds are calculated relative to the absolute value of current in the device. We now specify these thresholds in the "Automated device measurements" section of the Methods. This has the effect of roughly setting all gate voltages to equivalent points on their individual pinch-off curves.

The 1e voltage is extracted after a finer barrier voltage compensation (corner point fitting on a barrier-barrier scan). Therefore, I do not think that we can make a direct comparison between both

distributions. The 1e voltage distribution could be intrinsically larger but effectively reduced through the finer compensation on the barrier gates.

It would be very interesting to compare distributions with and without refined adjustment of the barriers to quantify its influence (either set to their value after recursive transconductance adjustment or even with both barriers set to a common gate voltage). How is the QD yield affected if the barrier-barrier fitting step is skipped? Is the experiment feasible?

We thank the referee for these comments and suggestions. To help understand this effect better, we have now performed the experiment suggested by the referee and included the results in Supplementary Information section II. Using a new wafer with a 50 nm SiGe barrier, we measure 1e voltage statistics using our standard procedure with barrier voltages fine-tuned and an alternate procedure with barrier voltages held fixed. Initially, we were unsure whether we would be able to obtain a significant quantity of 1e voltage data without fine tuning the barriers. While using this method did reduce the number of matched pair 1e voltage data for analysis by ~20%, we still obtained a sizeable enough dataset with both methods to do a meaningful statistical comparison. As detailed in the new Supplementary section, the standard deviation of the raw 1e voltage distribution does decrease in response to tuning the barrier voltages, following the referee's suggestion. However, the standard deviation of the distribution of matched pair 1e voltage differences actually increases in response to tuning the barrier voltages. This confirms that for the reports of voltage variation highlighted in the main text, which focus on the matched pair voltage difference distributions, tuning barrier voltages does not result in the under-reporting of voltage variation.

While this result might seem counter-intuitive on its face, we find that it agrees with analytical expectation. To increase understanding of the mechanisms behind this, we carry out new analysis of the impact of tuning barrier gates on resulting metrics of plunger voltage variation. A full discussion of this analysis is also included in the new section of the Supplementary Information (section II). To summarize here: we find that tuning the barrier voltages (in the presence of cross-capacitance) can increase or decrease the standard deviation of plunger voltages depending on the magnitude of correlation between gate voltages within the same device. For the case of the matched pair analysis, where such device-level correlation effects are factored out to reveal the variation within the length scale of a single device, tuning the barrier voltages strictly increases the plunger voltage variation, as a result of uncorrelated variation in barrier voltage offsets adding (in quadrature) to the variation in plunger voltage through cross-capacitance. We find quantitative agreement between the analytical calculations and the measurements of variation obtained from the new experiment. As we also discuss in the new section, we believe fine-tuning the barrier voltages can still be a net benefit for automated measurements as it allows for a more complete sampling of 1e voltages from a given set of devices. With an improved understanding of the impacts of cross-capacitance on this method, we believe the use of this method is fully justified. We also hope that our work here will increase understanding of the complexities that emerge in measuring voltage variation in a many-gated quantum dot devices, as well as make for easier comparison of voltage variation measured by different research groups, which could involve very different devices and/or different tuning methods.

Line 239-242: "Also, while the 1e voltage variation is nearly the same between the two wafers, the variation in chemical potential is better reflected by the ratios between 1e voltage variation and 1e-2e addition voltage (Fig. 4e-f). These ratios are 1.0 ± 0.1 and 0.76 ± 0.08 for the 30 nm and 50 nm barrier wafer, respectively. The observation that the wafer with a deeper quantum well has a reduced ratio of this kind suggests that the 1e voltage variation is dominated by sources in the gate stack above the heterostructure."

- I expect a lower gate lever arm for a 50 nm SiGe barrier (compared to 30 nm). Assuming similar charging energies between the two splits, a difference in lever arm translates in a shift of the mean value of the addition energy (as can be seen on Fig. 4e-f) but also in a proportional increase in the standard deviation (that I do not see). I would also expect a similar lever arm dependence on the matching analysis (Fig. 4c-d), which is not observed. Is it another way to infer that the impact of disorder above the heterostructure surface is reduced in the 50 nm SiGe case?

We agree that the relative standard deviation of addition voltage is less for the case of the 50 nm SiGe barrier compared to the 30 nm SiGe barrier case (~10% compared to ~12% respectively). While the difference is somewhat small, this could indeed be an indicator of less disorder in the charge transitions of the wafer with 50 nm SiGe barrier. This would be consistent with our observations that samples with 50 nm SiGe barrier tend to have fewer instances of disordered transitions (or “spurious dots”) than samples with 30 nm SiGe barrier. We also agree that the lack of a significant change in matched pair 1e voltage variation between the 30 nm and 50 nm SiGe wafers, given the change in lever arm, is an indicator of reduced effective variation in the wafer with deeper quantum well. Elucidating this point is one of the reasons why we subsequently take the ratio between this variation and the 1e-2e addition voltage for both wafers, as discussed below.

I understand the ratio between the matched pair standard deviation and the addition voltage as a renormalization by the lever arm to obtain an estimate of the chemical potential distribution. Again the underlying hypothesis is that the average addition energy is indeed the same in both configuration. However with gates further away from the Si well (and same gate pitch), I expect a shallower potential profile that may result in different charging energies.

To support the assumption that charging energies are similar in both cases, were you able to deduce it e.g. from Coulomb diamonds measurements? Could you use a more precise energy-voltage conversion in the few electron regime, for instance using PAT, or is it incompatible with the CWP setup?

We agree that if we wanted to compare variation in units of chemical potential energy, we would need to know the lever arm or, equivalently, the addition voltage in combination with the addition energy. However, our approach here is to compare the 1e voltage variation to the change in voltage needed to perturb the device from its 1e state. Therefore, the ratio that we report is simply the ratio of 1e voltage variation over the 1e-2e addition voltage. Knowledge of the lever arm and the charging energy are not necessary to make this conversion, since both quantities are measured in units of voltage. The resulting ratio can be thought of as the fraction of an electron number change represented by the voltage variation. To better emphasize this, we have removed the reference to chemical potential in the text and now report this ratio in units of electron number (e). We believe this is a useful way of reporting voltage variation, especially when assessing compatibility with voltage sharing protocols, whose basic requirements would include setting multiple quantum dots to the same charge state using a common voltage.

Line 414-418: "We note that the data used in this analysis comes from measurements of quantum dots tuned one at a time and that this method does not take into account the individualized setpoints of other gates in the array during measurements. Nevertheless, we believe it gives a reasonable estimate of the success rate of using shared voltages across a device to set a common charge state."

- Following a previous comment, I believe that the individual compensation on the barrier gates tends to narrow the distribution, resulting in an over-estimation of the percentage of QDs tunable to the 1e regime with a common voltage. Since this metrics is one of the novelty of the paper, I think it would be interesting to discuss the best extraction protocol and the associated uncertainties more extensively.

As discussed above, we do not find that tuning the barrier voltages has the effect of decreasing measurements of variation when the matched pair 1e voltage difference method is used. Based on our findings on this matter presented in the Supplementary Information, we expect that tuning the barrier voltages will not under-estimate any metric of variation when device-level correlations are factored out (as they are in the matched pair case). Our method of estimating the proportion of quantum dots within a device that could be tuned to 1e with a common voltage has the effect of comparing individual voltage offset to a common device level voltage, therefore it is also a measure of differential voltage variation within individual devices and should factor out the impact of device-level correlations. We have now included a discussion of this to the Methods section “Voltage sharing analysis” and refer from there to the findings presented in the Supplementary Information.

Referee #3 (Remarks to the Author):

This paper describes the operation of a 300mm cryogenic probe station for quantum dot devices and shows the level of uniformity that can be achieved with industrial fabrication. With the increased attentions on industrial integration of spin qubit devices, this result is very timely. The quantum dot performance has been presented in a statistical relevant manner, and this is important to understand where the current device stands for upscaling considerations. The wafer-scale charge sensing down to the last electron is indeed impressive.

We thank the referee for their careful reading of our manuscript and for these supportive comments. Below we give our full response to the referee’s comments and questions and discuss our modifications to the manuscript.

While the collection of cryogenic probe station results is an important step forward for industrial fabrication of better spin qubits, I am not convinced that the degree of advance, its interests to general readers, and the scientific insight for the research community meet the merits of publication in Nature. To be more specific:

1. Where to place this work on the roadmap towards the ambitious goal of a large scale spin qubit array? The works in Ref. 6-9, 12, 13, 14, 15, 16 showed proof-of-principle building blocks for a large-scale array, and the work in Ref. 17-18 explain qubit requirements for practical applications. It is hard for me to find a fit of this work as a novel achievement. Large-scale characterization is an important element in the industrial fabrication learning cycle, but by itself, readers cannot picture the advancement of spin qubit with charge only measurements, nor how to move forward with 1D design. The process variation can be a showcase of industrial fabrication, but qubits with industrial fab has been demonstrated. Studies on variations have also been reported before on both lab and fab devices (Ref. 21, Contamin et al IEDM2022 pp. 22-1, Elsayed et al arXiv:2212.06464, Bavdaz et al arXiv:2202.04482). The author also noticed that their reported uniformity is not enough for share control schemes (line 257, page 13). Moreover, not all spin qubit architecture requires voltage-sharing (e.g. Ref. 39).

We thank the referee for these questions and comments. On the path towards a large-scale spin qubit array, we argue that this work represents a key step. While the proof-of-principle works are both necessary and impressive, significant advances in scale will require an industry compatible approach to fabrication, which is why many of these works tout their platform’s compatibility with CMOS manufacturing as an indicator of its potential. Indeed, spin qubit devices have been made using CMOS manufacturing techniques, but these have so far been confined to Si-MOS platforms, where we argue

multi-qubit operation is currently hindered by disorder. This point is supported by a lack of reports of extensive coherent control across quantum dot arrays in Si-MOS comparable to those demonstrated in epitaxially grown heterostructures [Hendrickx et al. 2021, Philips et al. 2022, Weinstein et al. 2023]. The combination in the present work of a low disorder heterostructure stack (Si/SiGe) with a CMOS compatible process is therefore an important breakthrough on the road to achieving a scalable multi-qubit processor. We also note that the devices studied here are leading in size and complexity for spin qubit devices, containing arrays of up to 12 spin qubit sites with four companion charge sensor quantum dots. We believe that the increase in device size and complexity here, combined with demonstrations of high yield and reproducibility, represent significant novelty and suggest that this fabrication method can be extended to much larger array sizes.

The yield and reproducibility demonstrated in this work is a crucial point. Despite the impressiveness of today's best proof-of-principle qubit results, such results lack a corresponding report of device yield. While some papers discuss the challenges explicitly [Brauns et al. 2018, Dodson et al. 2020, Tahan 2021], we have not found a comparable dataset of spin qubit device yield to which to compare our work. We hope this work might set a standard for reporting such yield data. In the absence of widespread published data today, the authors here refer to their experience in the spin qubit field to say that many of the best spin qubit results occur after a lengthy screening process to search for a fully functional device. As spin qubit devices become larger, such a process will become far more difficult as the number of individual components that must function within a device increases. For this reason, improvements in yield through industrial fabrication, as well as the large-scale characterization needed to achieve such yield, are necessary steps towards achieving large-scale spin qubit devices. Furthermore, as spin qubit devices become larger and more complex, the marginal benefit of finding the "leading edge" devices on a wafer to package in a quantum computing stack will also become greater, implying the continued importance of wafer-scale cryogenic characterization even after a particular fabrication process has been optimized. To emphasize these arguments further to the reader, we have added new text to the introduction and conclusion of the manuscript.

We argue further that, in addition to yield concerns, the "charge only" measurements of electrostatic variation presented here are central to the challenges of spin qubit scaling. Despite these qubits being encoded through the spin degree of freedom, charge disorder is a leading cause of variation in these qubits, including the excited state energies of single spin qubits (set by a valley-orbit interaction in the case of Si/SiGe) and the exchange tunability between spins in neighboring quantum dots. Achieving low levels of electrostatic (voltage) variation will therefore be necessary to achieve any truly large-scale spin qubit array, and the characterization techniques shown here will be a vital part of that.

We do admit that, while the focus of this paper is on electrostatic characterization, the lack of qubit data in the manuscript could leave the question open for readers whether the devices discussed really represent an improvement over previously published devices. To address this, we have added a new figure to the Extended Data section showing measurements of Loss-DiVincenzo spin qubits on multiple wafers, including a case where the entire 12-qubit linear array in a single device is fully tuned up. In total, we now show data from 39 different qubits compared to three shown in an earlier work [Zwerver et al. 2022]. Furthermore, we also demonstrate single qubit gate errors more than an order of magnitude lower than in [Zwerver et al. 2022] and now in-line with state-of-the-art gate fidelities for Si/SiGe Loss-DiVincenzo spin qubits. This results in a gate fidelity that is by far the highest reported for a device made with CMOS industry manufacturing. We believe this will help clarify to the reader that this platform really is compatible with high performing spin qubits and, in combination with the improvements in yield and electrostatic reliability already demonstrated, represents a genuine leap forward in the scalability of semiconductor spin qubits. We believe this figure, as well as new text added

to the conclusion of the manuscript, will help clarify the important link from advancements in yield and variation to continued scaling of the spin qubit technology.

We also acknowledge that this work follows other publications presenting measurement and analysis of device variation, however we argue this work takes such analysis significantly further. This is made possible through the increased size of the charge sensing datasets presented here, including hundreds of spin qubit arrays (as now emphasized in the abstract) and thousands of quantum dots. This increase in measurement scale is achieved through the unique combination of the cryo-prober measurement system, the automated measurement and analysis procedure, the increased size of the 12-quantum-dot arrays, and the high yield and low disorder of the Si/SiGe devices. In addition to the matched pair 1e voltage variation analysis presented in the first version of the manuscript, we also now include in this revision: the analysis of correlation between 1e voltages of all plunger gate pairs in the 12 quantum dot devices; analysis of correlation between 1e voltages and threshold voltages (showing a high degree of correlation); and an analytical treatment of the effects of cross-capacitance and barrier voltage tuning on metrics of 1e voltage variation.

Finally, we thank the referee for pointing out that, in the discussion of voltage sharing, the citation of [Veldhorst et al. 2017] does not fit in with the other works cited, so we have removed that citation in this revision.

2. How relevant is this study to spin qubits? The high yield of wafer-scale single electron regime does not necessarily promise high spin qubit yield. Moreover, the devices are quantum dot arrays, not full spin qubit devices. As authors pointed out in line 149 page 8, there is no micromagnet for spin control. Is micromagnet compatible with your industrial fabrication, and will placing the micromagnet degrade the quantum dot array? The devices can be, in principle, operated as exchange-only qubits. However, exchange-only qubits could suffer from the Meissner screening effects of the superconducting material in the gate stack (Ref. 9), which is presented in this work to reduce the line resistance (line 109, page 5).

We thank the referee for these questions. As discussed above, we have now introduced a figure to the Extended Data showing the results of spin qubit characterization on a variety of devices and wafers. We hope this clarifies that, while the focus of this work is on quantum dot electrostatics characterization, these devices perform well as spin qubits. While we have not quantified spin qubit yield, we observe that the high component yield (e.g., gates, quantum dots) presented in the manuscript has enabled a high success rate for obtaining spin qubit results from each device cooldown in a dilution refrigerator. We now emphasize this in the concluding paragraph of the revised manuscript.

While the micromagnet layer is not included in the 300 nm process characterized in this work, the path to doing so is well understood. It has remained an optional process in our fabrication flow in order to support multiple qubit encodings, including both Loss-DiVincenzo qubits and exchange-only qubits. We have added more description of the micromagnet fabrication process to the Methods section “Micromagnet design and EDSR.” We also see no evidence of degradation after micromagnet deposition, as demonstrated by the high performance of Loss-DiVincenzo spin qubits showcased in the new Extended Data Fig. 1.

We also agree with the points raised about the superconducting material and its possibly detrimental effect on exchange-only qubits. We have now added more discussion about the advantages and drawbacks of using superconducting materials in our metal layers to Supplementary Information section I. We also now clarify in the Methods section “Gate line resistance measurements” that we present two superconducting layer processes for the gate and interconnect layers, respectively. While the greater DC resistance improvement comes from making the gate layer superconducting, there are still benefits to making the interconnect layer superconducting, which would avoid any Meissner screening in the vicinity of the qubits. We also note that the data and discussion of gate line resistance

test structures has now been moved to the Extended Data and Methods section “Gate line resistance measurements,” respectively, in order to make space to address other comments from all referees in the main text.

3. Can you give more insight on optimization in the fabrication and device layout? As authors write in the paper, one of the two advancement of this work is “we optimize an industry-compatible process to fabricate spin qubit devices on Si/SiGe heterostructures, combining low process variation with a low disorder host material” (line 62, page 4). However, the discussion on the optimization is really limited. I believe more discussion on the fabrication optimization and how the critical steps are reflected on a wafer-scale measurement will be very beneficial to the spin qubit community and increase the impact of the paper. Additionally, the gate layout of the quantum dots is not show (Fig. 3b is the upper fanout of the gate, rather than the quantum dot defining region). If the authors want this study to “set a new standard for what can be achieved with spin qubit devices today and pave the way for significantly larger and more complex spin qubit arrays of the future”, more detail about gate layout and geometry should be discussed, so that readers can have a starting point to contribute to the design of a large spin qubit array in an industrial compatible manner.

We accept the referee’s criticism (shared by other referees) that the manuscript could use further description of the methods used to improve device quality in order to be clearer to the reader. We are limited to some extent based on restrictions from our institution, but we have made changes in the revised manuscript that we think will help address this criticism. We now discuss the role that fixed charge in our dielectric stack plays in the observed threshold voltages and their cross-wafer variation, as well as the strategies we have used to reduce fixed charge. We have also added a new figure to the Extended Data section showing maps of barrier-barrier scans taken across wafers made with each version of fabrication, to better illustrate the improved reliability of quantum confinement after the fabrication optimization. While the top-down image in Fig. 3b does in fact show the gate patterning, we acknowledge it does not show the underlying structure (including a screening gate layer) that provides lateral confinement. To address this, we now include a 3D schematic in Fig. 2 showing more clearly the structure of the gate and dielectric layers that define the quantum dots. We hope that, with these additions, readers are better able to visualize the full structure of the active device area and can better understand how these results are achieved.

Some further comments about the paper:

1) Line 6, page 2. “Spin qubits [1, 4, 5] have shown impressive control fidelities [6–9] but have historically been challenged by yield and process variation.” The yield and process variation are important problems this paper tries to solve. Good to back it up with references or more clarifications.

The manuscript has been updated to include citations on this matter as well as more discussion of how the spin qubit field is impacted by yield today and the benefits that come with improving it.

2) Line 13, page 2. “...this fabrication process...” Not clear to me what “this” is referring to. I assume you mean the final optimized process.

We have clarified the wording in the abstract accordingly.

3) Line 15, page 2. “...to the fabrication and measurement of spin qubits.” It would be helpful to clarify that the study focuses on quantum dots rather than full spin qubits.

We have changed the wording here from “spin qubits” to “spin qubit devices.” We argue that for the abstract, this terminology is preferable to “quantum dots” since the latter can refer to devices with a variety of different applications. Although the cryo-prober measurements focus on the electrostatics of spin qubit devices, we hope that the coherence data included in the Extended Data section of the revised manuscript helps clarify that these devices’ ultimate application is to encode spin qubits.

4) Line 40, page 3. One of the biggest challenges of SiGe is the valley splitting. Good to also include that in the discussion for completeness.

We have added a mention of valley splitting challenges (with reference) to the introduction.

5) Line 54, page 3. Is “full wafer probing” really necessary for spin qubits? Should all qubits in the 300mm wafer be uniform for practical operation considerations? The size of a large-scale spin qubit array would be in the mm to cm scale, which is way smaller than the 300mm wafer. I would think qubits need to be uniform within the array area, rather than the 300mm wafer scale. Other way of saying is that the short-range uniformity could be more relevant.

We agree that uniformity within individual spin qubit arrays is a more important metric of performance than cross-wafer uniformity. This is why in this work we focus on the matched pair analysis method to extract voltage variation metrics, since this method factors out the impact of die-to-die variation and highlights sources of variation within the length scale of a single spin qubit array. Additionally, we argue that having access to a full wafer of devices in the cryo-prober system comes with significant benefits for characterization. Firstly, measurements of variation within individual devices become more confident when data is collected on many devices. Secondly, given that improvements in yield are critical for scaling up the device size, it is important to have yield statistics for every wafer in order to determine how a yield metric changes in response to process changes. Thirdly, even when device yield and variation are highly optimized, when qubit processor size becomes larger, the marginal benefit of finding the best die on a wafer increases, which increases the value of being able to characterize many different devices across a wafer before picking one to package in a quantum computing stack. We have added new text to the introduction and conclusion sections of the manuscript to further emphasize these points.

6) Line 72, page 4. As discussed above, it would be helpful to include a picture on the gate layout.

We have now included a schematic in Fig. 2 that we hope clarifies the layout for the reader.

7) Line 100, page 5. Technically speaking, the structure shown in Fig. 2b is not Hall bar. There is also no Hall effects in the measurements. The carrier number is determined from an estimated gate capacitance. Out of curiosity: can you perform CV measurements to extract the capacitance as well as the threshold voltage? If you do, will the signal be limited by the area of the gate? An additional remark: from this discussion you can see the importance of showing device layouts and geometries to readers for easier access of your work.

We thank the referee for these comments. We want to note that the data and discussion of Hall bar structures has now been moved to the Extended Data and Methods section “Carrier mobility measurements,” respectively, in order to make space in the main text. The original Hall bar schematic included only the components used to carry out the mobility estimation procedure in the cryo-prober.

We realize now that leaving some components out (i.e., ohmic contacts) can be confusing. We have now adjusted the Hall bar schematic (now in Extended Data Fig. 3) to show all ohmic contacts, including those used to measure transverse resistance in the conventional Hall measurements performed in a magnetic field.

We can indeed perform CV measurements on these Hall bar structures. The gate area is large enough to obtain a measurable signal, which could in principle give a more accurate measurement of the gate capacitance. We note that the extra ohmic contacts in these devices do introduce parasitic capacitance in the device “on” state that can be difficult to factor out accurately. After correcting for this, we still expect the largest source of uncertainty in this estimation to be the uncalibrated threshold carrier density at which the device first turns on, which can only be accurately measured with conventional Hall measurements in a magnetic field. We also note that our estimation of gate capacitance has worked well mainly due to these measurements focusing on wafers with nominally the same gate dielectric stack. By contrast, future work exploring mobility measurements across variable gate dielectric stacks could be aided by incorporating CV measurements to control for variation in expected gate capacitance.

8) Line 134, page 7. I would suggest rewriting for clarification. An example could be “The diagonal lines at the onset of the current is a signature of a single quantum dot between the barrier gates, where Coulomb blockade dominates the transport between source and drain”.

We thank the referee for this suggestion. We have now modified this line in the manuscript for clarity.

9) Line 175, page 9. The match-pair method in standard CMOS uses nearest neighbour pairs. The work here uses mirror-symmetric pairs. This new analysis method need justification for the relevance of qubit operations. My intuitional feeling is that the bare standard deviation of each gate could be more relevant for the shared-control scheme. It can also give insight on if the variation is gate position dependent.

We have now added more discussion of our reasoning to the sections on the matched pair analysis of VT data and of V1e data. Our modification of the method to use mirror-symmetric pairs is based on the more complex gate geometry of spin qubit devices. In this environment, nearest neighbor gates, while nominally identical, are subject to slightly different cross-capacitances from nearby charge sensors and reservoirs. Use of mirror-symmetric pairs avoids any systematic voltage offsets between gates due to these geometric differences, leaving only random voltage variation in the resulting extraction. Furthermore, we argue that for characterizing variation within individual devices, the matched pair standard deviation is more relevant than the bare standard deviation, due to the latter metric also incorporating die-to-die variation from systematic differences in processing across different locations on the wafer. By factoring out these systematic cross-wafer effects, the matched pair standard deviation gives a more accurate measurement of variation within an individual device, therefore making it better suited to benchmarking devices’ applicability for voltage-sharing schemes. (In principle, such schemes do not require the same voltages to be applied across an entire wafer and could allow for shared voltage values to be customized for each die.) We also agree it can be useful to check whether specific gates in an array are subject to more variation than other gates. This is still possible when analyzing matched pair voltage differences, since the distributions of each gate pair can be individually considered. Excessive variation on any one gate would then show up in the associated gate pair distribution. While we do observe fluctuations in the standard deviations of different gate pair distributions, we have not observed individual gates in the array having higher variation systematically across multiple wafers.

10) Line 192, page 10. What is the modulation energy? If it is too high, higher valley or orbital state could come into the pulsing window and affect the tunnelling rate extraction. I would also suggest more clarification or a reference on the charge sensing. It is not clear why electron number can be detected now.

We thank the referee for this question. We now specify the modulation voltage (3 mV RMS) when discussing the lock-in procedure in the Methods section “Automated device measurements.” We agree that this modulation voltage is likely larger (in units of chemical potential energy) than a typical valley splitting, which could have implications for tunnel rate extraction. However, our intention is not to extract accurate tunnel rates in this work. The regions of extreme tunnel rates shown on the example charge sensing plot in Fig. 4a are only intended to orient the reader in the different regions of the scan window. Our quantitative analysis of the scans focuses mainly on the location in plunger voltage of each $1e$ transition line. While the applied modulation voltage may broaden the transition lines somewhat beyond the linewidth set by the temperature, the magnitude of this broadening (~ 3 mV) is still much smaller than the magnitude of transition voltage variation we report (~ 60 mV), so we conclude that it does not significantly affect the results.

We have also moved up our reference to [Borselli et al. 2015] to the sentence where we first discuss our detection of electron number. This was referenced later in the paragraph, but because our charge sensing method follows this work very closely, we hope that inclusion of this citation up-front will provide the reader with an understanding of how our measurement of electron number is achieved.

11) Line 199, page 10. When the transition lines disappear, the tunnel rate is the lockin frequency or 2x the lockin frequency? In a lockin cycle, there should be a tunnel in and a tunnel out even.

We thank the referee for pointing this out. We have corrected this line in the manuscript to read “two times the lock-in frequency.”

12) Line 201, page 10. For charge sensing, one 12QD devices are measured per die. For gate V_T , four 12QD are measured per die (line 155). The reduced device for charge sensing is only measurement time concerns, or there are screening already based on gate V_T ?

We do not use the transport measurements (i.e., V_T data, barrier-barrier scans) to screen devices before taking charge sensing measurements. The number of devices sampled in the charge sensing dataset (one 12QD device per die) is more reflective of a typical dataset size we collect, based on a balance between collecting statistics and limiting the measurement runtime, as the referee suggests. The larger sample size (by 4x) of devices presented in the transport section of the paper is intended to generate greater statistics for our measurements of yield, particularly to back up the high yield percentage of device components that we report.

13) Line 204, page 10. What kind of “noise” are you referring to? Lockin based charge sensing should be very immune to noise, unless its device instability, which I would not call it noise.

Our intention here is to say that the charge sensing signal must be high enough relative to background noise such that it can be resolved in analysis. We have modified our language to clarify this. We agree that using lock-in amplification helps significantly here. However, in cases where the sensor dot and sensed dot are not well coupled, such as cases where dots form in unintended locations, the sensing signal may drop considerably such that, even when using a lock-in technique, the scan signal is too weak

relative to background noise to enable a robust automated analysis procedure. The high success rate that we observe in charge sensing measurements is another confirmation that dot location, and therefore charge sensor coupling, is highly reliable in these devices.

14) Line 208, page 10. It is not clear to me that lower charge noise would help charge sensing, unless the charge noise is really bad. It is more likely that at lower electron temperatures, the Coulomb peak will be narrower due to reduced thermal broadening, which gives larger slopes and increased the sensitivity.

We agree that lower electron temperature will bring the benefit of increased charge sensor sensitivity. Our comment about charge noise refers to fluctuations in charge sensor peak position over the relatively long timescale of each charge sensing measurement (several minutes). These fluctuations could manifest as a drop in charge sensor signal or a complete loss of signal when the scan is partially complete. In reference to the referee's previous comment, we argue that this instability belongs in the same category as conventional quantum dot charge noise. In particular, the phenomenon of Coulomb peak drift is an accepted method of characterizing low frequency charge noise [Zimmerman et al. 2014] and has been shown to agree with other measurement methods over frequencies below 1 Hz [Kranz et al. 2020]. We have added a reference to [Zimmerman et al. 2014] to hopefully clarify the connection we are making between these different manifestations of charge noise. Lastly, we note that this type of device instability is commonly found to "settle" and therefore improve as a device is measured over the timescale of hours and days, and because these automated charge sensing measurements occur within an hour after devices are first turned on, they are likely subject to the worst case scenario for this type of instability.

15) Line 216, page 11. I think it would be clearer to the reader to say that the 1e transition voltage is measured at a fixed barrier gate voltage (of 0.7V?)

We have added more explanation to the Methods section "Charge sensing transition curve analysis" to clarify that, since the charge sensing scan range is individualized for each barrier gate voltage, the extraction of the 1e transition voltage at the midpoint of the barrier voltage axis results in this extraction occurring at individually tuned barrier voltages rather than a fixed barrier voltage. We also hope that the new Methods section "Impact of tuning the barrier voltages on 1e voltage variation," along with the related Supplementary Information section II, help to emphasize for the reader that the standard procedure here is to fine tune these barrier voltage values, while an alternate method using fixed barrier voltages is also possible.

16) Line 235-239, page 12. It is not clear to me that the similarity in variation of gate threshold and the 1e transition could justify they are the same disorder limited. Can you make the scattered plot of the gate threshold vs the 1e transition of the same gate for different dots, and see if there is a correlation? Additionally, it would be good to add a reference to Contamin et al IEDM2022 pp. 22-1, where similar studies have been performed.

We have now generated the suggested scatter plots of threshold voltage versus 1e voltage and calculated the correlation coefficients for both wafers (with 30 nm and 50 nm SiGe barrier, respectively). On both wafers, we observe strong correlation (correlation coefficient > 0.9) between these voltage metrics, which should further justify our interpretation that voltage variation in both the 1-electron and many-electron regimes are limited by the same source(s) of disorder. We also thank the referee for pointing out the paper by Contamin, et al., and have now updated the manuscript to cite this work when we begin our discussion of 1e voltage analysis.

17) Line 240, page 12. It would be good to clarify why the ratio matters. Essentially, you are comparing the variation to the charging energy for share control.

We have now modified the manuscript to clarify that our goal here is to convert measurements of voltage variation into units of electron number variation to obtain a more relevant benchmark for voltage sharing applications.

18) Line 243-245, page 12. The discussion and reasoning are nice. I think more should be presented in the paper. Out of curiosity: do you have separate magnetic field based hall mobility analysis to compare the heterostructure of different interface depth?

We thank the referee for this comment. We hope that our added discussion of fixed charge in the revised manuscript helps to provide further context for this discussion on sources of voltage variation. We also now include in the Extended Data measurements of mobility for both quantum well depths (30 nm and 50 nm), showing reductions in remote scattering for the samples with a deeper quantum well. We have added this to our discussion on voltage variation to make clear that further reductions in charge scattering centers in our dielectric layers could enable further reductions in 1e voltage variation.

19) Line 318, page 15. Are you referring to threading dislocations? With the hallbar area and the reduced mobility percentage, you can estimate a defect density (assuming 1 per low mobility hallbar). Will that agree with the TDD of the SiGe wafer?

We thank the referee for this question. We have now expanded our discussion of these defects in the Methods section "Carrier mobility measurements." Specifically, we refer here to pile-ups of misfit dislocations, which we have observed in our heterostructure metrology. We also now include a discussion of the probabilistic comparison suggested by the referee, finding that the density of these dislocations pile-ups reasonably agrees with the frequency of observed mobility degradation.

20) Line 349, page 16. You define V_T with a threshold current method. In many CMOS studies, V_T is defined as the maximum transconductance voltage. Can you comment on why you use this V_T definition?

We have now added more context to the Methods section "Automated device measurements," specifying that we are using the constant-current method of threshold voltage extraction [Ortiz-Conde et al. 2002]. While this method may not be the most common across CMOS studies, we have found that the simplicity of this method makes it well suited towards robust automation. Earlier versions of our devices sometimes showed disordered pinch-off curves containing multiple points of high transconductance. Using the constant-current method to extract V_T consistently gives the "left-most" edge of such disordered features. While the optimized devices that we focus on in this work do not have such disordered pinch-off curves, as we continue to run experiments on materials and processes, it is always possible for such disorder to temporarily re-emerge. Therefore, we have retained this method, since it is most easily standardized across devices with different degrees of disorder, enabling the best comparisons between past, present, and future devices.

Referee #4 (Remarks to the Author):

Paper Ref : 2023-07-11832

Review of “Probing single electrons across 300 mm spin qubit wafers”

Authors : Samuel Neyens*,,, Otto Zietz*, Thomas Watson, Florian Luthi, Aditi Nethwewala, Hubert George, Eric Henry, Andrew Wagner, Mohammad Islam, Ravi Pillarisetty, Roza Kotlyar, Kent Millard, Stefano Pellerano, Nathan Bishop, Stephanie Bojarski, Jeanette Roberts, and James S. Clarke.

Feedbacks:

The authors present an in-depth study on a 300 mm wafer characterization methodology for linear spin devices and multi-qubits (12 QD) at cryogenic temperature (1.6K). The proposed solution is based on an advanced 50nm-100nm gate pitch CMOS process with a way to automate the measurements and a way to optimize the process step to improve QD quality/fidelity. This opens the door to a large-scale quantum dot solution with industrial techniques for massive integration/qualification. As a results, the text and context of the study are well structured with clear argumentation and data.

They present the methods and the results as well as the cryo-probe equipment used for this development based on their own technology with Si/SiGe heterostructure and specific gate stack with/without ²⁸Si isotopic substrate at cryogenic temperature of 1.6K. the Ohmic; The barrier/plunger line structures are characterized with their key parameters and describe the distribution before and after process correction with a standard deviation metric (at least). Then, the 12 QDs with charge sensor are also analyzed with the full distribution measurements along the device topology based on the barrier/plunger biases. Additionally, a Hall bar structure is carried out to evaluate the mobility of the carrier which is an important parameter for QD. The methodology leads to systematic characterization with a wide range of data and KPIs for the process and QD and their quality results provide a 0e/1e/2e stability diagram with detection lock in.

It is an interesting and significant study that reaches today with their industrial approach (according to KPIs measurements): 100% for ohmics and gates, 99.8% for 1 QD and 96% for 12QD. These results are very promising, even if other challenges remain to be overcome.

We thank the referee for their careful reading of our manuscript and for these supportive comments. Below we give our full response to the referee’s comments and questions and discuss our modifications to the manuscript.

To enrich this proposal, please find below some suggestions, comments and open questions:

- for the qubit, specify whether it is physical or logical and explain that it is an electronic qubit at the beginning of the text

We thank the referee for this suggestion. We now specify in the abstract, as well as in the paragraph on fabrication, that our work focuses on qubits defined with electron spin. We also clarify when introducing the 12-quantum-dot (12QD) arrays that we refer to physical qubits, as opposed to logical qubits, when discussing qubit encoding.

- for statistical metrics on Quantum Dot indicate which KPI will be used throughout the study

At this point in the abstract where we reference “statistical metrics,” we now specify, in addition to yield, that we focus on “voltage variation” as a general KPI category throughout the study.

- for process variation, indicate which physical parameters are monitored

To be more precise, we have now replaced the phrase “process variation” with the phrase “voltage variation” in the abstract. We believe this term represents the category of metrics we will cover (including variation in threshold voltage and 1e voltage) while remaining concise enough to fit in the abstract.

- scaling is also driven by the Ion/Ioff KPI or indicate the FOM (lines 30-31)

We thank the referee for this comment. We have modified the sentence discussing Moore’s law scaling to include an explicit reference to Ion/Ioff as well as the gate delay KPI.

- on line 45 indicate the parameters monitored with the statistical measurements

We have now modified this sentence to include the parameters we focus on in the manuscript: voltage variation and yield.

in line 69 a cross section device will be welcome (see fig. 2)

We thank the referee for this suggestion. We now include a more complete 3D schematic of our devices, including two cross-sectional cuts, as part of Fig. 2.

- It is indicated that it takes 2 hours for thermal cooling, what about the other steps?

This 2-hour time frame refers to the transfer of the wafer through the load lock as well as thermal cooling to base temperature. We have modified this sentence in the manuscript to clarify this.

- On line 89: what does “variety of metrics” mean, please list parameters

We have modified this sentence to include a list of device parameters we focus on throughout the study (gate line resistance, ohmic contact resistance, carrier mobility, gate threshold voltage, and transition voltages in the few-electron regime).

- On line 96, the PAD dimension must be indicated with the BEOL type/specification

We have added the pad dimension and pitch to the legend of Fig. 1. While we are unable to share the pad material due to restrictions from our institution, we hope these length scales give the reader an indication of the precision and tolerance of the probe alignment process.

- line 300: For the hall bar, indicate the type of carrier being investigated. What are the results when applying a magnetic field (1T? or with the EDR map)? What about in the 1e regime?

When introducing the measurements of Hall bar structures, we now specify that the carrier mobility we investigate refers to electron carriers. We also want to note that all data and discussion of Hall bar structures has now been moved to the Extended Data and Methods section “Carrier mobility

measurements,” respectively, in order to make space to address other comments from all referees in the main text. While we are unable to apply a magnetic field in the cryo-prober measurements due to hardware limitations, we now include in Extended Data Fig. 3 Hall measurements taken in a conventional cryostat with magnetic field control. This allows us to measure mobility with calibrated carrier density, providing a useful check on the accuracy of mobility estimations made with the cryo-prober as well as revealing the dependence of mobility on carrier density for different heterostructures. We note that all mobility comparisons are made in the many-electron regime. Repeating the statistical characterization of $1e$ voltage variation in a magnetic field would require a significant hardware change to the cryo-prober. We note that in qubit experiments performed in dilution refrigerators with a magnetic field applied in-plane, we do not observe a change of electrostatics, therefore we do not expect our $1e$ voltage data would be significantly affected by the addition of an in-plane magnetic field.

- On line 107, indicate the value of the RF signal applied to the device.

This sentence now includes a specification of the typical frequency range to which we refer (~ 0.1 -20 GHz). We also note that this discussion of gate line resistance test structures has been moved to the Methods section “Gate line resistance measurements” to make space in the main text.

- Indicate the gate stack structures (link to fig.2) with the threshold voltage 3V, 1.8V & 0.8V. What is the gate stack optimization strategy?

We hope the new device image in Fig. 2 helps clarify for the reader the gate stack structure(s) whose threshold voltage data we report. We have also added a deeper discussion of how devices are optimized, including both the reduction in the fixed charge of dielectric stacks and the integration of a designated screening gate layer. The significant reduction in median threshold voltage to which the referee refers is attributed to this reduction in fixed charge.

- What is the specification of the left/right barrier “immunity” and how to choose the right value (in figure 2g the range is 3.5 nA (StackA) for V [3.15 V – 3, 05 V] to 25 nA V [1 V – 1.05 V] (optimized stack)

We have now added more explanation of this measurement procedure to the Methods section titled “Automated device measurements.” Specifically, this measurement accounts for gate-to-gate variation in barrier voltage set points by customizing the voltage ranges of the barrier-barrier scan based on each gate's threshold voltage. This ensures that the “tune-up point” which we seek to identify with this measurement is always roughly centered in the scan window.

- Line 110, the temperature is indicated at 1.6K so what about IP self-heating?

We have now added more discussion about the electron temperature to the “Electron temperature measurement” section in the Methods. Specifically, we attribute the offset between device electron temperature and the temperature of the stage to two possible root causes related to system hardware: insufficient thermalization of the DC wiring and/or thermal resistance between the wafer and the chuck. We expect the effect of self-heating to be non-dominant, as the typical power dissipated by the devices through Joule heating is well below the cooling power of the 1 K cooling unit.

- Line 139, what is the way to optimize the process as the text claims?

We thank the referee for this question. We have now added a deeper discussion of how devices are optimized, including both the reduction in the fixed charge of dielectric stacks and the integration of a designated screening gate layer.

- Line 149 which type of micro magnet is targeted: Type and level in metal stack, magnitude of field B and orientation, Gradient with Map according to confinement.

We note that we have added new sections to the Methods to discuss qubit measurements in support of the new Extended Data Fig. 1. The section “Micromagnet design and EDSR” focuses on micromagnet design, specifying the material (cobalt) and the strength of its magnetic field gradients. The new Extended Data Fig. 1 also includes a SEM image showing the placement of the micromagnet layer at the top of the device stack.

- Please justify the choice of EDSR compared to ESR?

We choose the method of EDSR with micromagnets over ESR in order to engineer qubit addressability through the local magnetic field applied by the micromagnet. We have added an explanation of this to the new Methods section “Micromagnet design and EDSR.”

- On line 166, what is the root cause of the structure's VT boundary shifting (only due to gate accumulation)? How to solve it?

We attribute the shift in VT at the edges of the gate array to the cross-capacitance of the accumulation gates next to the outer barrier gates. The larger cross capacitance experienced by these gates in comparison with the gates inside the array can be thought of as “helping” those gates turn on. Therefore, they exhibit a lower threshold voltage. While we do not view this VT shift as a problem for current applications, future applications that require more uniform voltages could address this by tailoring the design of the accumulation gates where they approach the barrier gates to reduce their cross-capacitance. If this is still insufficient, applications involving voltage sharing could be designed under the assumption that gates neighboring the reservoirs use a different common voltage than that used by gates in the interior.

- Line 180, the standard deviation is improved to 58mV, what is the criterion/value to be considered “correct” ?

At this point, there is no agreed-upon target for how low this variation needs to be, and in general, it will depend upon the intended application of the device. As we discuss in the manuscript, for voltage-sharing applications, this variation metric needs to be significantly reduced. Our goal in reporting this value is to share where we are today as well as to discuss strategies to reduce this variation in future devices.

- Line 192, what is the voltage modulation range?

For the lock-in measurement, we use a voltage modulation of 3 mV. We have updated the Methods section “Automated device measurements” to include this along with other details of the measurement.

- On line 206, what does “as gauged by eye” mean (behind it will be a parameter value!)

Here we refer to an assessment of whether the strength of the transition line signal is large enough to resolve relative to the measurement noise background. Currently, this process of assessment is qualitative and relies on human judgment (i.e., deciding whether or not transition lines are visible in the scan). As we mention in our discussion of the charge sensing success rate, this task of assessing the “quality” of charge sensing measurements could be made more standardized with a machine learning algorithm. This is an area of ongoing/future work for us.

- Explain the “Gamma” factor in the text (see fig. 4)

We have updated the manuscript to clarify that “gamma” here refers to the tunnel rate.

- Line 201, what is the noise immunity for charge sensing to follow the $1e/2e$ transition? And what about for a large linear array?

To maximize our success rate in obtaining $1e/2e$ transition data, a variety of strategies are used. To maximize the charge sensing signal for a given quantum dot, the nearest charge sensor out of the four in the array is always selected. To increase the visibility of charge sensing transitions relative to background noise, a lock-in technique is used with a relatively large modulation amplitude (3 mV). Furthermore, to ensure that the $1e/2e$ transition lines appear in the scan window, the voltage ranges for both axes are calibrated based on earlier measurements (a VT measurement in the case of the plunger gate voltage and a barrier-barrier scan measurement in the case of the barrier gate voltages). We note that these are all discussed at various points in the main text and Methods sections of the manuscript, with some additional details in the revised manuscript. Altogether these strategies culminate in the charge sensing success rate (91%) that is reported in the main text. While this is still below 100%, strategies to improve this success rate are also discussed. We expect all strategies to be extensible to larger arrays, assuming that the charge sensor array is also extended.

- Line 245-250, fixed charge process parameters are discussed with reduced gate oxide thickness lever, what about sidewall (if relevant in this technology)?

We agree that reducing gate dielectric thickness would bring the gates closer, which would bring the gate sidewalls closer to the buried Si quantum well. If those gate sidewalls have significant defects, those would get closer as well. We expect such defects, if they are present, to have a reduced impact compared to those in the gate dielectric or at the interface between gate dielectric and Si/SiGe heterostructure, since those sidewalls are close to the gate metal itself and can be screened by image charges.

- The comparison on two wafers in the $1e$ transition measurement, is it the same batch?

These wafers were fabricated in separate batches, approximately 2 months apart.

- In line 271, the study is about QD, so what are arguments to extend to the Spin qubit?

We thank the referee for this question. In response to this as well as questions and comments from all referees, we have now added a new figure to the Extended Data section highlighting spin qubit results achieved across several wafers using this fabrication process. We have also added new sections to the Methods for details on how these measurements are taken. We believe this will help to clarify to the reader that this platform really is compatible with high performing spin qubits and, in combination with

the improvements in yield and electrostatic reliability already demonstrated, represents a genuine leap forward in the scalability of semiconductor spin qubits. We believe this figure, as well as new text added to the conclusion of the manuscript, will help clarify the important link from advancements in yield and variation to continued scaling up of spin qubit technology.

- Note: The Methods section is well introduced.

We thank the referee for this comment.

- Note line 349, what is the noise immunity strategy for all measurements?

This section of the Methods has now been updated to include more details about how the VT measurements are set up to ensure that the VT value is successfully extracted from the sweep.

- Note: It should be interesting to explain why the study focuses on the electron spin qubit rather than the hole spin qubit.

We have now updated the section where we first introduce the fabricated devices to include a comment about our choice of electron spin. While much progress has been shown recently using hole spins, the long coherence times of electron spins compared to hole spins today make them a versatile platform for encoding different spin qubit types, including Loss-DiVincenzo and exchange-only qubits. We also note that our process flow is fully compatible with Ge/SiGe heterostructures, allowing for the possibility of exploring Ge hole spin qubits in the future.

- Note: It should be interesting to comment on some perspectives/vision on the next steps in the full characterization of spin qubits: Rabi oscillation ... Exchange interaction function and entanglement ... Spin manipulation cross talk... Inter connection and power budget for large scale quantum computing solution(s).

We thank the referee for this suggestion. We hope the new qubit data now included in the revised manuscript, as well as the new sections in the Methods describing qubit measurements, help to provide this important context to the reader. In the interest of saving space and keeping our focus on cryo-prober characterization, we have focused new data and discussion on single qubit results and have not delved into discussions of multi-qubit operation. We hope the readers find this a satisfying extension of our main results with the cryo-prober, while leaving discussions of multi-qubit operation as an area of future work.

This robust and relevant study could be published with an improvement in the results following the previous questions/remarks. While waiting for the pleasure of seeing the update again if it is accepted for the future.

The results of this study will be useful to the community and industry in this field.

We thank the referee for their supportive comments.

Bibliography

- M. G. Borselli, K. Eng, R. S. Ross, et al., Undoped accumulation-mode Si/SiGe quantum dots, *Nanotechnology* 26, 375202 (2015).
- M. Brauns, S. V. Amitonov, P.-C. Spruijtenburg, and F. A. Zwanenburg, Palladium gates for reproducible quantum dots in silicon, *Scientific Reports* 8 (2018).
- L. Contamin, B. C. Paz, B. M. Diaz, et al., Methodology for an efficient characterization flow of industrial grade Si-based qubit devices, in 2022 International Electron Devices Meeting (IEDM), edited by IEEE (2022) pp. 22.1.1–22.1.4.
- J. P. Dodson, N. Holman, B. Thorgrimsson, et al., Fabrication process and failure analysis for robust quantum dots in silicon, *Nanotechnology* 31, 505001 (2020).
- W. Ha, S. D. Ha, M. D. Choi, et al., A flexible design platform for Si/SiGe exchange-only qubits with low disorder, *Nano Lett.* 22, 1443 (2022).
- N. W. Hendrickx, W.I.L. Lawrie, M. Russ, et al., A four-qubit germanium quantum processor. *Nature* 591, 580–585 (2021).
- L. Kranz, S. K. Gorman, B. Thorgrimsson, et al., Exploiting a Single-Crystal Environment to Minimize the Charge Noise on Qubits in Silicon, *Advanced Materials* 32, 2003361 (2020).
- A. Ortiz-Conde, F. J. García Sánchez, J. J. Liou, et al., A review of recent MOSFET threshold voltage extraction methods, *Microelectronics Reliability* 42, 583 (2002).
- S. G. J. Philips, M. T. Madzik, S. V. Amitonov, et al., Universal control of a six-qubit quantum processor in silicon, *Nature* 609, 919 (2022).
- C. Tahan, Opinion: Democratizing spin qubits, *Quantum* 5, 584 (2021).
- A. J. Weinstein, M. D. Reed, A. M. Jones, et al., Universal logic with encoded spin qubits in silicon, *Nature* 615, 817 (2023).
- N. M. Zimmerman, C.-H. Yang, N. S. Lai, et al., Charge offset stability in Si single electron devices with Al gates, *Nanotechnology* 25, 405201 (2014).
- A. M. J. Zwerver, T. Krähenmann, T. F. Watson et al., Qubits made by advanced semiconductor manufacturing, *Nat. Electron.* 5, 184 (2022).

Reviewer Reports on the First Revision:

Referees' comments:

Referee #1 (Remarks to the Author):

The revised manuscript by the Intel team convincingly addresses the critical concerns by the referees. Especially, the inclusion of material and process improvements and spin qubit data has substantially improved the quality of the manuscript.

Regarding the device quality improvements, still some details are somewhat lacking, (high-k stacks A and B, gate and interconnect materials), but I find it is acceptable given the limitation imposed by the authors' institution.

The spin qubit data in Extended Data Fig. 1 is noteworthy, showcasing measurements of 3Q and 12Q arrays, with one of the devices tuned up to a condition where all 12 spin qubits can be measured. The dataset on 39 spin qubits is the most extensive one reported to date, and the authors draw some valuable conclusions such as the limiting mechanism of dephasing (nuclear spin vs. charge noise).

Overall, I believe the revised manuscript is suitable for publication in Nature, with a few minor points to be considered:

Extended Data Fig. 1b, c: What is the reason for the reduced visibility in the RB experiment (c) compared to the Rabi oscillation (b)? Are these from different devices or using different readout schemes?

Estimation of ergodic T_2^* : The authors mention ^{29}Si and ^{73}Ge nuclei in the Si QW and the SiGe barrier limit T_2^* . I would appreciate further elaboration on which component, specifically the nuclear spins in the Si QW or the SiGe barrier, is dominantly affecting T_2^* .

Valley splitting: Some previous studies reported narrow Si QW (~ 5 nm) results in a sizable valley splitting, while the authors observe that the two-electron singlet-triplet energy splitting is typically comparable to T_e in their devices. Could this be related to the thermal budget and a resulting diffused interface between Si and SiGe?

Hahn echo: For a standard $X_{\pi/2} - X_{\pi} - X_{\pi/2}$ sequence, I do not expect the oscillation (or the linear phase accumulation with respect to wait time). Is there any phase intentionally added to the MW signal? Additionally, when fitting the echo data, is a Gaussian decay always applied, or are different exponents used?

SEM: Is it possible to include a scale bar? I understand this does not add any new information since the gate pitch of 60 nm is given in the text, but it may aid readers in identifying the device size quickly.

Line 597: The part number of the Keysight vector source may be E8267D instead of E8257D.

Referee #2 (Remarks to the Author):

First, I would like to thank the authors for their detailed responses to the points that were raised. I think this new manuscript is clearly improved and that the additional discussions strengthen the

interest of the paper.

As mentioned in the first round of review, I am still not certain that the scientific novelty of the results warrants publication in Nature. Multiple advances are indeed reported, but for me they remain individually incremental. It is true, however, that these results are a breakthrough in terms of technological maturity (yield, reproducibility, etc.) which is one of the main obstacles linked to spin qubits. I find hard to judge the adequacy of this type of results with the editorial line and I leave this decision to the editor.

Below are a few comments and additional questions on the new elements of the paper.

Qubit measurements

The addition of the spin manipulation results is interesting to illustrate that the technological platform presented can provide qubits with state-of-the-art performance. This aspect is only very briefly discussed in the main text, which I find legitimate since it is out of the paper scope. However, the volume of information reported in the Methods section somehow gives the impression that a second article is embedded there. I know that some of the reviewers opposed the absence of spin demonstrations, but personally, I find it affects the clarity of the message. It may be worth to refer the reader to a second manuscript dealing with these results in more details.

That said, seeing these results raises some additional questions to me.

- Are the micromagnets post-processed or directly integrated at wafer scale?
- What are the selection criteria for the devices measured in dilution fridge?
- Are there any relevant splits to be discussed on the studied wafers other than the isotopic purification of the substrate?
- To which extent are the tune-up polarizations reproducible between the first characterization on the CWP and the final dilution measurements (any impact from the thermal cycling, preparation of the chips, etc)?

In connection with the yield discussions of the paper:

- Did all the devices that were yielding on CWP allowed spin measurements? Or is valley splitting (or other mechanisms) limiting?
- Most of the devices in Extended Data Fig. 1. were only measured on one or a few adjacent quantum dots. Were the other QDs not satisfying or just not studied?
- Was there a spatial correlation observed on the ability to define a spin qubit within the devices?

Process optimization

Line 117: In particular, we find that fixed charge can be reduced in our devices by limiting the thermal budget of the spacer process that follows high-k deposition.

Intuitively I would think that a higher thermal budget may activate the dynamic re-structuring of defects in the high-k and reduce their number. Can you comment on the suspected underlying mechanism?

Supplemental Information section I

The references to subplots of Supplementary Fig. 1 are not correct in the text (labeled from c to f instead of a to d).

Supplemental Information section II & Methods - Voltage sharing analysis

I find the discussion on the impact of adjusting the tunnel barriers very interesting and thank the

authors for having carried out additional measurements and thoroughly analysed this point. I find the results and analysis convincing.

After reading the new version of the Voltage sharing analysis section, I realized that I had misunderstood the methodology used. In figure extended data 6, I initially thought that the common voltage was unique across the entire wafer. Now I realize that the common voltage is actually readjusted for each 12Q device.

This raises a few other questions:

- Mathematically, how is the fraction of dots at occupation $n=1e$ supposed to evolve as a function of the number of QDs in the device? I expect this fraction decreases as the number of QDs increases, since in the extreme case of single QD devices, this fraction is by definition 100%. I have the feeling that 12Q is large enough to be representative of an arbitrarily large number of QDs, but it would be interesting to quantify it. Is it possible to establish an analytical expression of this fraction by making the link with the ΔV_{1e} and $1e-2e$ addition voltage distributions?

Referee #3 (Remarks to the Author):

In this revised manuscript, the authors made significant efforts to address the comments by the reviewers. The newly added spin qubit data and more discussions on the process optimization will make the whole study more interesting to readers. However, the following points should be better addressed before I can recommend its publication in Nature.

1. Qubit data. The statistical report of qubit data will be much appreciated by the community. In my point of view, it should be part of the main story rather than the extended data. More importantly, more discussion support by numbers should be given to justify that high electrostatic reliability translates to high qubit success rate (line 291), which is a key message of this work.
2. Process and device details. The discussions are still very vague. It struck me as odd that authors have shown the TEM cross sections at multiple conferences, whereas in Fig. 2 only the schematic of the device is presented. On their MOS qubit structures, authors have shown details about the qubit devices (Ref 26). Another example is the high-k stack, where authors claim that the thermal treatment could affect the fixed charge without specifying the temperature. It is well known that thermal treatment can also influence the Si/SiGe interface for a temperature higher than 600-700 degree Celsius. It leaves the readers wondering what is the physical reason behind.

Followings are some comments on the newly added discussions in the revised manuscript.

1. I am not sure simply removing the citation to M. Veldhorst et al Nat. Commun 8, 1766 (2017) is the best practice for the purpose of discussing voltage sharing. The voltage sharing is an important motivation of the work so it should be treated more carefully. The references justifying voltage sharing (Ref 45 & 46) have less impact than the work by Veldhorst et al (based on the number of citations). Might be good to discuss that the shared control structure is more compatible with current CMOS technology, and highlight that higher level of uniformity allows less overheads in operating a larger qubit array (as well as higher coherence of electron spins can be desirable). Additionally, in line 266, ref 47 is still there.
2. Is the fabrication of micromagnet layer really well understood? In Ref 15 Fig 2b, the resonance frequency of the qubit along the 1D array is not as expected. Additionally, cobalt is a very mobile material and special care is need in introducing to the fab flow. The metallic phase and the grain size could strongly impact the field gradient. In line 654, the spin driving field gradient is roughly half of that in Ref. 58. However, the Rabi frequency is one order of magnitude lower. I would expect the real gradient is lower than the simulated value, which can also explain that the ^{28}Si qubits are nuclear spin limited rather than charge noise.
3. Line 215: it is not clear to me that the charge sensor drift is the same source as the charge noise. In the low frequency ranges ($\ll 1\text{Hz}$), the noise curve can have very different frequency

spectrums as reported in Ref. 61 Fig. 3a and Elsayed et al arXiv:2212.06464 Fig S5. As authors wrote that this type of instable could "settle" with time, I would expect it is related to the settle of mobile charge in the gate stack. Whereas charge noise affecting the qubit performance can originate from two level fluctuators.

4. Line 623: authors used 3e to avoid low valley splitting problem. However, this type of operation with different electron occupation weakens the argument high voltage uniformity for shared control. Good to have more discussion on the advantages and disadvantages of 1e vs 3e states.

5. I would encourage the authors to take out more careful analysis on their mirror-symmetric pairs analysis. The introduction of new statistical analysis needs careful examination. The standard CMOS matched pair has transistors with similar distance while here the outer gates are further away than inner ones.

Referee #4 (Remarks to the Author):

Paper Ref : 2023-07-11832A

Review of "Probing single electrons across 300 mm spin qubit wafers"

Authors : Samuel Neyens*,,, Otto Zietz*, Thomas Watson, Florian Luthi, Aditi Nethwewala, Hubert George, Eric Henry, Andrew Wagner, Mohammad Islam, Ravi Pillarisetty, Roza Kotlyar, Kent Millard, Stefano Pellerano, Nathan Bishop, Stephanie Bojarski, Jeanette Roberts, and James S. Clarke.

Feedbacks :

The authors present an updated version of a comprehensive study on a methodology for characterizing 300 mm wafers for monitoring cryogenic processes, linear spin devices and multi-electron spin qubits (12 Qubits) at cryogenic temperature (1.6K). The proposed new version takes into account the questions and comments from my first opinion. They have significantly improved this article with detailed and robust results/arguments/references. This study is ready for publication at this point in my opinion. The main results of this study will be useful to readers.

Author Rebuttals to First Revision:

Referees' comments:

Referee #1 (Remarks to the Author):

The revised manuscript by the Intel team convincingly addresses the critical concerns by the referees. Especially, the inclusion of material and process improvements and spin qubit data has substantially improved the quality of the manuscript.

Regarding the device quality improvements, still some details are somewhat lacking, (high-k stacks A and B, gate and interconnect materials), but I find it is acceptable given the limitation imposed by the authors' institution.

The spin qubit data in Extended Data Fig. 1 is noteworthy, showcasing measurements of 3Q and 12Q arrays, with one of the devices tuned up to a condition where all 12 spin qubits can be measured. The dataset on 39 spin qubits is the most extensive one reported to date, and the authors draw some valuable conclusions such as the limiting mechanism of dephasing (nuclear spin vs. charge noise).

Overall, I believe the revised manuscript is suitable for publication in Nature, with a few minor points to be considered:

We thank the referee again for their careful reading of our manuscript and for these supportive comments. Below we give our full response to the referee's comments and questions and discuss our modifications to the manuscript.

Extended Data Fig. 1b, c: What is the reason for the reduced visibility in the RB experiment (c) compared to the Rabi oscillation (b)? Are these from different devices or using different readout schemes?

The Rabi data and the randomized benchmarking data are taken from the same device. The reason for the difference in visibility is that the randomized benchmarking experiment was performed first, before additional tuning and setup changes were done to optimize the readout visibility for another ongoing experiment.

Estimation of ergodic T_2^* : The authors mention ^{29}Si and ^{73}Ge nuclei in the Si QW and the SiGe barrier limit T_2^* . I would appreciate further elaboration on which component, specifically the nuclear spins in the Si QW or the SiGe barrier, is dominantly affecting T_2^* .

We have now added additional explanation to the Methods section "Coherence modeling" to clarify that, while the amount of contribution from the SiGe barrier can depend sensitively on the width of the Si/SiGe interface, incorporating a realistic interface width of 1 nm into the simulations leads to the conclusion that the main limiter to coherence is residual ^{29}Si in the

quantum well. For reference, please see below a plot of simulated T_2^* as a function of both ^{29}Si concentration in the quantum well and ^{73}Ge concentration in the SiGe barrier. The red square represents the case of our purified (to 800 ppm ^{29}Si) quantum wells.

Valley splitting: Some previous studies reported narrow Si QW (~ 5 nm) results in a sizable valley splitting, while the authors observe that the two-electron singlet-triplet energy splitting is typically comparable to T_e in their devices. Could this be related to the thermal budget and a resulting diffused interface between Si and SiGe?

We do not expect the quantum well interface sharpness to be limited by inter-diffusion during fabrication, as we have confirmed with TEM measurements that this interface sharpness agrees between processed and unprocessed wafers. Rather we expect the baseline one-electron valley splitting to be limited by as-grown interface disorder. The exact details of this disorder are not known, but this could likely include alloy disorder [Losert, et al. 2023], interface roughness [Gamble, et al. 2016], or their convolution. We find the two-electron singlet-triplet splitting (which defines our readout window for Pauli spin blockade) is often further reduced, likely by electron-electron interactions that arise from details of the lateral confinement [Ercan, et al. 2021]. We have added a sentence to the revised manuscript indicating our interpretation of this.

Hahn echo: For a standard $X_{\pi/2} - X_{\pi} - X_{\pi/2}$ sequence, I do not expect the oscillation (or the linear phase accumulation with respect to wait time). Is there any phase intentionally added to the MW signal? Additionally, when fitting the echo data, is a Gaussian decay always applied,

or are different exponents used?

For both the Ramsey sequence and the Hahn echo sequence, an artificial oscillation is introduced to improve the reliability of the fit. We have updated the Methods section titled “Coherence measurements” to describe this more clearly. We also now clarify that the decay exponent is kept as a free parameter when fitting both the Ramsey and the Hahn echo data.

SEM: Is it possible to include a scale bar? I understand this does not add any new information since the gate pitch of 60 nm is given in the text, but it may aid readers in identifying the device size quickly.

In the revised manuscript, we now include a scale bar with the top-down SEM image in Fig. 3.

Line 597: The part number of the Keysight vector source may be E8267D instead of E8257D.

We thank the referee for pointing this out and apologize for the error. It has been corrected in the revised manuscript.

Referee #2 (Remarks to the Author):

First, I would like to thank the authors for their detailed responses to the points that were raised. I think this new manuscript is clearly improved and that the additional discussions strengthen the interest of the paper.

We thank the referee again for their careful reading of our manuscript and for these supportive comments. Below we give our full response to the referee’s comments and questions and discuss our modifications to the manuscript.

As mentioned in the first round of review, I am still not certain that the scientific novelty of the results warrants publication in Nature. Multiple advances are indeed reported, but for me they remain individually incremental. It is true, however, that these results are a breakthrough in terms of technological maturity (yield, reproducibility, etc.) which is one of the main obstacles linked to spin qubits. I find hard to judge the adequacy of this type of results with the editorial line and I leave this decision to the editor.

Below are a few comments and additional questions on the new elements of the paper.

Qubit measurements

The addition of the spin manipulation results is interesting to illustrate that the technological platform presented can provide qubits with state-of-the-art performance. This aspect is only very briefly discussed in the main text, which I find legitimate since it is out of the paper scope. However, the volume of information reported in the Methods section somehow gives the impression that a second article is embedded there. I know that some of the reviewers opposed

the absence of spin demonstrations, but personally, I find it affects the clarity of the message. It may be worth to refer the reader to a second manuscript dealing with these results in more details.

We acknowledge that a significant amount of data and information relating to qubit measurements was added in the previous revision, which has significantly lengthened the Methods section. Since the focus of the paper is on high volume data collection and a report of high yield and high electrostatic reliability, we believe the proper way to demonstrate qubit performance is also to show a significant sample size of devices, hence the volume of qubit data included in Extended Data Fig. 1. The amount of supporting information added to the Methods is based on an expectation that sharing such qubit data will invite many questions from readers about the measurement methods and a general desire to share as much about these methods as is practical. We do not have plans to write a second manuscript based on these qubit results, as we do not see these measurements by themselves constituting significant novelty. We do acknowledge the concerns about the scope of the paper and will consult with the editor about the best ways to support the main messages without compromising on overall clarity.

That said, seeing these results raises some additional questions to me.

- Are the micromagnets post-processed or directly integrated at wafer scale?

For all data shown in the manuscript, micromagnets are processed at the coupon level rather than at wafer scale. More recently, we have also integrated the micromagnet fabrication into the 300 mm process and have begun producing full wafers with micromagnets.

- What are the selection criteria for the devices measured in dilution fridge?

Wafers are routinely characterized with the cryo-prober using the set of measurements featured in the manuscript, from basic checks of yield to charge sensing at the few-electron regime. When a 12QD device does not fully pass our requirements for “yield,” other devices from the same die are typically avoided for dilution refrigerator measurements, out of caution. Beyond this level of screening, the selection process for qubit measurements can vary depending on the goal of the experiment. For most experiments, devices are selected arbitrarily from the (majority) set of yielding die. For experiments focused on multi-qubit operation and/or tune-up of a full array, devices are sometimes selected by examining charge sensing data from the cryo-prober and checking the orderliness of the charge transitions in the few-electron regime.

- Are there any relevant splits to be discussed on the studied wafers other than the isotopic purification of the substrate?

Overall, these wafers do not include process splits that are expected to affect qubit coherence. The heterostructure growth (besides isotopic purification) processes are the same. There is a change in the gate process between w4 and w5 (of the 28-Si wafers), and an experimental dielectric process in w2 (dev7), but these were not expected to affect qubit performance

metrics, and we do not observe a systematic effect on the coherence times as reflected by $T_2^*(\infty)$.

- To which extent are the tune-up polarizations reproducible between the first characterization on the CWP and the final dilution measurements (any impact from the thermal cycling, preparation of the chips, etc)?

While we have not carried out a careful study comparing the operating voltages of the same device between cryo-prober measurement and subsequent dilution refrigerator measurement, we have not observed significant changes in the behavior of devices or in the disorder after micromagnet fabrication and packaging. We also find negligible impact from thermal cycling of devices, as we have also cycled wafers multiple times in the cryo-prober and obtained consistent results each time.

In connection with the yield discussions of the paper:

- Did all the devices that were yielding on CWP allowed spin measurements? Or is valley splitting (or other mechanisms) limiting?

We find that the success of obtaining spin measurements (e.g., measurements of spin resonance, coherence times, and/or Rabi oscillations) has not been limited by any issues of device yield or electrostatics, thanks to the high device yield and screening capabilities of the cryo-prober.

While we find low valley splitting does impact a fraction of qubit devices (roughly ~20%), this does not prevent qubit measurements, but can lead to complications such as extra spin resonances [Kawakami, et al. 2016] and overall worse performance.

We also note that there can sometimes be issues between the 300 mm fabrication and dilution refrigerator measurements. These tend to occur in batches and are unrelated to the underlying device quality. For example, out of the full experimental set featured in Extended Data Fig. 1g, two dilution refrigerator cooldowns (containing two devices each) did not result in spin qubit data being obtained. In one case, this was due to a wire bonding issue (likely ESD), and in the other case, this was due to an issue with the micromagnet processing. In the first case, this was confirmed with measurements of gate leakage which had not been observed in the cryo-prober measurements. In the second case, every aspect of the devices functioned well, but no spin resonance was observed. While these occasional cases can still present obstacles to qubit measurement, the overall picture supports our statement in the revised conclusion, that improvements in the yield and electrostatic reliability of devices combined with cryo-prober testing have eliminated underlying device quality as a barrier to progress in the dilution refrigerator measurements.

- Most of the devices in Extended Data Fig. 1. were only measured on one or a few adjacent quantum dots. Were the other QDs not satisfying or just not studied?

In these cases, the other quantum dots were simply not studied. The experiments summarized in Extended Data Fig. 1g had a variety of different objectives, some more focused on qubit

control questions as opposed to full array operation. We note that dev13 and dev14, which feature data from a subset of qubits in Extended Data Fig. 1g, are current experiments that are still in progress.

- Was there a spatial correlation observed on the ability to define a spin qubit within the devices?

We have not observed any spatial dependence of the ability to define qubits along the array.

Process optimization

Line 117: In particular, we find that fixed charge can be reduced in our devices by limiting the thermal budget of the spacer process that follows high- κ deposition. Intuitively I would think that a higher thermal budget may activate the dynamic re-structuring of defects in the high- κ and reduce their number. Can you comment on the suspected underlying mechanism?

We acknowledge that while the previous revision introduced the method of reducing temperature budget to explain device improvements, this left open the question of the underlying mechanism. In the revised manuscript we now explicitly attribute the generation of fixed charge at higher spacer temperatures to crystallization within the high- κ stack, which can lead to trapped charge at grain boundaries. We have also contextualized our reference to “high” and “low” temperature processes by comparing them to a commonly recognized thermal budget for back-end-of-line (BEOL) compatibility, where $T_{\text{BEOL}} \sim 400$ °C. In the revised manuscript, we now specify that the improvements from changing the spacer process come after switching from a BEOL-incompatible process ($T > T_{\text{BEOL}}$) to a BEOL-compatible process ($T < T_{\text{BEOL}}$).

Supplemental Information section I

The references to subplots of Supplementary Fig. 1 are not correct in the text (labeled from c to f instead of a to d).

We thank the referee for pointing this out and apologize for the error. The figure labels are fixed in the revised manuscript.

Supplemental Information section II & Methods - Voltage sharing analysis

I find the discussion on the impact of adjusting the tunnel barriers very interesting and thank the authors for having carried out additional measurements and thoroughly analysed this point. I find the results and analysis convincing.

After reading the new version of the Voltage sharing analysis section, I realized that I had misunderstood the methodology used. In figure extended data 6, I initially thought that the common voltage was unique across the entire wafer. Now I realize that the common voltage is

actually readjusted for each 12Q device.

This raises a few other questions:

- Mathematically, how is the fraction of dots at occupation $n=1e$ supposed to evolve as a function of the number of QDs in the device? I expect this fraction decreases as the number of QDs increases, since in the extreme case of single QD devices, this fraction is by definition 100%. I have the feeling that 12Q is large enough to be representative of an arbitrarily large number of QDs, but it would be interesting to quantify it. Is it possible to establish an analytical expression of this fraction by making the link with the ΔV_{1e} and $1e-2e$ addition voltage distributions?

We note that our choice to assign individual common voltage values to each device is based on the expectation that future voltage sharing applications could entail finding and setting optimal common voltages for each qubit array chip, as opposed to using the same common voltage values for all chips taken from a wafer. We have added the term “device-specific” to the Methods section “Voltage sharing analysis” where this is described, to hopefully make our approach clearer.

To answer the referee’s question about the dependence of the $n=1e$ fraction on array size, we have updated the revised manuscript to include new analysis, simulations, and discussion in the Methods section titled “Voltage sharing analysis” as well as two new panels to the related figure (Extended Data Fig. 5). These new results confirm the referee’s suggestion that the success rate will be trivially 100% at an array size $N=1$ and will tend to decrease with array size. This decrease in success rate will saturate at what we call the “large array limit,” where V_{1e} and V_{2e} data from each device are well approximated by a normal distribution. We now include an analytical formula for the expected success rate in this limit based on the cumulative distribution function of the standard normal distribution and using statistical values for addition voltage and V_{1e} variation. In the intermediate range between the two limits, we find that the rate of decrease will depend on the method for choosing the common voltage. Since such methods inherently involve sampling noise and can include an optimization algorithm, we do not expect this intermediate range can be easily described with an analytical formula, hence our choice instead to explore it using simulations. These simulations also show that, when the common voltage is chosen based on mean V_{1e} values for each device, the success rate quickly drops off to its limiting value, nearly saturating by $N=12$ quantum dots, as the referee suggests. By contrast, our method of optimizing this common voltage to maximize the $n=1e$ fraction in each device leads to a boost in the success rate over a large range of array sizes, up to around $N \sim 1000$ quantum dots. We also now discuss the implications of these findings for large arrays and how improvements to the V_{1e} variation can significantly increase the success rate even in the large array limit.

Referee #3 (Remarks to the Author):

In this revised manuscript, the authors made significant efforts to address the comments by the reviewers. The newly added spin qubit data and more discussions on the process optimization will make the whole study more interesting to readers. However, the following points should be

better addressed before I can recommend its publication in Nature.

We thank the referee again for their careful reading of our manuscript and for these supportive comments. Below we give our full response to the referee's comments and questions and discuss our modifications to the manuscript.

1. Qubit data. The statistical report of qubit data will be much appreciated by the community. In my point of view, it should be part of the main story rather than the extended data. More importantly, more discussion support by numbers should be given to justify that high electrostatic reliability translates to high qubit success rate (line 291), which is a key message of this work.

While the qubit data added to the previous revision is an important part of the overall message of the paper, we still believe it is best placed in the Extended Data rather than the main text. This is based on length constraints in the main text as well as our belief that there is less novelty in this figure compared with other content currently in the main text. We will also consult with the editor about the best way to include this qubit data in the paper for purposes of clarity.

We agree that the term "success rate" does imply that quantitative data should be included. For the experimental set of qubit devices featured in Extended Data Fig. 1g, quantifying that success rate was not the goal, hence why certain devices only feature data from a subset of qubits, where the focus of the experiment was more on qubit control as opposed to full array operation. Given this, we have decided to modify our language in the conclusion to say instead that the high yield and electrostatic reliability, coupled with the cryo-prober testing methodology, has eliminated failures due to yield and electrostatics at the dilution refrigerator stage. We consider this an important message since such issues are frequently barriers to progress across the spin qubit research field, and the combination of the results and methods described in the present manuscript has led to the removal of these barriers.

To elaborate on these claims, we note here that there can sometimes be issues between the 300 mm fabrication and dilution refrigerator measurements. These tend to occur in batches and are unrelated to the underlying device quality. For example, out of the full experimental set featured in Extended Data Fig. 1g, two dilution refrigerator cooldowns (containing two devices each) did not result in spin qubit data being obtained. In one case, this was due to a wire bonding issue (likely ESD), and in the other case, this was due to an issue with the micromagnet processing. In the first case, this was confirmed with measurements of gate leakage which had not been observed in the cryo-prober measurements. In the second case, every aspect of the devices functioned well, but no spin resonance was observed. While these occasional cases can still present obstacles to qubit measurement, the overall picture supports our statement in the revised conclusion, that improvements in the yield and electrostatic reliability of devices combined with cryo-prober testing have eliminated underlying device quality as a barrier to progress in the dilution refrigerator measurements.

2. Process and device details. The discussions are still very vague. It struck me as odd that authors have shown the TEM cross sections at multiple conferences, whereas in Fig. 2 only the

schematic of the device is presented. On their MOS qubit structures, authors have shown details about the qubit devices (Ref 26). Another example is the high-k stack, where authors claim that the thermal treatment could affect the fixed charge without specifying the temperature. It is well known that thermal treatment can also influence the Si/SiGe interface for a temperature higher than 600-700 degree Celsius. It leaves the readers wondering what is the physical reason behind.

We believe the 3D schematic presently used in Fig. 2 has advantages for clarity in illustrating how devices function, but we also agree that it would be beneficial to include a TEM image of the device under study for better transparency and for continuity with conference presentations that readers in the community may have seen. We have now added a cross sectional TEM image of one of our devices to Fig. 1 of the latest revision. We have also included a scale bar on this image and added a scale bar to the top-down SEM image in Fig. 3 as well.

In the discussion of temperature, we agree that the current description leaves open questions for the reader and could benefit from more context. While we are restricted from sharing specific temperatures due to certain processes being proprietary, we can contextualize our reference to “high” and “low” temperature processes by comparing them to a commonly recognized thermal budget for back-end-of-line (BEOL) compatibility, where $T_{\text{BEOL}} \sim 400 \text{ }^\circ\text{C}$. In the revised manuscript, we now specify that the improvements from changing the spacer process come after switching from a BEOL-incompatible process ($T > T_{\text{BEOL}}$) to a BEOL-compatible process ($T < T_{\text{BEOL}}$). Also, while the previous revision left open the question of whether the change in thermal budget caused a change in the heterostructure or the gate dielectric, we now explicitly attribute the generation of fixed charge at higher spacer temperatures to crystallization within the high-k stack, which can lead to trapped charge at grain boundaries. (Following a response above to another referee, we also note that we have not found any observable change in the Si/SiGe heterostructure resulting from thermal effects of fabrication.) We believe these updates will help the reader to understand the significance of the change in temperature as well as the regions of the device that are most sensitive to these temperature changes. While every fabrication flow will have different process interactions, materials details, and thermal budgets, we believe this discussion in the revised manuscript gives an example of a kind of process interaction that could occur in other flows and explains a general approach for iterating through process changes to mitigate it.

Followings are some comments on the newly added discussions in the revised manuscript.

1. I am not sure simply removing the citation to M. Veldhorst et al Nat. Commun 8, 1766 (2017) is the best practice for the purpose of discussing voltage sharing. The voltage sharing is an important motivation of the work so it should be treated more carefully. The references justifying voltage sharing (Ref 45 & 46) have less impact than the work by Veldhorst et al (based on the number of citations). Might be good to discuss that the shared control structure is more compatible with current CMOS technology, and highlight that higher level of uniformity allows less overheads in operating a larger qubit array (as well as higher coherence of electron spins can be desirable). Additionally, in line 266, ref 47 is still there.

We agree that it is better for this discussion to include all three original references and to discuss how they are differently impacted by improvements to uniformity/variation. In the section where we discuss the benefits of improving variation, we now make a clearer distinction between the Veldhorst proposal based on floating memory and the other two proposals based on voltage sharing. We discuss how each of the two approaches would specifically benefit from reducing variation in spin qubit operating voltages. Later in the manuscript, when we reference voltage sharing specifically, we have updated the citations to refer only to proposals based explicitly on voltage sharing.

2. Is the fabrication of micromagnet layer really well understood? In Ref 15 Fig 2b, the resonance frequency of the qubit along the 1D array is not as expected. Additionally, cobalt is a very mobile material and special care is needed in introducing it to the fab flow. The metallic phase and the grain size could strongly impact the field gradient.

In line 654, the spin driving field gradient is roughly half of that in Ref. 58. However, the Rabi frequency is one order of magnitude lower. I would expect the real gradient is lower than the simulated value, which can also explain that the ^{28}Si qubits are nuclear spin limited rather than charge noise.

In our previous response, we stated, “While the micromagnet layer is not included in the 300 mm process characterized in this work, the path to doing so is well understood.” Our intention here was to imply that including micromagnets at the 300 mm level was well understood. While the spin qubit data presented in this work is taken on devices where micromagnets were added at the coupon level, we have since begun fabricating 300 mm wafers with micromagnets. Also, while cobalt does come with challenges, there is precedent for its use in CMOS logic processes [Auth, et al. 2017].

We agree that factors like grain size can strongly impact the field gradient, and we have taken steps to understand the role this plays in our devices. For the micromagnets in this work, we find good agreement between simulations and experiments by incorporating grain sizes of 14 nm, a grain size which is consistent also with X-ray diffraction measurements of our films. For reference, please see the figure below comparing simulated and measured qubit frequencies across the 12-qubit array. We note that the factor of two lower driving gradient in our devices compared with those studied in [Yoneda, et al. 2017] is primarily a result of the stack height, causing our micromagnets to be at a greater distance away from the qubits. The decoherence gradients we report (0.007-0.030 mT/nm) are similarly reduced by this distance as well as by the details of the design. Our decoherence gradients are about an order of magnitude less than that reported in [Yoneda, et al. 2017] (~ 0.2 mT/nm), causing qubit coherence to be much less sensitive to charge noise and therefore still limited by nuclear spins even after isotopic purification to 800 ppm ^{29}Si .

Comparisons between Rabi frequencies in our work and [Yoneda, et al. 2017] are difficult without knowing the power delivered at the sample in that work. For the Rabi oscillation in Extended Data Fig. 1b, we estimate a total power of ~ -35 dBm delivered at the sample, based on the applied power of -2 dBm from the MW source and the total line attenuation (-21 dBm from attenuators and an additional -14 dBm from higher frequency

attenuation at 7.5 GHz from coaxial cables). We have now added this detail into the Methods so that others can compare to our work.

In addition to the unknown drive power used in [Yoneda, et al. 2017], we note that the quantum dots used in that work were formed in “depletion mode” and were likely subject to weaker lateral confinement than the “accumulation mode” devices we study, which have screening gates held at a low voltage on either side of the quantum dot. Due to this difference in confinement, we suspect that the quantum dot displacement for a given voltage swing could be much larger in [Yoneda, et al. 2017] than in our devices. We believe that a better comparison may be found with [Philips, et al. 2022] and the follow-up paper [Undseth, et al. 2023], where with a very similar device and micromagnet design they achieve ~ 3 MHz Rabi frequency at an estimated -30 dBm power applied to the sample.

3. Line 215: it is not clear to me that the charge sensor drift is the same source as the charge noise. In the low frequency ranges ($\ll 1$ Hz), the noise curve can have very different frequency spectrums as reported in Ref. 61 Fig. 3a and Elsayed et al arXiv:2212.06464 Fig S5. As authors wrote that this type of instable could “settle” with time, I would expect it is related to the settle of mobile charge in the gate stack. Whereas charge noise affecting the qubit performance can originate from two level fluctuators.

We agree that the situation of charge noise and/or charge offset stability in these devices is complicated, and multiple physical mechanisms may contribute. In our discussion in the manuscript and in our previous referee response, our intention was not to oversimplify or to make strong claims about the physical mechanisms behind these effects. Rather the goal of the statement in the manuscript is solely to propose a vector for improving the quality (and therefore the success rate) of automated charge sensing measurements in the cryo-prober. We have now updated the relevant sentence in the revised manuscript in two ways, to improve

(hopefully) its clarity and accuracy. First, we have added the comment that lowering the electron temperature will improve the sensitivity of the charge sensor. This was suggested by the referee in the previous round of comments, and we agree that this should be included in the discussion. Second, we have modified our language to replace “charge noise” with “charge offset stability” and suggest that it will “possibly” improve with reduced temperature due to “de-activation of two-level fluctuators.” While the work we cite regarding the temperature dependence of charge noise [Connors, et al. 2020] focuses on the amplitude at 1 Hz, our charge sensing measurements sample charge stability over a frequency scale from $\sim 10^{-2}$ to 10^1 Hz. We argue therefore that it is reasonable to expect some improvement of the total stability and/or noise captured by the measurement. We agree that not all charge sensor jumps have the same underlying cause, but here we are positing a general improvement of charge stability with reduced temperature and not an improvement of all underlying mechanisms. Since this is simply a hypothesis for the results of future work (lowering cryo-prober temperature), we hope these modifications are sufficient to clarify the message to the reader and that these statements do not contradict any published work on the matter.

We also note that this particular discussion on charge sensor success rate has now been moved to the Methods section “Charge sensing success rate” to satisfy length requirements in the main text.

4. Line 623: authors used 3e to avoid low valley splitting problem. However, this type of operation with different electron occupation weakens the argument high voltage uniformity for shared control. Good to have more discussion on the advantages and disadvantages of 1e vs 3e states.

We agree that while using the 3e state has advantages for achieving a large readout window in Pauli spin blockade, the practice of alternating between 1e and 3e states across spin qubit arrays has drawbacks for scaling, in particular adding overhead to solutions based on voltage sharing. We have added a comment to this effect to the Methods section “Qubit readout and initialization” in the revised manuscript.

5. I would encourage the authors to take out more careful analysis on their mirror-symmetric pairs analysis. The introduction of new statistical analysis needs careful examination. The standard CMOS matched pair has transistors with similar distance while here the outer gates are further away than inner ones.

We agree that new methods of analysis deserve careful motivation and justification. For clarity, we have changed the language when introducing this technique to say that we “adapt” rather than “follow” the standard CMOS technique. To further justify our adaptation of the method to multi-gate quantum dot devices, we have made several updates to the revised manuscript.

First, we have added a designated section in the Methods titled “matched pair voltage difference analysis” to expand the discussion behind our motivation for using the method of mirror-symmetric pairs in the analysis. We note that we have moved some of this discussion from the main text into this new Methods section both to improve the cohesiveness of the discussion and to satisfy length requirements from the journal.

Second, we have now performed an alternate version of our analysis, where we analyze nearest-neighbor pairs rather than mirror-symmetric pairs, to quantify the differences between our preferred method and one that is based more directly on the standard approach for transistors. We have added a figure to the Extended Data section and a full discussion to the Methods section summarizing the results of this analysis. In this analysis, we find that using mirror-symmetric pairs does reduce systematic effects of geometric variation along the multi-gate quantum dot array, suggesting it is the better method for estimating the random variation than the analogous method based on nearest-neighbor pairs. We note that while more complex alternatives to this analysis may exist, perhaps by excluding certain gates or separating the analysis out by each unique gate pair, we argue that our method has the benefit of including all gates in the array and generates a single variation metric that can be used to compare average device performance between wafers. While we acknowledge that the variable distance between gate pairs in our method does present limitations as the arrays become significantly larger in size, we also now suggest a future alternative based on analyzing the variation within repeating unit cells of an array, where each unit cell is roughly the same size as the 12-quantum-dot (12QD) arrays studied here.

Third, in the Supplementary Information section where we analyze the effects of tuning barrier voltages on the matched pair variation analysis, we have updated Supplementary Fig. 3e, which summarizes the correlation coefficients between plunger gate pairs, to show that we do not find a strong dependence of correlation coefficient on the separation distance between gate pairs in these 12QD arrays. We argue that this adds further justification to the approximation used in that analysis where we assume that all gate pairs have the same correlation coefficient representing correlation within the length scale of a 12QD device. Again, as devices become significantly larger, this approximation may no longer hold, as correlation may decrease at greater distances than those considered here. But our suggestion of the use of a repeating unit cell should account for this, so long as the size of the unit cell is less than the correlation length of the systematic components of voltage variation one is intending to factor out.

Referee #4 (Remarks to the Author):

Paper Ref : 2023-07-11832A

Review of “Probing single electrons across 300 mm spin qubit wafers”

Authors : Samuel Neyens*, Otto Zietz*, Thomas Watson, Florian Luthi, Aditi Nethwewala, Hubert George, Eric Henry, Andrew Wagner, Mohammad Islam, Ravi Pillarisetty, Roza Kotlyar, Kent Millard, Stefano Pellerano, Nathan Bishop, Stephanie Bojarski, Jeanette Roberts, and James S. Clarke.

Feedbacks :

The authors present an updated version of a comprehensive study on a methodology for characterizing 300 mm wafers for monitoring cryogenic processes, linear spin devices and

multi-electron spin qubits (12 Qubits) at cryogenic temperature (1.6K). The proposed new version takes into account the questions and comments from my first opinion. They have significantly improved this article with detailed and robust results/arguments/references. This study is ready for publication at this point in my opinion. The main results of this study will be useful to readers.

We thank the referee again for their careful reading of our manuscript and for these supportive comments.

Note to all reviewers: when checking the decay exponents extracted from the fits for qubit coherence times, we found a bug in some of the earlier coherence fits that resulted in some nat-Si T_2^* times being over-estimated. For the latest revision, we have now re-done all extractions of qubit coherence times and updated the plot in Extended Data Fig. 1g. All conclusions to the data are unchanged. We have also added error bars to this plot to show the uncertainty of the fit results.

Bibliography

- C. Auth, A. Aliyarukunju, M. Asoro, et al., A 10nm high performance and low-power CMOS technology featuring 3rd generation FinFET transistors, Self-Aligned Quad Patterning, contact over active gate and cobalt local interconnects, 2017 IEEE International Electron Devices Meeting (IEDM), San Francisco, CA, USA, 2017, pp. 29.1.1-29.1.4
- E. J. Connors, J. J. Nelson, H. Qiao, et al., Low-frequency charge noise in Si/SiGe quantum dots, *Phys. Rev. B* 100, 165305 (2020).
- H. E. Ercan, S. N. Coppersmith, and M. Friesen, Strong electron-electron interactions in Si/SiGe quantum dots, *Phys. Rev. B* 104, 235302 (2021).
- J. K. Gamble, P. Harvey-Collard, N. T. Jacobson, et al., Valley splitting of single-electron Si MOS quantum dots, *Appl. Phys. Lett.* 109, 253101 (2016).
- E. Kawakami, T. Jullien, P. Scarlino, et al., Gate fidelity and coherence of an electron spin in an Si/SiGe quantum dot with micromagnet, *Proceedings of the National Academy of Sciences* 113, 11738 (2016).
- M. P. Losert, M. A. Eriksson, R. Joynt, et al., Practical strategies for enhancing the valley splitting in Si/SiGe quantum wells, *Phys. Rev. B* 108, 125405 (2023).
- S. G. J. Philips, M. T. Madzik, S. V. Amitonov, et al., Universal control of a six-qubit quantum processor in silicon, *Nature* 609, 919 (2022).
- B. Undseth, X. Xue, M. Mehmandoost, et al., Nonlinear Response and Crosstalk of Electrically Driven Silicon Spin Qubits, *Phys. Rev. Applied* 19, 044078 (2023).
- J. Yoneda, K. Takeda, T. Otsuka, et al., A quantum-dot spin qubit with coherence limited by charge noise and fidelity higher than 99.9%, *Nature Nanotechnology* 13, 102 (2018).